# A Probabilistic Representation for Deep Learning: Delving into The Information Bottleneck Principle

## Abstract

1  The Information Bottleneck (IB) principle has recently attracted great attention to
2  explaining Deep Neural Networks (DNNs), and the key is to accurately estimate the
3  mutual information between a hidden layer and dataset. However, some unsettled
4  limitations weaken the validity of the IB explanation for DNNs. To address these
5  limitations and fully explain deep learning in an information theoretic fashion, we
6  propose a probabilistic representation for deep learning that allows the framework
7  to estimate the mutual information, more accurately than existing non-parametric
8  models, and also quantify how the components of a hidden layer affect the mutual
9  information. Leveraging the probabilistic representation, we take into account the
10  back-propagation training and derive two novel Markov chains to characterize the
11  information flow in DNNs. We show that different hidden layers achieve different
12  IB trade-offs depending on the architecture and the position of the layers in DNNs,
13  whereas a DNN satisfies the IB principle no matter the architecture of the DNN.

## 1 Introduction

15  Deep learning [18] has already achieved great success in numerous applications. Deep Neural
16  Networks (DNNs), however, are still commonly viewed as 'black boxes' [27]. Considerable efforts
17  have been devoted to explaining the internal mechanism of DNNs from various perspectives, such as
18  mathematics [5, 12], statistics [14, 20, 23], computer vision [37, 21], *etc.* Recently, the Information
19  Bottleneck (IB) principle has attracted attention in opening the 'black boxes' of DNNs [30, 33].

20  Given a joint distribution $P(X, Y)$, the IB principle posits a random variable $T = f(X)$ obeying the
21  Markov chain $Y \rightarrow X \rightarrow T$ and optimizes $T$ by the IB Lagrangian [32, 31]

$$\min_{P(T|X)} I(X; T) - \beta I(Y; T), \tag{1}$$

22  where $f(\cdot)$ is an arbitrary function, $I(\cdot; \cdot)$ denotes mutual information, and the Lagrange multiplier
23  $\beta > 0$ controls the IB trade-off between compressing the input $X$ and preserving the information
24  of the label $Y$. In the seminal work [30], Tishby *et al.* manifest the IB trade-off in every layer of
25  DNNs $= \{\boldsymbol{x}; \boldsymbol{t}_1; \cdots; \boldsymbol{t}_I; \hat{\boldsymbol{y}}\}$ via studying $I(X; T_i)$ and $I(Y; T_i)$, where $T_i$ is the random variable of
26  the $i$th hidden layer $\boldsymbol{t}_i$. Especially, the authors ascribe DNN generalization to the compression [29].

27  In the context of deterministic DNNs, recent works reveal some limitations of the IB principle for
28  explaining DNNs. Amjad *et al.* argue that the IB principle becomes an ill-posed optimization problem
29  due to $I(X; T_i) = \infty$ [1], and Kolchinsky *et al.* demonstrate that not every layer of DNNs satisfies a
30  strict IB trade-off, *i.e.*, different layers only differ in $I(X; T_i)$ but $I(Y; T_i)$ keeps consistent in all
31  layers [15]. In addition, Saxe *et al.* experimentally show that the compression does not occur in
32  DNNs with non-saturating activation functions, *e.g.*, the popular ReLU function [28], and Goldfeld
33  *et al.* doubt the causality between the generalization of DNNs and the compression [10, 7]. These
34  unsettled limitations greatly weakens the validity of the IB explanations for DNNs.

The key to examining the IB principle in DNNs is the accurate estimation of the mutual information. However, regarding DNNs as deterministic models hinders us from specifying the random variable $T_i$ and the distribution $P(T_i)$, thus it is difficult to accurately estimate $I(X; T_i)$ and $I(Y; T_i)$. More specifically, in the absence of a clear definition of $T_i$, simply assuming the activations of $\boldsymbol{t}_i$ as the *i.i.d.* samples of $T_i$ induces $T_i$ being a continuous random variable and $I(X; T_i) = \infty$ in deterministic DNNs (see Appendix C in [28]). The complicated architecture of DNNs makes it challenging to specify $P(T_i)$. Therefore, most previous works have to indirectly estimate $P(T_i)$ via non-parametric models [35], such as the empirical distribution [30], Kernel Density Estimation (KDE) [28], and Gaussian convolution [10]. However, we experimentally confirm that classical non-parametric models derives poor mutual information estimation [24, 22] in DNNs, and one reason is because activations do not satisfy the *i.i.d.* prerequisite of non-parametric models (see Appendix G). In summary, the limitations mainly stem from the lack of an explicit probabilistic representation for deep learning.

The IB principle only formulates the information flow in DNNs $= \{\boldsymbol{x}, \boldsymbol{t}_1, \cdots, \boldsymbol{t}_I, \hat{\boldsymbol{y}}\}$ after training, and the corresponding Markov chain (see Fig. 1 in [30])

$$Y \to X \to T_1 \cdots \to T_I \to \hat{Y} \tag{2}$$

indicates that the information of $Y$ transfers to $T_i$ in the forward direction and $T_i$ receives the information of $Y$ only via $X$. However, training DNNs by the back-propagation [25] implies that the information of $Y$ transfers to $T_i$ in the backward direction during training and retains information in $T_i$ after training. Notably, Zhang *et al.* show that a DNN can fit labels well even using Gaussian noise as input to train the DNN [38], which implies that $T_i$ can directly receive the information of $Y$. Hence, the IB principle does not comprehensively characterize the information flow in DNNs.

To address the above limitations and comprehensively explain DNNs in an information theoretic fashion, we introduce the probability space $(\Omega_{T_i}, \mathcal{F}, P_{T_i})$ [6] for the $i$th hidden layer $\boldsymbol{t}_i$ in DNNs. Compared to previous works, the probability space $(\Omega_{T_i}, \mathcal{F}, P_{T_i})$ enables us to: (i) accurately estimate $I(X; T_i)$ and $I(Y; T_i)$ via specifying $T_i$ and $P(T_i)$, and (ii) quantify the effect of the architecture of $\boldsymbol{t}_i$ and the back-propagation on $I(X; T_i)$ and $I(Y; T_i)$ via explicitly modeling all the ingredients of $\boldsymbol{t}_i$, such as the activation function and the weights in a probabilistic way. To the best of our knowledge, this is the first time the probability space of a hidden layer in DNNs is as defined.

Leveraging $(\Omega_{T_i}, \mathcal{F}, P_{T_i})$, we derive information theoretic explanations for DNNs as follows:

- Two Markov chains[1] characterize the information flow in DNNs $= \{\boldsymbol{x}, \boldsymbol{t}_1, \cdots, \boldsymbol{t}_I, \hat{\boldsymbol{y}}\}$

$$\begin{aligned} \bar{X} \to T_1 \to \cdots \to T_I \to \hat{Y} \\ T_1 \leftarrow \cdots \leftarrow T_I \leftarrow \hat{Y} \leftarrow Y. \end{aligned} \tag{3}$$

- Different hidden layers manifest different IB trade-offs depending on the architecture and the position of hidden layers in DNNs.

- A DNN satisfies the IB principle no matter the architecture of the DNN.

**Preliminaries.** $P(X, Y) = P(X)P(Y|X)$ is an unknown joint distribution between $X$ and $Y$. A dataset $\mathcal{D} = \{(\boldsymbol{x}^j, y^j) | \boldsymbol{x}^j \in \mathbb{R}^M, y^j \in \mathbb{Z}\}_{j=1}^J$ consists of $J$ *i.i.d.* samples generated from $P(X, Y)$ with finite $L$ labels, i.e., $y^j \in \{1, \cdots, L\}$. In the context of supervised learning, we focus on feedfworad fully connected DNNs $= \{\boldsymbol{x}, \boldsymbol{t}_1, \cdots, \boldsymbol{t}_I, \hat{\boldsymbol{y}}\}$, *i.e.*, Multi-Layer Perceptions (MLPs) [8] for the image classification task. Without loss of generality, we use the MLP $= \{\boldsymbol{x}, \boldsymbol{t}_1, \boldsymbol{t}_2, \hat{\boldsymbol{y}}\}$ with the cross-entropy loss $\ell_{\text{CE}}$ for most theoretical derivations. In addition, $H(\cdot)$ denotes entropy.

In the MLP, $\boldsymbol{t}_1$ and $\boldsymbol{t}_2$ have $N$ and $K$ neurons, respectively, and $\boldsymbol{t}_1 = \{t_{1n} = \sigma_1[\langle \boldsymbol{\omega}_n^{(1)}, \boldsymbol{x} \rangle]\}_{n=1}^N$, where $\langle \boldsymbol{\omega}_n^{(1)}, \boldsymbol{x} \rangle = \sum_{m=1}^M \omega_{mn}^{(1)} \cdot x_m + b_{1n}$ is the $n$th dot-product given the weight $\omega_{mn}^{(1)}$ and the bias $b_{1n}$, and $\sigma_1(\cdot)$ denotes an activation function, *e.g.*, ReLU. Similarly, $\boldsymbol{t}_2 = \{t_{2k} = \sigma_2[\langle \boldsymbol{\omega}_k^{(2)}, \boldsymbol{t}_1 \rangle]\}_{k=1}^K$, where $\langle \boldsymbol{\omega}_k^{(2)}, \boldsymbol{t}_1 \rangle = \sum_{n=1}^N \omega_{nk}^{(2)} \cdot t_{1n} + b_{2k}$. The output layer $\hat{\boldsymbol{y}}$ is softmax with $L$ nodes

$$\hat{\boldsymbol{y}} = \{\hat{y}_l = \frac{1}{Z_Y} \exp[\langle \boldsymbol{\omega}_l^{(3)}, \boldsymbol{t}_2 \rangle] = \frac{1}{Z_Y} \exp[g_l(\boldsymbol{t}_2(\boldsymbol{t}_1(\boldsymbol{x})))]\}_{l=1}^L, \tag{4}$$

where $\langle \boldsymbol{\omega}_l^{(3)}, \boldsymbol{t}_2 \rangle = \sum_{k=1}^K \omega_{kl}^{(3)} \cdot t_{2k} + b_{yl}$ and $Z_Y = \sum_{l=1}^L \exp[\langle \boldsymbol{\omega}_l^{(3)}, \boldsymbol{t}_2 \rangle]$ is the partition function.

---

[1]In which the virtual random variable $\bar{X}$ has all the information of $X$ except $Y$, namely $H(\bar{X}) = H(X|Y)$.

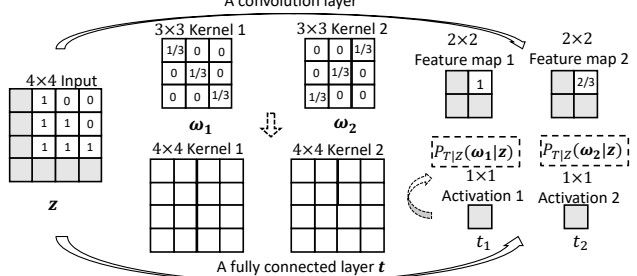

Figure 1: Given a $4 \times 4$ input $\boldsymbol{z}$, a fully connected layer $\boldsymbol{t}$ is equivalent to a convolution layer with $4 \times 4$ convolution kernels. The definition of convolution (Chapter 9.1 in [11]) implies that the $4 \times 4$ weights $\boldsymbol{\omega}_1$ and $\boldsymbol{\omega}_2$ define two global features, and the two activations $t_1, t_2$ indicate the cross-correlation between $\boldsymbol{\omega}_1, \boldsymbol{\omega}_2$ and $\boldsymbol{z}$, respectively. $P_{T|Z}(\boldsymbol{\omega}_1|\boldsymbol{z})$ and $P_{T|Z}(\boldsymbol{\omega}_2|\boldsymbol{z})$ measure the probability of $\boldsymbol{\omega}_1$ and $\boldsymbol{\omega}_2$ being recognized as the feature with the largest cross-correlation to $\boldsymbol{z}$, respectively.

## 2 A probabilistic representation for deep learning

To accurately estimate $I(X; T_i)$ and $I(Y; T_i)$, in this section, we specify the probability space [6] for a fully connected layer and derive the probabilistic explanations of the entire MLP.

It is known that a convolution kernel (namely the weights of convolution) defines a local feature, and a convolution operation derives a feature map to measure the cross-correlation between the local feature and input in a receptive field (Chapter 9.1 in [11]). Notably, a fully connected layer is equivalent to a convolution layer with the kernel size having the same dimension as input. Thus the weights of a neuron can be viewed as a global feature, and a fully connected layer with multiple neurons derives activations to measure the cross-correlation between the multiple global features and the input. The cross-correlation explanation for a fully connected layer is visualized in Figure 1.

Assuming that a fully connected layer $\boldsymbol{t}$ consists of $N$ neurons $\{t_n = \sigma[\langle \boldsymbol{\omega}_n, \boldsymbol{z} \rangle]\}_{n=1}^N$, where $\boldsymbol{z} \in \mathbb{R}^M$ is the input of $\boldsymbol{t}$, $\langle \boldsymbol{\omega}_n, \boldsymbol{z} \rangle = \sum_{m=1}^M \omega_{mn} \cdot z_m + b_n$ is the dot-product between $\boldsymbol{z}$ and $\boldsymbol{\omega}_n$, and $\sigma(\cdot)$ is an activation function. Based on the cross-correlation explanation, the behavior of $\boldsymbol{t}$ is to measure the cross-correlations between $\boldsymbol{z}$ and the $N$ possible features defined by the the weights $\{\boldsymbol{\omega}_n\}_{n=1}^N$. In the context of pattern recognition [34], we define a virtual random process or 'experiment' as $\boldsymbol{t}$ recognizing one of the patterns/features with the largest cross-correlation to $\boldsymbol{z}$ from the $N$ possible features. The experiment characterizes the behavior of $\boldsymbol{t}$ (*i.e.*, before recognizing the features with the largest cross-correlation, $\boldsymbol{t}$ must measure the cross-correlations between $\boldsymbol{z}$ and all the $N$ possible features) while meets the requirement of the 'experiment' definition (*i.e.*, only one outcome will occur on each trial of the experiment [6]). The probability space $(\Omega_T, \mathcal{F}, P_T)$ is defined as follows:

**Definition 1.** $(\Omega_T, \mathcal{F}, P_T)$ consists of three components: the sample space $\Omega_T$ has $N$ possible outcomes (features) $\{\boldsymbol{\omega}_n = \{\omega_{mn}\}_{m=1}^M\}_{n=1}^N$ defined by the weights[2] of the $N$ neurons; the event space $\mathcal{F}$ is the $\sigma$-algebra; and the probability measure $P_T$ is a Gibbs distribution [19] to quantify the probability of $\boldsymbol{\omega}_n$ being recognized as the feature with the largest cross-correlation to $\boldsymbol{z}$.

Taking into account the randomness of $\boldsymbol{z}$, the conditional distribution $P_{T|Z}$ is formulated as

$$P_{T|Z}(\boldsymbol{\omega}_n | \boldsymbol{z}) = \frac{1}{Z_T} \exp(t_n) = \frac{1}{Z_T} \exp[\sigma(\langle \boldsymbol{\omega}_n, \boldsymbol{z} \rangle)], \tag{5}$$

where $Z$ is the random variable of $\boldsymbol{z}$ and $Z_T = \sum_{n=1}^N \exp(f_n)$ is the partition function.

$(\Omega_T, \mathcal{F}, P_T)$ clearly explains all the ingredients of $\boldsymbol{t}$ in a probabilistic fashion. The $n$th neuron defines a global feature by the weights $\boldsymbol{w}_n$ and the activation $t_n = \sigma(\langle \boldsymbol{\omega}_n, \boldsymbol{z} \rangle)$ measures the cross-correlation between $\boldsymbol{w}_n$ and $\boldsymbol{z}$. The Gibbs distribution $P_{T|Z}$ indicates that if $\boldsymbol{w}_n$ has the higher activation, *i.e.*, the larger cross-correlation to $\boldsymbol{z}$, it has the larger probability being recognized as the feature with largest cross-correlation to $\boldsymbol{z}$. For instance, if $\boldsymbol{z} \in \mathbb{R}^{16}$ and $\boldsymbol{t}$ includes $N = 2$ neurons, then $\Omega_T = \{\boldsymbol{\omega}_1, \boldsymbol{\omega}_2\}$ defines two possible outcomes (features), where $\boldsymbol{\omega}_n = \{\omega_{mn}\}_{m=1}^{16}$. $\mathcal{F} = \{\emptyset, \{\boldsymbol{\omega}_1\}, \{\boldsymbol{\omega}_2\}, \{\boldsymbol{\omega}_1, \boldsymbol{\omega}_2\}\}$ means that neither, one, or both of the features are recognized by $\boldsymbol{t}$ given $\boldsymbol{z}$, respectively. $P_{T|Z}(\boldsymbol{\omega}_1 | \boldsymbol{z})$ and $P_{T|Z}(\boldsymbol{\omega}_2 | \boldsymbol{z})$ are the probability of $\boldsymbol{\omega}_1$ and $\boldsymbol{\omega}_2$ being recognized as the feature with the largest cross-correlation to $\boldsymbol{z}$, respectively.

---

[2]We do not take into account the scalar value $b_n$ for defining $\Omega_T$, as it not affects the feature defined by $\boldsymbol{\omega}_n$.

$(\Omega_T, \mathcal{F}, P_T)$ explains the representation ability of deep learning. Compared to Restricted Boltzmann Machines (RBMs) [26] simply using binary units to indicate features being recognized or not given input, the Gibbs distribution[3] $P_{T|Z}(\boldsymbol{\omega}_n|\boldsymbol{z})$ measures the probability of $\boldsymbol{\omega}_n$ being recognized with the largest cross-correlation to $\boldsymbol{z}$, i.e., it characterizes the relation between features and input more accurately. Moreover, Equation 5 shows that $t_n = \sigma(\langle\boldsymbol{\omega}_n, \boldsymbol{z}\rangle)$ is the negative energy function [19] of the Gibbs distribution, thus $P_{T|Z}(\boldsymbol{\omega}_n|\boldsymbol{z})$ can be derived as long as $\sigma(\langle\boldsymbol{\omega}_n, \boldsymbol{z}\rangle)$ are known because the energy function is the sufficient statistics [2] of the Gibbs distribution. That enables subsequent hidden layers to generate high-level features of input via directly processing the activations $\{t_n\}_{n=1}^{N}$, thus deep learning can form a hierarchical structure to represent much complex features.

$(\Omega_T, \mathcal{F}, P_T)$ answers a fundamental question: which component of a hidden layer contains the information of the layer? Since $\boldsymbol{\omega}_n$ defines $\Omega_T$, the weights contain all the information of a layer. In particular, since the activation $t_n = \sigma(\langle\boldsymbol{\omega}_n, \boldsymbol{z}\rangle)$ is a function of $\boldsymbol{\omega}_n$, the data processing inequality [4] indicates that the information of $t_n$ is no more than the information of $\boldsymbol{\omega}_n$. Simulations in Section 4.2 demonstrate that if activations do not correctly characterize the cross-correlation between weights and input, activations contain less information than weights do.

Based on $(\Omega_T, \mathcal{F}, P_T)$, we define the random variable $T$ as follows:

**Definition 2.** Given the fully connected layer $\boldsymbol{t}$, we define the random variable $T : \Omega_T \rightarrow E_T$ as

$$T(\boldsymbol{\omega}_n) \triangleq n, \tag{6}$$

where the measurable space $E_T = \{1, \cdots, N\}$.

Since $\Omega_T$ is composed of finite $N$ possible outcomes, $T$ is a discrete random variable. Notably, the one-to-one correspondence between $\boldsymbol{\omega}_n$ and $n$ indicates

$$P_{T|Z}(\boldsymbol{\omega}_n|\boldsymbol{z}) = P_{T|Z}(n|\boldsymbol{z}). \tag{7}$$

If not considering the back-propagation training, the weights (namely $\Omega_{T_i}$) of each layer are fixed. Thus $T_{i+1}$ entirely depends on $T_i$ and the MLP $= \{\boldsymbol{x}; \boldsymbol{t}_1; \boldsymbol{t}_2; \hat{\boldsymbol{y}}\}$ forms a Markov chain

$$X \rightarrow T_1 \rightarrow T_2 \rightarrow \hat{Y}. \tag{8}$$

Based on the corresponding joint distribution $P(\hat{Y}, T_2, T_1|X) = P(T_1|X)P(T_2|T_1)P(\hat{Y}|T_2)$ and Definition 2, we derive a probabilistic explanation for the entire MLP, which is summarized in Theorem 1. The detailed derivation is presented in Appendix B.

**Theorem 1.** The MLP $= \{\boldsymbol{x}; \boldsymbol{t}_1; \boldsymbol{t}_2; \hat{\boldsymbol{y}}\}$ formulates a conditional Gibbs distribution

$$P_{\hat{Y}|X}(l|\boldsymbol{x}) = \sum_{k=1}^{K}\sum_{n=1}^{N} P(\hat{Y} = l, T_2 = k, T_1 = n|X = \boldsymbol{x}) = \frac{1}{Z_{\text{MLP}}(\boldsymbol{x})}\exp[g_l(\boldsymbol{t}_2(\boldsymbol{t}_1(\boldsymbol{x})))], \tag{9}$$

where $Z_{\text{MLP}}(\boldsymbol{x}) = \sum_{l=1}^{L}\sum_{k=1}^{K}\sum_{n=1}^{N} P_{\hat{Y}, T_2, T_1|X}(l, k, n|x)$ is the partition function.

Since $P_{\hat{Y}|X}(l|\boldsymbol{x})$ exactly equals the output $\hat{y}_l$ of the MLP, namely Equation (4), we conclude that the entire architecture of the MLP forms a family of Gibbs distribution $P_{\hat{Y}|X}(l|\boldsymbol{x})$. In general, the back-propagation updates a weight $\omega$ based on the gradient of $\ell_{\text{CE}}$ with respect to $\omega$,

$$\omega(s+1) = \omega(s) - \alpha \cdot \frac{\partial\ell_{\text{CE}}}{\partial\omega(s)} = \omega(s) - \alpha \cdot \frac{\partial\text{KL}[P(Y|X)||P(\hat{Y}|X)]}{\partial\omega(s)}, \tag{10}$$

where $s$ is the index of training iteration, $\alpha$ is the training rate, and $\text{KL}[\cdot||\cdot]$ is the KL-divergence.

Figure 2 summarizes the probabilistic explanation for deep learning based on the MLP. In general, a single learning iteration, an epoch, consists of two phases: training and inference (after training). During inference, the MLP bridges $X$ and $\hat{Y}$ via multiple intermediate features $\Omega_{T_1}$, $\Omega_{T_2}$, and $\Omega_{\hat{Y}}$ defined by weights, and formulates the statistical relation between $\hat{Y}$ and $X$ as a family of conditional Gibbs distribution $P(\hat{Y}|X)$. During training, the back-propagation updates weights to learn optimal intermediate features for searching an optimal $P(\hat{Y}|X)$ to accurately approximate $P(Y|X)$.

---

[3]Recent works about Gibbs explanations for a hidden layer are discussed in Appendix A.

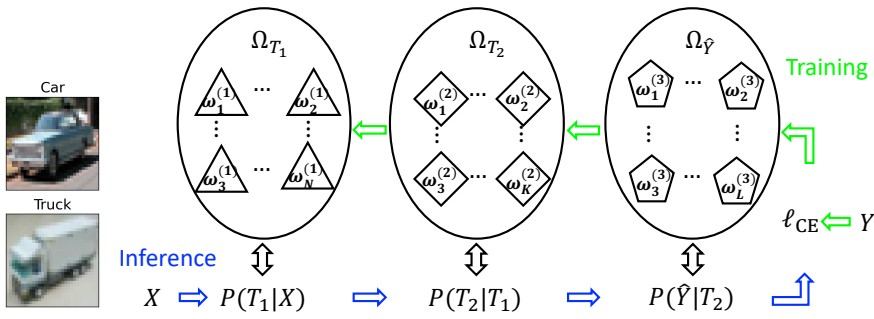

Figure 2: The visualization of the probabilistic explanation for deep learning based on the MLP.

## 3 The information theoretic explanations for deep learning

To address the limitations of existing IB explanations, this section proposes some novel information theoretic explanations for DNNs based on the proposed probabilistic representation.

**Proposition 1.** The mutual information between a fully connected layer and dataset is finite.

$$I(X;T) < \infty. \tag{11}$$

*Proof:* Definition 2 shows $E_T = \{1, \cdots N\}$. Thus $T$ is a discrete random variable and $H(T) < \infty$, thereby $I(X;T) \leq H(T) < \infty$.

Proposition 1 circumvents the infinite mutual information problem. In the absence of a clear definition $T : \Omega_T \to E_T$, most previous works [28, 3, 1] simply viewing the activation $t_n$ as the sample of $T$, namely $t_n \in E_T = \mathbb{R}$, implies $T$ being continuous and gives rise to the infinite mutual information problem in deterministic DNNs. However, $(\Omega_T, \mathcal{F}, P_T)$ indicates that $t_n$ actually is a variable measuring the cross-correlation between $\boldsymbol{w}_n$ and $\boldsymbol{z}$ rather than the sample of $T$, namely $t_n \notin E_T$.

**Theorem 2.** The information of $Y$ flows into the MLP in the backward direction during training

$$T_1 \leftarrow T_2 \leftarrow \hat{Y} \leftarrow Y. \tag{12}$$

*Proof:* First, since $\Omega_T$ is defined by $\omega$ in $(\Omega_T, \mathcal{F}, P_T)$ and Equation (10) shows that $\omega(s+1)$ is determined by all the previous gradients $\{\frac{\partial \ell_{CE}}{\partial \omega(s)}\}_{s=1}^{S}$, and $\omega(0)$ is randomly initialized and $\alpha$ is a constant, we can derive that $\Omega_T$ is determined by $\frac{\partial \ell_{CE}}{\partial \omega}$. Second, based on the back-propagation, the relation between gradients in two adjacent layers in the MLP $= \{\boldsymbol{x}; \boldsymbol{t}_1; \boldsymbol{t}_2; \hat{\boldsymbol{y}}\}$ is formulated as

$$\frac{\partial \ell_{CE}^{\star}}{\partial \omega_{kl}^{(3)}} = [P_{\hat{Y}|X}(l|\boldsymbol{x}) - P_{Y|X}(l|\boldsymbol{x})] \cdot t_{2k},$$

$$\frac{\partial \ell_{CE}^{\odot}}{\partial \omega_{nk}^{(2)}} = \sum_{l=1}^{L} \frac{\partial \ell_{CE}^{\star}}{\partial \omega_{kl}^{(3)}} \cdot \omega_{kl}^{(3)} \cdot \frac{\sigma_2'(\langle \boldsymbol{\omega}_k^{(2)}, \boldsymbol{t}_1 \rangle)}{f_{2k}} \cdot t_{1n}, \quad \frac{\partial \ell_{CE}^{\diamond}}{\partial \omega_{mn}^{(1)}} = \sum_{k=1}^{K} \frac{\partial \ell_{CE}^{\odot}}{\partial \omega_{nk}^{(2)}} \cdot \omega_{nk}^{(2)} \cdot \frac{\sigma_1'(\langle \boldsymbol{\omega}_n^{(1)}, \boldsymbol{x} \rangle)}{t_{1n}} \cdot x_m. \tag{13}$$

Equation 13 shows that $\frac{\partial \ell_{CE}}{\partial \omega^{(3)}}$ is a function of $P_{Y|X}(l|\boldsymbol{x})$ and $\frac{\partial \ell_{CE}}{\partial \omega^{(i)}}$ is a function of $\frac{\partial \ell_{CE}}{\partial \omega^{(i+1)}}$, where $\omega^{(3)}$ denotes the weight of $\hat{\boldsymbol{y}}$. The two points above enable us to derive that $\Omega_{T_i}$ is a function of $\Omega_{T_{i+1}}$ and $\Omega_{\hat{Y}}$ is a function of $P(Y|X)$. Based on Definition 2, we can further derive that $T_i$ is a function of $T_{i+1}$ and $\hat{Y}$ is a function of $Y$, *i.e.*, $T_1 \leftarrow T_2 \leftarrow \hat{Y} \leftarrow Y$. (See the detailed proof in Appendix C).

Theorem 2 is consistent with the prevailing explanation for deep learning. LeCunn *et al.* show that deep learning exploits the hierarchical property of signals [18], *i.e.*, the layers farther from output learn lower-level features, such as edges, whereas the layers closer to output assemble lower-level features into the higher-level features corresponding to labels (see Figure 2 in [37]). Notably, since lower-level features commonly exist in signals with different labels (*e.g.*, lower-level features, such as the edges of the vehicle frame and the circular contour of wheels, exist in both the car and the truck classes in the CIFAR-10 dataset [16] in Figure 2), lower-level features do not contain much information of labels. Therefore, the layers farther from output do not have much information of labels, which is consistent with the Markov chain $T_1 \leftarrow T_2 \leftarrow \hat{Y} \leftarrow Y$.

Since all the information of $Y$ stems from $X$ (*i.e.*, $H(Y) = I(X;Y)$ proven in Appendix D), Theorem 2 implies that partial information of $X$ flows into the MLP in the backward direction during training. Equation (2) shows the information of $X$ flowing into the MLP in the forward direction during inference. Overall, the information of $X$ flows in the backward and forward directions during training and inference, respectively. As a result, the Markov chain, Equation (2), proposed by recent works could not fully characterize the information flow of $X$ in the MLP in each epoch. In other words, $I(X;T_i)$ is not necessarily greater than $I(X;T_{i+1})$ in the MLP in each epoch.

Equation (2) shows that $T_i$ receives the information of $Y$ via $X$ during inference. Theorem 2 shows that $T_i$ also directly receives information of $Y$ during training, because the back-propagation updates weights (i.e., $\Omega_{T_i}$) based on the label $Y$. Thus Equation (2) cannot fully characterize the information flow of $Y$ in the MLP in each epoch, when we take into account the back-propagation training.

To fully characterize the information flow in the MLP in each epoch, we introduce Corollary 1.

**Corollary 1.** The information flow in the MLP can be characterized by two Markov chains as

$$\bar{X} \to T_1 \to T_2 \to \hat{Y}$$
$$T_1 \leftarrow T_2 \leftarrow \hat{Y} \leftarrow Y. \tag{14}$$

The virtual random variable $\bar{X}$ contains all the information of $X$ except $Y$, *i.e.*, $H(\bar{X}) = H(X|Y)$.

*Proof of the first Markov chain:* Since $\bar{X}$ does not have any information of $Y$, it can only flow into the MLP in the forward direction during inference. Again since $\bar{X}$ does not have any information of $Y$, the information flow of $Y$ during training will not affect the information flow of $\bar{X}$. Therefore, $\bar{X} \to T_1 \to T_2 \to \hat{Y}$ characterizes the information flow of $\bar{X}$ in both training and inference phases.

*Proof of the second Markov chain:* Since the weights are fixed after training, the sample space and the distribution of hidden layers are fixed after training. Therefore, the information of $Y$ transferred into hidden layers during training will retain there after training (*i.e.*, during inference). In addition, Definition 1 indicates that a fully connected layer $\boldsymbol{t} = \{t_n = \sigma(\langle \boldsymbol{\omega}_n, \boldsymbol{z}\rangle)\}_{n=1}^{N}$ measures the cross-correlation between $\boldsymbol{\omega}_n^{(1)}$ and $\boldsymbol{z}$ during inference, thus $\{\boldsymbol{\omega}_n^{(1)}\}_{n=1}^{N}$ can be viewed as a representation of $Z$. As a result, even though $Z$ has all the information of $Y$, the information of $Y$ that $\boldsymbol{t}$ can learn from $Z$ is determined by how much information of $Y$ the representation $\{\boldsymbol{\omega}_n^{(1)}\}_{n=1}^{N}$ has. Overall, the information flow of $Y$ during inference will be the same as that during training. Based on Theorem 2, we conclude that $T_1 \leftarrow T_2 \leftarrow \hat{Y} \leftarrow Y$ characterizes the information flow of $Y$ in the MLP in both training and inference phases. Detailed derivations and explanations are presented in Appendix E.

To quantify how much information of $X$ and $Y$ is learned by the MLP, we introduce Corollary 2.

**Corollary 2.** The mutual information between dataset and the entire MLP can be expressed as

$$I(X; T_{\text{MLP}}) = I(\bar{X}; T_1) + I(Y; \hat{Y})$$
$$I(Y; T_{\text{MLP}}) = I(Y; \hat{Y}) \tag{15}$$

where $T_{\text{MLP}}$ denotes a random variable corresponding to the entire architecture of the MLP.

*Proof:* Since $H(Y) = I(X;Y)$ (Appendix D), $H(X) = H(\bar{X}) + I(X;Y) = H(\bar{X}) + H(Y)$. Hence, Corollary 2 can be derived by Corollary 1 and the chain rule. The proof is in Appendix F.

# 4  Simulations

In this section, we propose a mutual information estimator based on $(\Omega_T, \mathcal{F}, P_T)$ and demonstrate the probabilistic representation and information theoretic explanations for deep learning on a synthetic dataset with known entropy. Additional experiments on benchmark datasets are in Appendix H.

## 4.1  Setup

**Mutual information estimator.**  Based on the definition of mutual information, we have

$$I(X; T_i) = H(T_i) - H(T_i|X). \tag{16}$$

Previous works simply estimate $I(X;T_i) = H(T_i)$, because $T_i$ is assumed to be entirely dependent on $X$ in the Markov chain, Equation (2), thereby $H(T_i|X) = 0$. However, Corollary 1 shows that $T_i$ depends on both $X$ and $Y$ if taking into account the training phase, thereby $H(T_i|X) \neq 0$.

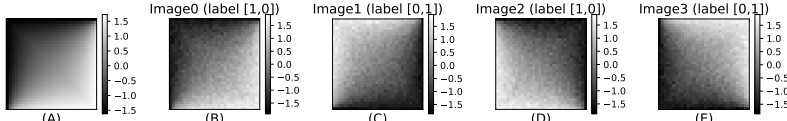

Figure 3: (A) the deterministic image $\hat{\boldsymbol{x}}$. Image0 is generated by adding $\mathcal{N}(\mu, \sigma^2)$ without rotation, Image1 is generated by rotating $\hat{\boldsymbol{x}}$ along the secondary diagonal direction and adding $\mathcal{N}(\mu, \sigma^2)$, Image2 and Image are generated by rotating $\hat{\boldsymbol{x}}$ along the vertical and horizontal directions, respectively, and adding $\mathcal{N}(\mu, \sigma^2)$.

Table 1: The number of neurons(nodes) and the activation function in the layers of the MLPs

|  | $\boldsymbol{x}$ | $\boldsymbol{t}_1$ | $\boldsymbol{t}_2$ | $\hat{\boldsymbol{y}}$ | $\sigma(\cdot)$ |
|---|---|---|---|---|---|
| MLP1 | $1024\ (32 \times 32)$ | 8 | 6 | 2 | $\text{ReLU}(z) = \max(0, z)$ |
| MLP2 | $1024\ (32 \times 32)$ | 8 | 6 | 2 | $\text{Tanh}(z) = (e^z - e^{-z})/(e^z + e^{-z})$ |
| MLP3 | $1024\ (32 \times 32)$ | 2 | 6 | 2 | ReLU |

To accurately estimate $I(X; T_i)$, we need to specify $P(T_i|X)$ and $P(T_i)$. Based on $(\Omega_{T_i}, \mathcal{F}, P_{T_i})$, we formulate $P_{T_i|X}(n|\boldsymbol{x}^j)$ of the three fully connected layers in the MLP as

$$P_{T_1|X}(n|\boldsymbol{x}^j) = \tfrac{1}{Z_{F_1}}\exp[\sigma_1(\langle \boldsymbol{\omega}_n^{(1)}, \boldsymbol{x}^j \rangle)], \ \ P_{T_2|X}(k|\boldsymbol{x}^j) = \tfrac{1}{Z_{F_2}}\exp[\sigma_2(\langle \boldsymbol{\omega}_k^{(2)}, \boldsymbol{t}_1(\boldsymbol{x}^j) \rangle)],$$
$$P_{T_Y|X}(l|\boldsymbol{x}^j) = \tfrac{1}{Z_{F_Y}}\exp[\langle \boldsymbol{\omega}_l^{(3)}, \boldsymbol{t}_2(\boldsymbol{t}_1(\boldsymbol{x}^j)) \rangle]. \tag{17}$$

To derive the marginal distribution $P(T_i)$, we sum the joint distribution $P(T_i, X)$ over $\boldsymbol{x} \in \mathcal{X}$,

$$P(T_i = n) = \sum_{\boldsymbol{x} \in \mathcal{X}} P_X(\boldsymbol{x}) P_{T_i|X}(n|\boldsymbol{x}) \approx \sum_{\boldsymbol{x}^j \in \mathcal{D}} P_X(\boldsymbol{x}^j) P_{T_i|X}(n|\boldsymbol{x}^j) = \tfrac{1}{J} \sum_{\boldsymbol{x}^j \in \mathcal{D}} P_{T_i|X}(n|\boldsymbol{x}^j), \tag{18}$$

where $P_X(\boldsymbol{x}^j)$ is estimated by the empirical distribution $1/J$ given $\mathcal{D}$. Finally, we can derive $I(X; T_i)$ by Equation 16, 17, and 18. Similarly, based on the definition of mutual information, we have

$$I(Y; T_i) = H(T_i) - H(T_i|Y). \tag{19}$$

To estimate $H(T_i|Y)$, we reformulate $P(T_i|Y)$ as

$$P_{T_i|Y}(n|l) = \sum_{\boldsymbol{x} \in \mathcal{X}} P_{T_i|X}(n|\boldsymbol{x}) P_{X|Y}(\boldsymbol{x}|l) \approx \tfrac{1}{N(l)} \sum_{\boldsymbol{x}^j \in \mathcal{D}, y^j = l} P_{T_i|\boldsymbol{X}}(n|\boldsymbol{x}^j), \tag{20}$$

where $P_{X|Y}(\boldsymbol{x}^j|l)$ is estimated by the empirical distribution $1/N(l)$ and $N(l)$ denotes the number of samples with the label $l$ in $\mathcal{D}$. Finally, we can derive $I(Y; T_i)$ by Equation 18, 19, and 20.

**Synthetic dataset.** The dataset consists of 512 gray-scale $32 \times 32$ images, which are evenly generated by rotating a deterministic image $\hat{\boldsymbol{x}}$ in four different orientations and adding Gaussian noise with expectation $\mu = \mathbb{E}(\hat{\boldsymbol{x}})$ and variance $\sigma^2 = 1$, namely $\boldsymbol{x} = r(\hat{\boldsymbol{x}}) + \mathcal{N}(\mu, \sigma^2)$, where $r(\cdot)$ denotes the rotation method shown in Figure 3. The reason for adding Gaussian noise is to avoid DNNs directly memorizing the deterministic image. In addition, the binary labels [1,0] and [0,1] evenly divide the synthetic dataset into two classes. As a result, the synthetic dataset has (approximately) 2 bits information and the labels have 1 bit information. Compared to popular benchmark dataset with unknown features and entropy, *e.g.*, MNIST [17] and Fashion-MNIST [36], the features and the entropy of the synthetic dataset are clear and known, which enables us to examine the probabilistic representation and the mutual information estimator.

**Neural Networks.** We train three MLPs, namely MLP1, MLP2 and MLP3, on the synthetic dataset by a variant of Stochastic Gradient Descent (SGD) method, namely Adam [13], over 1000 epochs with the learning rate $\alpha = 0.03$. Table 1 summarizes the architecture of the three MLPs.

### 4.2 Validating the probability space and the mutual information estimator

We demonstrate the sample space $\Omega_T$ by visualizing the weights[4] of the eight neurons in $\boldsymbol{t}_1$, *i.e.*, $\boldsymbol{\omega}_n^{(1)} = \{\omega_{mn}^{(1)}\}_{m=1}^{1024}$, in 5 different epochs (*i.e.*, 0,1,4,128,1000) in Figure 4 (Left). As training continues, we observe that $\boldsymbol{\omega}_n^{(1)}$ quickly learns all the spatial features of the synthetic dataset. For instance, $\boldsymbol{\omega}_2^{(1)}$ has low magnitude at top-left positions and high magnitude at bottom-right positions, which correctly characterizes the spatial feature of Image0. Similarly, $\boldsymbol{\omega}_3^{(1)}$, $\boldsymbol{\omega}_4^{(1)}$, and $\boldsymbol{\omega}_5^{(1)}$ correctly characterize the spatial feature of Image1, Image2, and Image3 in Figure 3, respectively.

---

[4]We only show the learned weights in MLP1 because we observe that the learned weights in MLP1 and MLP2 are very similar, though they use different activation functions.

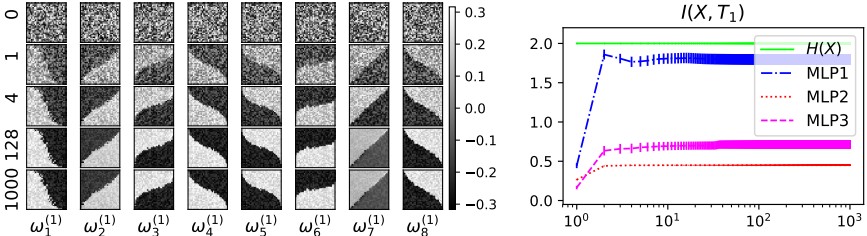

Figure 4: (Left) The eight features $\{\boldsymbol{\omega}_n^{(1)}\}_{n=1}^8$ learned by the weights of the eight neurons in 5 different epochs (*i.e.*, 0,1,4,128,1000), where $\boldsymbol{\omega}_n^{(1)} = \{\omega_{mn}^{(1)}\}_{m=1}^{1024}$ are reshaped into $32 \times 32$ to show the spatial structure. (Right) The variation of $I(X;T_1)$ in the MLP1, MLP2, and MLP3 during 1000 epochs.

Table 2: The Gibbs probability $P_{F_1|X}(\boldsymbol{\omega}_n^{(1)}|\text{Image0})$ in MLP1 and MLP2 in the 1000 epoch

| | $\boldsymbol{\omega}_1^{(1)}$ | $\boldsymbol{\omega}_2^{(1)}$ | $\boldsymbol{\omega}_3^{(1)}$ | $\boldsymbol{\omega}_4^{(1)}$ | $\boldsymbol{\omega}_5^{(1)}$ | $\boldsymbol{\omega}_6^{(1)}$ | $\boldsymbol{\omega}_7^{(1)}$ | $\boldsymbol{\omega}_8^{(1)}$ |
|---|---|---|---|---|---|---|---|---|
| $\langle \boldsymbol{\omega}_n^{(1)}, \boldsymbol{x} \rangle$ | -63.6 | **208.8** | -181.6 | 45.1 | -55.6 | 157.5 | -210.0 | -30.1 |
| $f_{1n}^{\text{ReLU}}(\boldsymbol{x})$ | 0.0 | **208.8** | 0.0 | 45.1 | 0.0 | 157.5 | 0.0 | 0.0 |
| $\exp[f_{1n}^{\text{ReLU}}(\boldsymbol{x})]$ | 1.0 | **4.79e+90** | 1.0 | 3.86e+19 | 1.0 | 2.51e+68 | 1.0 | 1.0 |
| $P_{T_1|X}^{\text{ReLU}}$ | 0.0 | **1.0** | 0.0 | 0.0 | 0.0 | 0.0 | 0.0 | 0.0 |
| $f_{1n}^{\text{Tanh}}(\boldsymbol{x})$ | -1.0 | **1.0** | -1.0 | **1.0** | -1.0 | **1.0** | -1.0 | -1.0 |
| $\exp[f_{1n}^{\text{Tanh}}(\boldsymbol{x})]$ | 0.36 | **2.71** | 0.36 | **2.71** | 0.36 | **2.71** | 0.36 | 0.36 |
| $P_{T_1|X}^{\text{Tanh}}$ | 0.037 | **0.272** | 0.037 | **0.272** | 0.037 | **0.272** | 0.037 | 0.037 |

$f_{1n}^{\text{Tanh}}(\boldsymbol{x}) = \sigma^{\text{Tanh}}(\langle \boldsymbol{\omega}_n^{(1)}, \boldsymbol{x} \rangle)$ and $f_{1n}^{\text{ReLU}}(\boldsymbol{x}) = \sigma^{\text{ReLU}}(\langle \boldsymbol{\omega}_n^{(1)}, \boldsymbol{x} \rangle)$ are the activations given the same $\langle \boldsymbol{\omega}_n^{(1)}, \boldsymbol{x} \rangle$.

249 We demonstrate that $P(T_1|X)$ correctly measures the probability of $\{\boldsymbol{\omega}_n^{(1)}\}_{n=1}^8$ being recognized the
250 feature with the largest cross-correlation to $\boldsymbol{x}$ in Table 2. For instance, $\boldsymbol{\omega}_2^{(1)}$ correctly characterizes
251 the feature of Image0 and has the largest cross-correlation $\langle \boldsymbol{\omega}_2^{(1)}, \boldsymbol{x} \rangle = 190.8$, thus it has the largest
252 probability $P_{T_1|X}^{\text{ReLU}}(\boldsymbol{\omega}_2^{(1)}|\text{Image0}) = 1.0$ being recognized as the feature with largest cross-correlation
253 to Image0. In contrast, since $\boldsymbol{\omega}_7^{(1)}$ incorrectly characterizes the feature of Image0 and has the lowest
254 cross-correlation $\langle \boldsymbol{\omega}_7^{(1)}, \boldsymbol{x} \rangle = -210.0$, so it has the lowest probability $P_{T_1|X}^{\text{ReLU}}(\boldsymbol{\omega}_7^{(1)}|\text{Image0}) = 0.0$
255 being recognized as the feature with largest cross-correlation to Image0.

256 We observe that an activation function (abbr. ACT) plays an important role in the distribution.
257 Specifically, ReLU, a non-saturating (unbounded) ACT [9], preserves the positive cross-correlations
258 while resets all the negative ones as zero. $P_{T_1|X}^{\text{ReLU}}(\boldsymbol{\omega}_2^{(1)}|\text{Image0}) = 1.0$ shows that ReLU derives the
259 correct probability of $\boldsymbol{\omega}_2^{(1)}$ being recognized as the feature with largest cross-correlation. In contrast,
260 though $\boldsymbol{\omega}_2^{(1)}$ has stronger cross-correlation to Image0 than $\boldsymbol{\omega}_4^{(1)}$, *i.e.*, $\langle \boldsymbol{\omega}_2^{(1)}, \boldsymbol{x} \rangle > \langle \boldsymbol{\omega}_4^{(1)}, \boldsymbol{x} \rangle$, Tanh, a
261 saturating (bounded) ACT, derives $f_{12}^{\text{Tanh}}(\boldsymbol{x}) = f_{14}^{\text{Tanh}}(\boldsymbol{x}) = 1.0$, and makes $\boldsymbol{\omega}_4^{(1)}$ to incorrectly have
262 the same probability $0.272$ to $\boldsymbol{\omega}_2^{(1)}$ being recognized as the feature with the largest cross-correlation
263 to Image0, *i.e.*, Tanh hinders $\boldsymbol{t}_1$ from correctly recognizing the features of input. The simulations for
264 validating the probability space based on other synthetic images are presented in Appendix G.

265 To validate the mutual information estimator, we follow recent works [30, 28] to train the three
266 MLPs with 50 different random initialization and study the average mutual information. Figure 4
267 (Right) shows that $I(X;T_1)$ quickly increases to 1.81 and keeps stable in the MLP1, *i.e.*, $\boldsymbol{t}_1$ learns
268 most information of the dataset as $H(X) = 2.0$. Notably, the result is consistent with the variation
269 of the weights in Figure 4 (Left), which shows that the weights correctly characterize the features
270 of the dataset and keeps stable after the fourth epoch. As a comparison, we observe that $I(X;T_1)$
271 keeps stable at 0.44 in the MLP2, which confirms the statement that Tanh hinders $\boldsymbol{t}_1$ from correctly
272 recognizing the features of input. In addition, Figure 4 (Right) shows that $I(X;T_1) \approx 0.79$ in MLP3
273 is smaller than $I(X;T_1) \approx 1.81$ in MLP1, which is consistent with Definition 1, *i.e.*, a layer with
274 fewer neurons would represent fewer possible features, thus it contains less information.

275 In summary, we demonstrate the probability space $(\Omega_T, \mathcal{F}, P_T)$ and show that if an ACT cannot
276 preserve the cross-correlation between weights(features) and input, it would distort the distribution
277 of a layer, thereby affecting the mutual information between the layer and data/labels. In addition,
278 we show that the proposed mutual information estimator outperforms the existing non-parametric
279 models, *e.g.*, empirical distribution [30] and KDE [28], based on the synthetic dataset. Especially,
280 activations do not satisfy the *i.i.d.* prerequisite of non-parametric models is an important reason for
281 non-parametric models deriving inaccurate mutual information in DNNs. Due to limited space, the
282 experimental comparison and study of non-parametric models are presented in Appendix G.

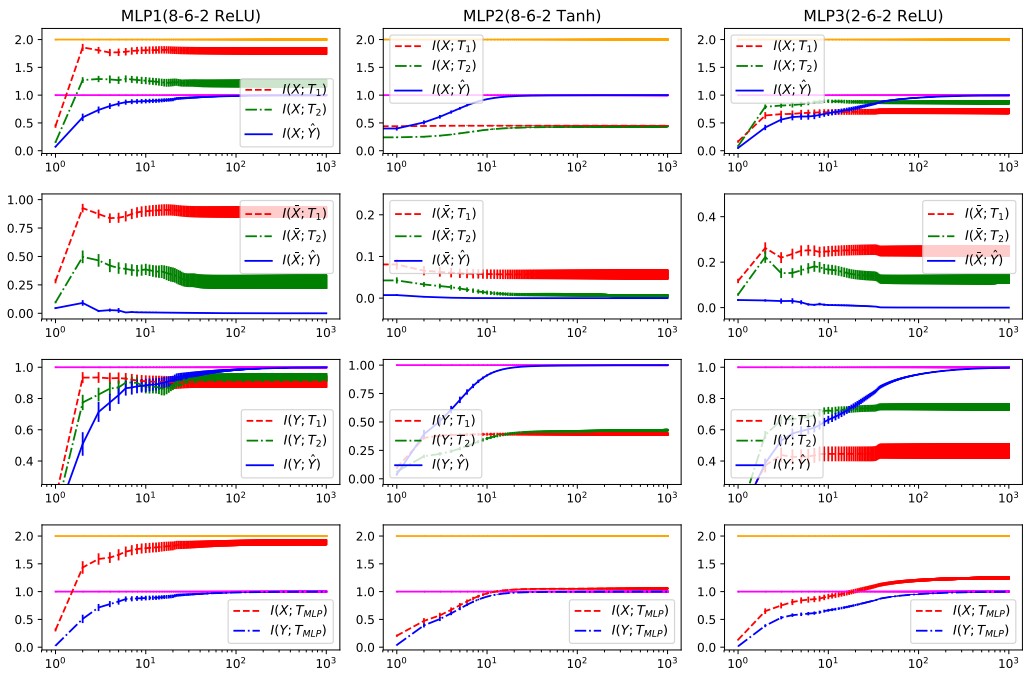

Figure 5: All the x-axis index training epochs. In each column, the first three figures show $I(X; T_i)$, $I(\bar{X}; T_i)$, and $I(Y; T_i)$ respectively. The forth figure shows $I(X; T_{\text{MLP}})$ and $I(Y; T_{\text{MLP}})$ in a MLP. The pink line denotes $H(Y) = 1.0$ and the orange line denotes $H(X) = 2.0$.

### 4.3 Validating the information theoretic explanations for DNNs

In Figure 5, we observe $I(X; T_i) \leq I(X; \hat{Y})$ in MLP2 and MLP3, which confirms that the Markov chain proposed by previous works, Equation (2), cannot fully explain the information flow in MLPs, if taking into account the back-propagation training. As a comparison, the second and third row show $I(\bar{X}; T_1) \geq I(\bar{X}; T_2) \geq I(\bar{X}; \hat{Y})$ and $I(Y; T_1) \leq I(Y; T_2) \geq I(Y; \hat{Y})$ in all the three MLPs, which validates that Corollary 1, *i.e.*, Equation (14) characterizes the information flow in MLPs.

Figure 5 demonstrates that different hidden layers achieve different IB trade-offs depending on the architecture and the position of the layers in MLPs. In terms of architecture, $I(Y; T_1) > 0.8$ and $I(\bar{X}; T_1) > 0.75$ in MLP1 indicate that $t_1$, with ReLU, achieves a good prediction without much compression, whereas $I(Y; T_1) < 0.5$ and $I(\bar{X}; T_1) < 0.1$ in MLP2 show that $t_1$, with Tanh, achieves a different IB trade-off. In addition, $I(Y; T_1) \approx 0.45$ and $I(\bar{X}; T_1) \approx 0.25$ in MLP3 show the effect of neuron numbers on the IB trade-off. In terms of position, $I(Y; \hat{Y}) = 1$ and $I(\bar{X}; \hat{Y}) = 0$ in MLP1 means that $\hat{y}$ has a different IB trade-off to $t_1$ in MLP1.

We demonstrate that a MLP satisfies the IB principle no matter what the architecture of the MLP is. Figure 5 visualizes $I(X; T_{\text{MLP}})$ and $I(Y; T_{\text{MLP}})$ based on Corollary 2. It shows that all of three MLPs satisfy the IB principle, namely $I(X; T_{\text{MLP}}) < H(X) = 2$ and $I(Y; T_{\text{MLP}}) = H(Y) = 1$, though they have different architectures. Importantly, in contrast to previous work [28] claiming that the compression not exists in DNNs with non-saturating ACT, such as ReLU, Figure 5 clearly shows that the compression exists in all the MLPs, no matter the activation function of MLPs.

We further demonstrate the information theoretic explanations for DNNs on the benchmark MNIST and Fashion-MNIST datasets. The experiments are presented in Appendix H.

## 5 Conclusion and future work

In this work, we (1) specify the probability space for a hidden layer for (2) accurately estimating the mutual information and (3) clearly explaining how the components of the layer affect the mutual information. We take into account the back-propagation training and derive two novel Markov chains to characterize the information flow in DNNs. Furthermore, we demonstrate that a DNN satisfies the IB principle no matter the architecture of the DNN. In contrast, different hidden layers show different IB trade-offs depending on the architecture and the position of the layers in DNNs. A potential direction is to study the generalization of DNNs based on the probabilistic representation.

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
