$\omega_1, \omega_2$ and $z$, respectively. $P_{T|Z}(\omega_1|z)$ and $P_{T|Z}(\omega_2|z)$ measure the probability of $\omega_1$ and $\omega_2$ being recognized as the feature with the largest cross-correlation to $z$, respectively.

## 2 A probabilistic representation for deep learning

To accurately estimate $I(X; T_i)$ and $I(Y; T_i)$, in this section, we specify the probability space [6] for a fully connected layer and derive the probabilistic explanations of the entire MLP.

It is known that a convolution kernel (namely the weights of convolution) defines a local feature, and a convolution operation derives a feature map to measure the cross-correlation between the local feature and input in a receptive field (Chapter 9.1 in [12]). Notably, a fully connected layer is equivalent to a convolution layer with the kernel size having the same dimension as input. Thus the weights of a neuron can be viewed as a global feature, and a fully connected layer with multiple neurons derives activations to measure the cross-correlation between the multiple global features and the input. The cross-correlation explanation for a fully connected layer is visualized in Figure 1.

Assuming that a fully connected layer $t$ consists of $N$ neurons $\{t_n = \sigma[\langle \omega_n, z \rangle]\}_{n=1}^N$, where $z \in \mathbb{R}^M$ is the input of $t$, $\langle \omega_n, z \rangle = \sum_{m=1}^M \omega_{mn} \cdot z_m + b_n$ is the dot-product between $z$ and $\omega_n$, and $\sigma(\cdot)$ is an activation function. Based on the cross-correlation explanation, the behavior of $t$ is to measure the cross-correlations between $z$ and the $N$ possible features defined by the the weights $\{\omega_n\}_{n=1}^N$. In the context of pattern recognition [39], we define a virtual random process or 'experiment' as $t$ recognizing one of the patterns/features with the largest cross-correlation to $z$ from the $N$ possible features. The experiment characterizes the behavior of $t$ (*i.e.*, before recognizing the features with the largest cross-correlation, $t$ must measure the cross-correlations between $z$ and all the $N$ possible features) while meets the requirement of the 'experiment' definition (*i.e.*, only one outcome will occur on each trial of the experiment [6]). The probability space $(\Omega_T, \mathcal{F}, P_T)$ is defined as follows:

**Definition 1.** $(\Omega_T, \mathcal{F}, P_T)$ consists of three components: the sample space $\Omega_T$ has $N$ possible outcomes (features) $\{\omega_n = \{\omega_{mn}\}_{m=1}^M\}_{n=1}^N$ defined by the weights[2] of the $N$ neurons; the event space $\mathcal{F}$ is the $\sigma$-algebra; and the probability measure $P_T$ is a Gibbs distribution [22] to quantify the probability of $\omega_n$ being recognized as the feature with the largest cross-correlation to $z$.

Taking into account the randomness of $z$, the conditional distribution $P_{T|Z}$ is formulated as

$$P_{T|Z}(\omega_n|z) = \frac{1}{Z_T}\exp(t_n) = \frac{1}{Z_T}\exp[\sigma(\langle \omega_n, z \rangle)], \qquad (5)$$

where $Z$ is the random variable of $z$ and $Z_T = \sum_{n=1}^N \exp(t_n)$ is the partition function.

$(\Omega_T, \mathcal{F}, P_T)$ clearly explains all the ingredients of $t$ in a probabilistic fashion. The $n$th neuron defines a global feature by the weights $w_n$ and the activation $t_n = \sigma(\langle \omega_n, z \rangle)$ measures the cross-correlation between $w_n$ and $z$. The Gibbs distribution $P_{T|Z}$ indicates that if $w_n$ has the higher activation, *i.e.*, the larger cross-correlation to $z$, it has the larger probability being recognized as the feature with largest cross-correlation to $z$. For instance, if $z \in \mathbb{R}^{16}$ and $t$ includes $N = 2$ neurons, then $\Omega_T = \{\omega_1, \omega_2\}$ defines two possible outcomes (features), where $\omega_n = \{\omega_{mn}\}_{m=1}^{16}$. $\mathcal{F} = \{\emptyset, \{\omega_1\}, \{\omega_2\}, \{\omega_1, \omega_2\}\}$ means that neither, one, or both of the features are recognized by $t$ given $z$, respectively. $P_{T|Z}(\omega_1|z)$ and $P_{T|Z}(\omega_2|z)$ are the probability of $\omega_1$ and $\

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

|  | $x$ | $t_1$ | $t_2$ | $\hat{y}$ | $\sigma(\cdot)$ |
|---|---|---|---|---|---|
| MLP1 | 1024 ($32 \times 32$) | 8 | 6 | 2 | $\mathrm{ReLU}(z) = \max(0, z)$ |
| MLP2 | 1024 ($32 \times 32$) | 8 | 6 | 2 | $\mathrm{Tanh}(z) = (e^z - e^{-z})/(e^z + e^{-z})$ |
| MLP3 | 1024 ($32 \times 32$) | 2 | 6 | 2 | ReLU |

To accurately estimate $I(X; T_i)$, we need to specify $P(T_i|X)$ and $P(T_i)$. Based on $(\Omega_{T_i}, \mathcal{F}, P_{T_i})$, we formulate $P_{T_i|X}(n|x^j)$ of the three fully connected layers in the MLP as

$$P_{T_1|X}(n|x^j) = \frac{1}{Z_{F_1}}\exp[\sigma_1(\langle \omega_n^{(1)}, x^j \rangle)], \quad P_{T_2|X}(k|x^j) = \frac{1}{Z_{F_2}}\exp[\sigma_2(\langle \omega_k^{(2)}, t_1(x^j) \rangle)],$$
$$P_{\hat{Y}|X}(l|x^j) = \frac{1}{Z_{F_Y}}\exp[\langle \omega_l^{(3)}, t_2(t_1(x^j)) \rangle]. \tag{17}$$

To derive the marginal distribution $P(T_i)$, we sum the joint distribution $P(T_i, X)$ over $x \in \mathcal{X}$,

$$P(T_i = n) = \sum_{x \in \mathcal{X}} P_X(x) P_{T_i|X}(n|x) \approx \sum_{x^j \in \mathcal{D}} P_X(x^j) P_{T_i|X}(n|x^j) = \frac{1}{J}\sum_{

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

As a fundamental probabilistic graphic model, the Gibbs distribution (a.k.a., Boltzmann distribution, the energy based model, or the renormalization group) formulates the dependence within $X$ by associating an energy $E(\boldsymbol{x}; \boldsymbol{\theta})$ to each dependence structure [9].

$$P(X; \boldsymbol{\theta}, \beta) = \frac{1}{Z(\boldsymbol{\theta}, \beta)} \exp[-\beta E(\boldsymbol{x}; \boldsymbol{\theta})], \tag{21}$$

where $E(\boldsymbol{x}; \boldsymbol{\theta})$ is the energy function, $\boldsymbol{\theta}$ denote the parameters of $E(\boldsymbol{x}; \boldsymbol{\theta})$, $\beta$ is the inverse temperature constant. Since $\beta$ can be absorbed into $\boldsymbol{\theta}$, $P(X; \boldsymbol{\theta}, \beta)$ can be simplified as

$$P(X; \boldsymbol{\theta}) = \frac{1}{Z(\boldsymbol{\theta})} \exp[-E(\boldsymbol{x}; \boldsymbol{\theta})], \tag{22}$$

where the partition function[5] is defined as

$$Z(\boldsymbol{\theta}) = \sum_{\boldsymbol{x} \in \mathcal{X}} \exp[-E(\boldsymbol{x}; \boldsymbol{\theta})]. \tag{23}$$

The Gibbs distribution has three appealing properties. First, the deterministic energy function $E(\boldsymbol{x}; \boldsymbol{\theta})$ is a sufficient statistics of $P(X; \boldsymbol{\theta})$. The property allows us to explain a deterministic function, e.g., a hidden layer, in a probabilistic way. Second, a Gibbs distribution can be easily reformulated as various probabilistic models via redefining $E(\boldsymbol{x}; \boldsymbol{\theta})$, which allows us to clarify the complicated architecture of a hidden layer. For example, if the energy function is defined as the summation of multiple functions, namely $E(\boldsymbol{x}; \boldsymbol{\theta}) = -\sum_k f_k(\boldsymbol{x}; \boldsymbol{\theta}_k)$, the Gibbs distribution would be the Product of Experts (PoE) model, *i.e.*, $P(\boldsymbol{x}; \boldsymbol{\theta}) = \frac{1}{Z(\boldsymbol{\theta})} \prod_k F_k$, where $F_k = \exp[-f_k(\boldsymbol{x}; \boldsymbol{\theta}_k)]$ and $Z(\boldsymbol{\theta}) = \prod_k Z(\boldsymbol{\theta}_k)$ [13]. Third, the energy minimization is a typical optimization for $\boldsymbol{\theta}$, namely $\boldsymbol{\theta}^* = \arg\min_{\boldsymbol{\theta}} E(\boldsymbol{x}; \boldsymbol{\theta})$ [22], which allows us to explain the back-propagation training, as the energy minimization can be implemented by the gradient descent algorithm as long as $E(\boldsymbol{x}; \boldsymbol{\theta})$ is differentiable.

A well-known Gibbs distribution model in machine learning is the Restricted Boltzmann Machines (RBMs) [31, 27]. Though Yaida indirectly proves the distribution of a fully connected layer as a Gibbs distribution [42], and Lin *et al.* clarify certain advantages of DNNs based on the Gibbs distribution [24], there is few work to extend the Gibbs explanation to complicated hidden layers, e.g., fully connected layers and convolutional layers.

## B  The marginal distribution of the MLP

Since the entire architecture of the MLP $= \{\boldsymbol{x}, \boldsymbol{t}_1, \boldsymbol{t}_2, \hat{\boldsymbol{y}}\}$ corresponds to a joint distribution

$$P(\hat{Y}, T_2, T_1 | X) = P(\hat{Y} | T_2) P(T_2 | T_1) P(T_1 | X), \tag{24}$$

the marginal distribution $P(\hat{Y} | X)$ can be formulated as

$$
\begin{aligned}
P_{\hat{Y}|X}(l|\boldsymbol{x}) &= \sum_{k=1}^{K} \sum_{n=1}^{N} P(\hat{Y} = l, T_2 = k, T_1 = n | X = \boldsymbol{x}) \\
&= \sum_{k=1}^{K} \sum_{n=1}^{N} P_{\hat{Y}|T_2}(l|k) P_{T_2|T_1}(k|n) P_{T_1|X}(n|\boldsymbol{x}).
\end{aligned}
\tag{25}
$$

Based on the definition of the Gibbs probability measure (Equation 5), we have

$$P_{T_1|X}(n|x) = \frac{1}{Z_{T_1}} \exp(t_{1n}) = \frac{1}{Z_{T_1}} \exp[\sigma_1(\langle \boldsymbol{\omega}_n^{(1)}, \boldsymbol{x} \rangle)], \tag{26}$$

where $\langle \boldsymbol{\omega}_n^{(1)}, \boldsymbol{x} \rangle = \sum_{m=1}^{M} \omega_{mn}^{(1)} \cdot x_m + b_{1n}$. Similarly, we have

$$P_{T_2|T_1}(k|n) = \frac{1}{Z_{T_2}} \exp(t_{2k}) = \frac{1}{Z_{T_2}} \exp[\sigma_2(\langle \boldsymbol{\omega}_k^{(2)}, \boldsymbol{t}_1 \rangle)], \tag{27}$$

where $\langle \boldsymbol{\omega}_k^{(2)}, \boldsymbol{t}_1 \rangle = \sum_{n=1}^{N} \omega_{nk}^{(2)} \cdot t_{1n} + b_{2k}$. Thus we have

$$
\begin{aligned}
& \sum_{n=1}^{N} P_{T_2|T_1}(k|n) P_{T_1|X}(n|x) \\
& = \frac{1}{Z_{T_2}} \frac{1}{Z_{T_1}} \sum_{n=1}^{N} \exp[\sigma_2(\langle \boldsymbol{\omega}_k^{(2)}, \boldsymbol{t}_1 \rangle)] \exp[\sigma_1(\langle \boldsymbol{\omega}_n^{(1)}, \boldsymbol{x} \rangle)].
\end{aligned}
\tag{28}
$$

---

[5]We only consider the discrete case in the paper.

463 Since $\langle \boldsymbol{\omega}_k^{(2)}, \boldsymbol{t}_1 \rangle = \sum_{n=1}^N \omega_{nk}^{(2)} \cdot t_{1n} + b_{2k}$ is a constant with respect to $n$, we have

$$
\sum_{n=1}^N P_{T_2|T_1}(k|n) P_{T_1|X}(n|x)
$$
$$
= \frac{1}{Z_{T_2}} \frac{1}{Z_{T_1}} \exp[\sigma_2(\langle \boldsymbol{\omega}_k^{(2)}, \boldsymbol{t}_1 \rangle)] \sum_{n=1}^N \exp[\sigma_1(\langle \boldsymbol{\omega}_n^{(1)}, \boldsymbol{x} \rangle)].
$$

(29)

464 In addition, $\sum_{n=1}^N \exp[\sigma_1(\langle \boldsymbol{\omega}_n^{(1)}, \boldsymbol{x} \rangle)] = Z_{T_1}$, thus we have

$$
\sum_{n=1}^N P_{T_2|T_1}(k|n) P_{T_1|X}(n|\boldsymbol{x}) = \frac{1}{Z_{T_2}} \exp[\sigma_2(\langle \boldsymbol{\omega}_k^{(2)}, \boldsymbol{t}_1 \rangle)].
$$

(30)

465 Therefore, we can simplify $P_{\hat{Y}|X}(l|\boldsymbol{x})$ as

$$
P_{\hat{Y}|X}(l|\boldsymbol{x}) = \sum_{k=1}^K P_{\hat{Y}|T_2}(l|k) \sum_{n=1}^N P_{T_2|T_1}(k|n) P_{T_1|X}(t|\boldsymbol{x})
$$
$$
= \sum_{k=1}^K P_{\hat{Y}|T_2}(l|k) \frac{1}{Z_{T_2}} \exp[\sigma_2(\langle \boldsymbol{\omega}_k^{(2)}, \boldsymbol{t}_1 \rangle)].
$$

(31)

466 Since $P_{\hat{Y}|T_2}(l|k) = \frac{1}{Z_{\hat{Y}}} \exp[\sigma_3(\langle \boldsymbol{\omega}_l^{(3)}, \boldsymbol{t}_2 \rangle)]$ and $\langle \boldsymbol{\omega}_l^{(3)}, \boldsymbol{t}_2 \rangle = \sum_{k=1}^K \omega_{lk}^{(3)} f_{2k} + b_{yl}$ is a constant with respect
467 to $k$, we can derive

$$
P_{\hat{Y}|X}(l|\boldsymbol{x}) = P_{\hat{Y}|T_2}(l|k) \sum_{k=1}^K \frac{1}{Z_{T_2}} \exp[\sigma_2(\langle \boldsymbol{\omega}_k^{(2)}, \boldsymbol{t}_1 \rangle)].
$$

(32)

468 Since $Z_{T_2} = \sum_{k=1}^K \exp[\sigma_2(\langle \boldsymbol{\omega}_k^{(2)}, \boldsymbol{t}_1 \rangle)]$ is also constant to $k$,

$$
P_{\hat{Y}|X}(l|\boldsymbol{x}) = P_{\hat{Y}|T_2}(l|k) \frac{1}{Z_{T_2}} \sum_{k=1}^K \exp[\sigma_2(\langle \boldsymbol{\omega}_k^{(2)}, \boldsymbol{t}_1 \rangle)].
$$
$$
= P_{\hat{Y}|T_2}(l|k) = \frac{1}{Z_{F_Y}} \exp[\langle \boldsymbol{\omega}_l^{(3)}, \boldsymbol{t}_2 \rangle].
$$

(33)

469 In addition, since $\boldsymbol{t}_2 = \{t_{2k}\}_{k=1}^K = \{\sigma_2(\langle \boldsymbol{\omega}_k^{(2)}, \boldsymbol{t}_1 \rangle)\}_{k=1}^K$, we can extend $P_{\hat{Y}|X}(l|\boldsymbol{x})$ as

$$
P_{\hat{Y}|X}(l|\boldsymbol{x}) = P_{\hat{Y}|F_2}(l|k) = \frac{1}{Z_{F_Y}} \exp[\langle \boldsymbol{\omega}_l^{(3)}, \boldsymbol{t}_2 \rangle]
$$
$$
= \frac{1}{Z_{\hat{Y}}} \exp[\langle \boldsymbol{\omega}_l^{(3)}, \begin{pmatrix} \sigma_2(\langle \boldsymbol{\omega}_1^{(2)}, \boldsymbol{t}_1 \rangle) \\ \vdots \\ \sigma_2(\langle \boldsymbol{\omega}_K^{(2)}, \boldsymbol{t}_1 \rangle) \end{pmatrix} \rangle].
$$

(34)

470 Since $\boldsymbol{t}_1 = \{t_{1n}\}_{n=1}^N = \{\sigma_1(\langle \boldsymbol{\omega}_n^{(1)}, \boldsymbol{x} \rangle)\}_{n=1}^N$, we can further extend $P_{\hat{Y}|X}(l|\boldsymbol{x})$ as

$$
P_{\hat{Y}|X}(l|\boldsymbol{x}) = \frac{1}{Z_{\hat{Y}}} \exp[\langle \boldsymbol{\omega}_l^{(3)}, \begin{pmatrix} \sigma_2(\langle \boldsymbol{\omega}_1^{(2)}, \begin{pmatrix} \sigma_1(\langle \boldsymbol{\omega}_1^{(1)}, \boldsymbol{x} \rangle) \\ \vdots \\ \sigma_1(\langle \boldsymbol{\omega}_N^{(1)}, \boldsymbol{x} \rangle) \end{pmatrix} \rangle) \\ \vdots \\ \sigma_2(\langle \boldsymbol{\omega}_K^{(2)}, \begin{pmatrix} \sigma_1(\langle \boldsymbol{\omega}_1^{(1)}, \boldsymbol{x} \rangle) \\ \vdots \\ \sigma_1(\langle \boldsymbol{\omega}_N^{(1)}, \boldsymbol{x} \rangle) \end{pmatrix} \rangle) \end{pmatrix} \rangle]
$$
$$
= \frac{1}{Z_{\mathrm{MLP}}(\boldsymbol{x})} \exp[g_l(\boldsymbol{t}_2(\boldsymbol{t}_1(\boldsymbol{x})))].
$$

(35)

471 Overall, we prove $P_{\hat{Y}|X}(l|\boldsymbol{x})$ as the Gibbs distribution expressed as

$$
P_{\hat{Y}|X}(l|\boldsymbol{x}) = \frac{1}{Z_{\mathrm{MLP}}(\boldsymbol{x})} \exp[g_l(\boldsymbol{t}_2(\boldsymbol{t}_1(\boldsymbol{x})))].
$$

(36)

where $E_l(x) = -g_l(\boldsymbol{t}_2(\boldsymbol{t}_1(\boldsymbol{x})))$ is the energy function and the partition function

$$Z_{\mathrm{MLP}}(\boldsymbol{x}) = \sum_{l=1}^{L} \sum_{k=1}^{K} \sum_{t=1}^{T} P(\hat{Y}, T_2, T_1 | X = \boldsymbol{x})$$

$$= \sum_{l=1}^{L} \exp[g_l(\boldsymbol{t}_2(\boldsymbol{t}_1(\boldsymbol{x})))]. \tag{37}$$

## C The proof of Theorem 2

Based on the definition of the cross entropy, $\ell_{\mathrm{CE}}$ can be formulated as

$$\ell_{\mathrm{CE}} = -\sum_{l=1}^{L} P_{Y|X}(l|\boldsymbol{x}) \log P_{\hat{Y}|X}(l|\boldsymbol{x}). \tag{38}$$

where $P_{\hat{Y}|X}(l|\boldsymbol{x})$ is the output of the MLP, and $P_{Y|X}(l|\boldsymbol{x})$ is the one-hot probability of $\boldsymbol{x}$ given the label $y$, *i.e.*,

$$P_{Y|X}(l|\boldsymbol{x}) = \begin{cases} 1 & \text{for} \quad l = y \\ 0 & \text{for} \quad l \neq y \end{cases} \tag{39}$$

The derivative of $\ell_{\mathrm{CE}}$ with respect to $P_{\hat{Y}|X}(l|\boldsymbol{x})$ is

$$\frac{\partial \ell_{\mathrm{CE}}}{\partial P_{\hat{Y}|X}(l|\boldsymbol{x})} = -\frac{P_{Y|X}(l|\boldsymbol{x})}{P_{\hat{Y}|X}(l|\boldsymbol{x})}. \tag{40}$$

Since $P_{\hat{Y}|X}(l|\boldsymbol{x})$ can be expressed as

$$P_{\hat{Y}|X}(l|\boldsymbol{x}) = \frac{1}{Z_{\mathrm{MLP}}(\boldsymbol{x})} \exp[g_l(\boldsymbol{t}_2 \boldsymbol{t}_1(\boldsymbol{x}))], \tag{41}$$

the derivative of $P_{\hat{Y}|X}(z|\boldsymbol{x})$ with respect to $g_l(\boldsymbol{t}_2 \boldsymbol{t}_1(\boldsymbol{x}))$ is

$$\frac{\partial P_{\hat{Y}|X}(z|\boldsymbol{x})}{\partial g_l} = \frac{\frac{1}{Z_{\mathrm{MLP}}} \exp(g_z)}{\partial g_l} = \begin{cases} P_{\hat{Y}|X}(l|\boldsymbol{x}) \cdot [1 - P_{\hat{Y}|X}(l|\boldsymbol{x})] & \text{for} \quad z = l \\ -P_{\hat{Y}|X}(l|\boldsymbol{x}) \cdot P_{\hat{Y}|X}(z|\boldsymbol{x}) & \text{for} \quad z \neq l \end{cases}. \tag{42}$$

Overall, the derivative of $\ell_{\mathrm{CE}}$ with respect to $g_l$ can be expressed as

$$\frac{\partial \ell_{\mathrm{CE}}}{\partial g_l} = \sum_{z=1}^{L} \frac{\partial \ell_{\mathrm{CE}}}{\partial P_{\hat{Y}|X}(z|\boldsymbol{x})} \frac{\partial P_{\hat{Y}|X}(z|\boldsymbol{x})}{\partial g_l}$$

$$= -P_{Y|X}(l|\boldsymbol{x})(1 - P_{\hat{Y}|X}(l|\boldsymbol{x}) + \sum_{z \neq l} P_{Y|X}(z|\boldsymbol{x}) P_{\hat{Y}|X}(l|\boldsymbol{x}) \tag{43}$$

$$= P_{\hat{Y}|X}(l|\boldsymbol{x}) - P_{Y|X}(l|\boldsymbol{x}).$$

Since $g_l = \langle \boldsymbol{\omega}_l^{(3)}, \boldsymbol{t}_2 \rangle = \sum_{k=1}^{K} \omega_{kl}^{(3)} \cdot t_{2k} + b_{yl}$, the derivative of $\ell_{\mathrm{CE}}$ with respect to $\omega_{kl}^{(3)}$ can be expressed as

$$\frac{\partial \ell_{\mathrm{CE}}}{\partial \omega_{kl}^{(3)}} = \frac{\partial \ell_{\mathrm{CE}}}{\partial g_l} \frac{\partial g_l}{\partial \omega_{kl}^{(3)}} = [P_{\hat{Y}|X}(l|\boldsymbol{x}) - P_{Y|X}(l|\boldsymbol{x})] t_{2k}. \tag{44}$$

Similarly, the derivative of $\ell_{\mathrm{CE}}$ with respect to $\langle \boldsymbol{\omega}_k^{(2)}, \boldsymbol{t}_1 \rangle$ can be expressed as

$$\frac{\partial \ell_{\mathrm{CE}}}{\partial \langle \boldsymbol{\omega}_k^{(2)}, \boldsymbol{t}_1 \rangle} = \sum_{l=1}^{L} \frac{\partial \ell_{\mathrm{CE}}}{\partial g_l} \frac{\partial g_l}{\partial t_{2k}} \frac{\partial t_{2k}}{\partial \langle \boldsymbol{\omega}_k^{(2)}, \boldsymbol{t}_1 \rangle}$$

$$= \sum_{l=1}^{L} [P_{\hat{Y}|X}(l|\boldsymbol{x}) - P_{Y|X}(l|\boldsymbol{x})] \omega_{kl}^{(3)} \sigma_2'(\langle \boldsymbol{\omega}_k^{(2)}, \boldsymbol{t}_1 \rangle). \tag{45}$$

Since $\langle \boldsymbol{\omega}_k^{(2)}, \boldsymbol{t}_1 \rangle = \sum_{n=1}^{N} \omega_{nk}^{(2)} \cdot t_{1n} + b_{2k}$, the derivative of $\ell$ with respect to $\omega_{nk}^{(2)}$ can be expressed as

$$\frac{\partial \ell_{\mathrm{CE}}}{\partial \omega_{nk}^{(2)}} = \frac{\partial \ell_{\mathrm{CE}}}{\partial \langle \boldsymbol{\omega}_k^{(2)}, \boldsymbol{t}_1 \rangle} \frac{\partial \langle \boldsymbol{\omega}_k^{(2)}, \boldsymbol{t}_1 \rangle}{\partial \omega_{nk}^{(2)}}$$

$$= \sum_{l=1}^{L} [P_{\hat{Y}|X}(l|\boldsymbol{x}) - P_{Y|X}(l|\boldsymbol{x})] \omega_{kl}^{(3)} \sigma_2'(\langle \boldsymbol{\omega}_k^{(2)}, \boldsymbol{t}_1 \rangle) t_{1n} \tag{46}$$

483 Similarly, the derivative of $\ell_{\text{CE}}$ with respect to $\langle \boldsymbol{\omega}_n^{(1)}, \boldsymbol{x} \rangle$ can be expressed as

$$\frac{\partial \ell_{\text{CE}}}{\partial \langle \boldsymbol{\omega}_n^{(1)}, \boldsymbol{x} \rangle} = \sum_{k=1}^{K} \frac{\partial \ell_{\text{CE}}}{\partial \langle \boldsymbol{\omega}_k^{(2)}, \boldsymbol{t}_1 \rangle} \frac{\partial \langle \boldsymbol{\omega}_k^{(2)}, \boldsymbol{t}_1 \rangle}{\partial t_{1n}} \frac{\partial t_{1n}}{\partial \langle \boldsymbol{\omega}_n^{(1)}, \boldsymbol{x} \rangle}$$
$$= \sum_{k=1}^{K} \sum_{l=1}^{L} [P_{\hat{Y}|X}(l|\boldsymbol{x}) - P_{Y|X}(l|\boldsymbol{x})] \omega_{kl}^{(3)} \sigma_2'(\langle \boldsymbol{\omega}_k^{(2)}, \boldsymbol{t}_1 \rangle) \omega_{nk}^{(2)} \sigma_1'(\langle \boldsymbol{\omega}_n^{(1)}, \boldsymbol{x} \rangle). \tag{47}$$

484 Since $\langle \boldsymbol{\omega}_n^{(1)}, \boldsymbol{x} \rangle = \sum_{m=1}^{M} \omega_{mn}^{(1)} \cdot x_m + b_{1n}$, the derivative of $\ell_{\text{CE}}$ with respect to $\omega_{mn}^{(1)}$ can be expressed as

$$\frac{\partial \ell_{\text{CE}}}{\partial \omega_{mn}^{(1)}} = \frac{\partial \ell_{\text{CE}}}{\partial \langle \boldsymbol{\omega}_n^{(1)}, \boldsymbol{x} \rangle} \frac{\partial \langle \boldsymbol{\omega}_n^{(1)}, \boldsymbol{x} \rangle}{\partial \omega_{mn}^{(1)}}$$
$$= \sum_{k=1}^{K} \sum_{l=1}^{L} [P_{\hat{Y}|X}(l|\boldsymbol{x}) - P_{Y|X}(l|\boldsymbol{x})] \omega_{kl}^{(3)} \sigma_2'(\langle \boldsymbol{\omega}_k^{(2)}, \boldsymbol{t}_1 \rangle) \omega_{nk}^{(2)} \sigma_1'(\langle \boldsymbol{\omega}_n^{(1)}, \boldsymbol{x} \rangle) x_m. \tag{48}$$

485 Overall, the derivative of $\ell_{\text{CE}}$ with respect to the weight in each layer is summarized as

$$\frac{\partial \ell_{\text{CE}}}{\partial \omega_{kl}^{(3)}} = [P_{\hat{Y}|X}(l|\boldsymbol{x}) - P_{Y|X}(l|\boldsymbol{x})] t_{2k}$$
$$\frac{\partial \ell_{\text{CE}}}{\partial \omega_{nk}^{(2)}} = \sum_{l=1}^{L} [P_{\hat{Y}|X}(l|\boldsymbol{x}) - P_{Y|X}(l|\boldsymbol{x})] \omega_{kl}^{(3)} \sigma_2'(\langle \boldsymbol{\omega}_k^{(2)}, \boldsymbol{t}_1 \rangle) t_{1n} \tag{49}$$
$$\frac{\partial \ell_{\text{CE}}}{\partial \omega_{mn}^{(1)}} = \sum_{k=1}^{K} \sum_{l=1}^{L} [P_{\hat{Y}|X}(l|\boldsymbol{x}) - P_{Y|X}(l|\boldsymbol{x})] \omega_{kl}^{(3)} \sigma_2'(\langle \boldsymbol{\omega}_k^{(2)}, \boldsymbol{t}_1 \rangle) \omega_{nk}^{(2)} \sigma_1'(\langle \boldsymbol{\omega}_n^{(1)}, \boldsymbol{x} \rangle) x_m.$$

486 Based on the above three equations, we can reformulate the derivatives as

$$\frac{\partial \ell_{\text{CE}}^{\star}}{\partial \omega_{kl}^{(3)}} = [P_{\hat{Y}|X}(l|\boldsymbol{x}) - P_{Y|X}(l|\boldsymbol{x})] \cdot t_{2k},$$
$$\frac{\partial \ell_{\text{CE}}^{\odot}}{\partial \omega_{nk}^{(2)}} = \sum_{l=1}^{L} \frac{\partial \ell_{\text{CE}}^{\star}}{\partial \omega_{kl}^{(3)}} \cdot \omega_{kl}^{(3)} \cdot \frac{\sigma_2'(\langle \boldsymbol{\omega}_k^{(2)}, \boldsymbol{t}_1 \rangle)}{t_{2k}} \cdot t_{1n} \tag{50}$$
$$\frac{\partial \ell_{\text{CE}}^{\diamond}}{\partial \omega_{mn}^{(1)}} = \sum_{k=1}^{K} \frac{\partial \ell_{\text{CE}}^{\odot}}{\partial \omega_{nk}^{(2)}} \cdot \omega_{nk}^{(2)} \cdot \frac{\sigma_1'(\langle \boldsymbol{\omega}_n^{(1)}, \boldsymbol{x} \rangle)}{t_{1n}} \cdot x_m.$$

487 The above three equations indicates that $\frac{\partial \ell_{\text{CE}}^{\star}}{\partial \omega_{kl}^{(3)}}$ is a function of $P_{Y|X}(l|\boldsymbol{x})$, $\frac{\partial \ell_{\text{CE}}^{\odot}}{\partial \omega_{nk}^{(2)}}$ is a function of $\frac{\partial \ell_{\text{CE}}^{\star}}{\partial \omega_{kl}^{(3)}}$, and

488 $\frac{\partial \ell_{\text{CE}}^{\diamond}}{\partial \omega_{mn}^{(1)}}$ is a function of $\frac{\partial \ell_{\text{CE}}^{\odot}}{\partial \omega_{nk}^{(2)}}$. In addition, the back-propagation algorithm shows that

$$\omega_{mn}^{(1)}(s+1) = \omega_{mn}^{(1)}(s) - \alpha \frac{\partial \ell_{\text{CE}}}{\partial \omega_{mn}^{(1)}(s)}$$
$$\omega_{nk}^{(2)}(s+1) = \omega_{nk}^{(2)}(s) - \alpha \frac{\partial \ell_{\text{CE}}}{\partial \omega_{nk}^{(2)}(s)} \tag{51}$$
$$\omega_{kl}^{(3)}(s+1) = \omega_{kl}^{(3)}(s) - \alpha \frac{\partial \ell_{\text{CE}}}{\partial \omega_{kl}^{(3)}(s)}$$

489 where $\alpha$ is the learning rate and $s$ denotes the index of the $s$th learning iteration. Therefore, $\omega(s+1)$ is
490 determined by all the previous gradients $\{\frac{\partial \ell_{\text{CE}}}{\partial \omega(s)}\}_{s=1}^{S}$ as $\omega(0)$ is randomly initialized and $\alpha$ is a constant.

491 Definition 1 indicates that the weights define the sample space $\Omega_{T_i}$, thus we can derive that the gradients $\frac{\partial \ell_{\text{CE}}}{\partial \omega^{(i)}}$
492 determine $\Omega_{T_i}$. As a result, $\Omega_{T_i}$ is a function of $\Omega_{T_{i+1}}$ and $\Omega_{\hat{Y}}$ is a function of $P(Y|X)$. Based on Definition
493 2, we can further derive that $T_i$ is a function of $T_{i+1}$ and $\hat{Y}$ is a function of $Y$, $i.e.$, $T_1 \leftarrow T_2 \leftarrow \hat{Y} \leftarrow Y$.

## D  The proof of $H(Y) = I(X;Y)$

495 Given a training sample $\boldsymbol{x}^j$ and the corresponding label $y^j$, the target distribution $P_{Y|X}(y^j|\boldsymbol{x}^j)$ is commonly
496 formulated as the one-hot format, $i.e.$,

$$P_{Y|X}(l|\boldsymbol{x}^j) = \begin{cases} 1 & \text{for} \quad l = y^j \\ 0 & \text{for} \quad l \neq y^j \end{cases} \tag{52}$$

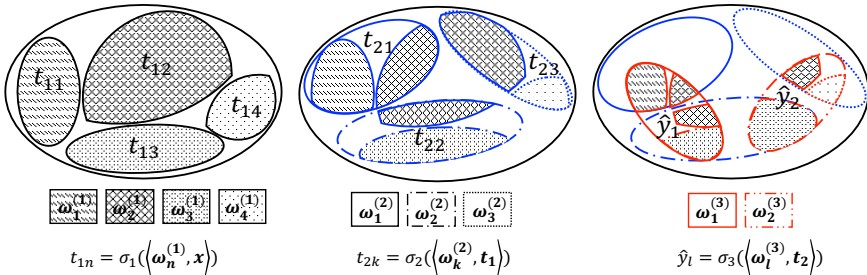

$$t_{1n} = \sigma_1(\langle \boldsymbol{\omega}_n^{(1)}, \boldsymbol{x} \rangle) \qquad t_{2k} = \sigma_2(\langle \boldsymbol{\omega}_k^{(2)}, \boldsymbol{t_1} \rangle) \qquad \hat{y}_l = \sigma_3(\langle \boldsymbol{\omega}_l^{(3)}, \boldsymbol{t_2} \rangle)$$

Figure 6: The graphical explanation for Corollary 1 based on the MLP $= \{\boldsymbol{x}, \boldsymbol{t}_1, \boldsymbol{t}_2, \hat{\boldsymbol{y}}\}$. The largest oval represents of the input $\boldsymbol{x}$, and each small shape indicates the representation capacity of a single feature. For example, if $t_{12}$ is the largest activation, then the feature $\boldsymbol{\omega}_2^{(1)}$ has the largest cross-correlation to $\boldsymbol{x}$, *i.e.*, $\boldsymbol{\omega}_2^{(1)}$ has largest representation capacity. Therefore, $\{\boldsymbol{\omega}_n^{(1)}\}_{n=1}^4$ can be viewed as a representation of $\boldsymbol{x}$, and the representation capacity of $\{\boldsymbol{\omega}_n^{(1)}\}_{n=1}^4$ is measured by $\{t_{1n}\}_{n=1}^N$, which is visualized by the left figure. The blue ovals indicates the representation capacity of the three features $\{\boldsymbol{\omega}_k^{(2)}\}_{k=1}^3$ generated by combining the four features $\{\boldsymbol{\omega}_n^{(1)}\}_{n=1}^4$. The two red ovals indicates the representation capacity of the two features $\{\boldsymbol{\omega}_l^{(3)}\}_{l=1}^2$ generated by combining the three features $\{\boldsymbol{\omega}_k^{(2)}\}_{k=1}^3$.

As a result, the conditional entropy $H(Y|X)$ can be formulated as

$$H(Y|X) = -\sum_{(\boldsymbol{x}^j, y^j) \in \mathcal{D}} P_{X,Y}(\boldsymbol{x}^j, y^j) \log P_{Y|X}(y^j | \boldsymbol{x}^j) = 0. \tag{53}$$

Therefore, we can derive $H(Y) = I(X;Y)$ because $H(Y) = H(Y|X) + I(X;Y)$.

# E    The detailed derivations and explanations for Corollary 1

Definition 1 indicates that $\boldsymbol{t}_1 = \{t_{1n} = \sigma_1(\langle \boldsymbol{\omega}_n^{(1)}, \boldsymbol{x} \rangle)\}_{n=1}^N$ defines $N$ features of $\boldsymbol{x}$, namely $\{\boldsymbol{\omega}_n^{(1)}\}_{n=1}^N$, thus $\{\boldsymbol{\omega}_n^{(1)}\}_{n=1}^N$ can be viewed as a representation of $\boldsymbol{x}$. In addition, $\{t_{1n}\}_{n=1}^N$ measures the cross-correlation between $\{\boldsymbol{\omega}_n^{(1)}\}_{n=1}^N$ and $\boldsymbol{x}$, (*i.e.*, if $\boldsymbol{\omega}_n^{(1)}$ describes $\boldsymbol{x}$ more accurately and comprehensively, then $t_{1n}$ is larger.), thus $\{t_{1n}\}_{n=1}^N$ quantifies the representation capacity of $\{\boldsymbol{\omega}_n^{(1)}\}_{n=1}^N$. For example, in Figure 6 (Left), $\boldsymbol{t}_1$ defines 4 features to describe $\boldsymbol{x}$ and $t_{12}$ is the largest activation, thus $\boldsymbol{\omega}_2^{(1)}$ has the largest representation capacity of $\boldsymbol{x}$.

During inference, the second hidden layer $\boldsymbol{t}_2$ will process $\{t_{1n}\}_{n=1}^N$, and $t_{2k} = \sigma_2(\langle \boldsymbol{w}_k^{(2)}, \boldsymbol{t}_1 \rangle)$ can be explained to generating a new feature via combining all the features $\{\boldsymbol{\omega}_n^{(1)}\}_{n=1}^N$, *i.e.*,

$$\{\omega_{1k}^{(2)} \otimes \boldsymbol{\omega}_1^{(1)}, \cdots, \omega_{Nk}^{(2)} \otimes \boldsymbol{\omega}_N^{(1)}\}. \tag{54}$$

Since the new feature is the linear combination of $\{\boldsymbol{\omega}_n^{(1)}\}_{n=1}^N$, it can be simply noted as

$$\{\omega_{1k}^{(2)}, \cdots, \omega_{Nk}^{(2)}\} = \boldsymbol{\omega}_k^{(2)}, \tag{55}$$

and the representation capacity of the new feature $\boldsymbol{\omega}_k^{(2)}$ is

$$t_{2k} = \omega_{1k}^{(2)} \cdot t_{11} + \cdots + \omega_{Nk}^{(2)} \cdot t_{1N} \tag{56}$$

For example, if $N = 4$ and $K = 3$, the representation capacity of the three new features is visualized by Figure 6 (Middle). Similarly, $\hat{\boldsymbol{y}}$ generates $L$ new features via combining all the features $\{\boldsymbol{\omega}_k^{(2)}\}_{k=1}^K$.

$$\{\omega_{1l}^{(3)} \otimes \boldsymbol{\omega}_1^{(2)}, \cdots, \omega_{Kl}^{(3)} \otimes \boldsymbol{\omega}_K^{(2)}\}. \tag{57}$$

Since the new feature is the linear combination of $\{\boldsymbol{\omega}_k^{(2)}\}_{k=1}^K$, it can be simply noted as

$$\{\omega_{1l}^{(3)}, \cdots, \omega_{Kl}^{(3)}\} = \boldsymbol{\omega}_l^{(3)}, \tag{58}$$

and the representation capacity of the new feature $\boldsymbol{\omega}_l^{(3)}$ is

$$\hat{y}_l = \omega_{1l}^{(3)} \cdot t_{21} + \cdots + \omega_{Kl}^{(3)} \cdot t_{2K} \tag{59}$$

For example, if $L = 2$, the representation capacity of the two new features is visualized by Figure 6 (Right).

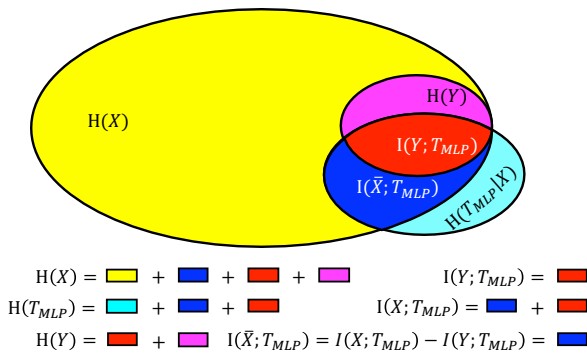

Figure 7: The Venn diagram of $H(X)$, $H(Y)$, and $I(X; T_{\mathrm{MLP}})$.

514   Overall, the inference phase is a procedure of feature combination, *i.e.*, $\boldsymbol{\omega}_l^{(3)}$ is a combination of $\{\boldsymbol{\omega}_k^{(2)}\}_{k=1}^K$,
515   and $\boldsymbol{\omega}_k^{(2)}$ is a combination of $\{\boldsymbol{\omega}_n^{(1)}\}_{n=1}^N$. Theorem 2 proves that the layer closer to output has more information
516   of labels, *i.e.*, $T_1 \leftarrow T_2 \leftarrow \hat{Y} \leftarrow Y$, during training. Since the weights are fixed after training, the sample space
517   and the distribution of hidden layers are fixed after training. Therefore, the information of $Y$ transferred into
518   hidden layers during training will retain there after training (*i.e.*, during inference), i.e., $T_1 \leftarrow T_2 \leftarrow \hat{Y} \leftarrow Y$
519   characterizes the information flow of $Y$ in the MLP in both training and inference phases.

520   For example, Figure 6 (Right) shows that the representation capacity of $\boldsymbol{\omega}_1^{(3)}$ is the weighted combination of $t_{11}$,
521   $t_{12}$, and $t_{13}$, and the representation capacity of $\boldsymbol{\omega}_2^{(3)}$ is the weighted combination of $t_{12}$, $t_{13}$, and $t_{14}$. Therefore,
522   $\boldsymbol{\omega}_2^{(1)}$ and $\boldsymbol{\omega}_3^{(1)}$ exist in both classes, *i.e.*, the low-level features in $\boldsymbol{t}_1$ do not represent too much information of
523   the labels, though we combine low-level features to generate high-level features for representing labels.

## F   The proof of Corollary 2

525   Based on the property of mutual information, we have

$$
\begin{aligned}
H(X) &= H(X|Y) + I(X;Y) \\
&= H(X|Y) + H(Y) \quad \text{(Appendix D)} \\
&= H(\bar{X}) + H(Y)
\end{aligned}
\tag{60}
$$

526   where $\bar{X}$ is the virtual random variable containing all the information of $X$ except $Y$, namely $H(\bar{X}) = H(X|Y)$.

527   Therefore, $I(X; T_{\mathrm{MLP}})$ can be reformulated as

$$
I(X; T_{\mathrm{MLP}}) = I(\bar{X}; T_{\mathrm{MLP}}) + I(Y; T_{\mathrm{MLP}}).
\tag{61}
$$

528   The Venn diagram of $H(X)$, $H(Y)$, and $I(X; T_{\mathrm{MLP}})$ are visualized in Figure 7. Corollary 1 indicates that all
529   the information of $\bar{X}$ and $Y$ learned by a MLP retains in $T_1$ and $\hat{Y}$, respectively. Therefore, we can derive

$$
\begin{aligned}
I(X; T_{\mathrm{MLP}}) &= I(\bar{X}; T_1) + I(Y; \hat{Y}) \\
I(Y; T_{\mathrm{MLP}}) &= I(Y; \hat{Y})
\end{aligned}
\tag{62}
$$

## G   Studying non-parametric models for mutual information estimation

531   In this section, we use the synthetic dataset to show that non-parametric models are sensitive to hyper-parameters
532   for mutual information estimation. In addition, we show that the proposed mutual information estimator derives
533   more accurate mutual information estimation than non-parametric models. Furthermore, we demonstrate that
534   one reason for non-parametric models deriving poor mutual information estimation is because activations do not
535   satisfy the *i.i.d.* prerequisite of non-parametric models. The experiment codes are available online[6].

### G.1   Non-parametric models are sensitive to hyper-parameters

537   To show non-parametric models being sensitive to hyper-parameters, we choose two commonly used non-
538   parametric models, namely the empirical distribution [35] and KDE [33], to measure the information flow in
539   MLP1 and MLP2 defined in Table 1 on the synthetic dataset.

---

[6] https://github.com/Dlib-NeurIPS/Deep-Learning-Information-Theory

Table 3: The hyper-parameters of empirical distributions and KDE

| $bs$ | 0.001 | 0.01 | 0.1 | 1.0 | 2.0 | 4.0 | 6.0 | 8.0 |
|---|---|---|---|---|---|---|---|---|
| $\sigma_n^2$ | 0.01 | 0.05 | 0.1 | 1.0 | 2.0 | 4.0 | 8.0 | 16.0 |

540 The empirical distribution is defined as

$$P(T = n) = \frac{1}{J}\mathbb{1}(\boldsymbol{t}, \boldsymbol{l}_n, \boldsymbol{r}_n) \tag{63}$$

541 where $J$ is the number of samples, $n$ denotes the $n$th bin, $\boldsymbol{t}$ denotes an activation vector, $\boldsymbol{l}_n$ and $\boldsymbol{r}_n$ are the left
542 and right boundary vectors, respectively. The indicator function $\mathbb{1}(\boldsymbol{t}, \boldsymbol{l}_n, \boldsymbol{r}_n)$ is defined as

$$\mathbb{1}(\boldsymbol{t}, \boldsymbol{l}_n, \boldsymbol{r}_n) = \left\{ \begin{array}{ll} 1 & \text{for} \qquad \boldsymbol{l}_n \leq \boldsymbol{t} < \boldsymbol{r}_n \\ 0 & \text{otherwise} \end{array} \right. \tag{64}$$

543 Given a specific range, the hyper-parameter of the empirical distribution is the bin size, namely $bs = |\boldsymbol{r}_n - \boldsymbol{l}_n|$.
544 Based on the empirical distribution, Tishby *et al.* estimate $I(X; T_i)$ and $I(Y; T_i)$ (see Section 3.2 in [35]).

545 To estimate $I(X; T_i)$ and $I(Y; T_i)$ via KDE, Saxe *et al.* assume that the empirical distribution of input samples
546 is the true distribution and the distribution of a hidden layer is a mixture of Gaussian. In addition, Saxe *et al.*
547 regard a hidden layer as a deterministic function of input samples, thus the Gaussian noise $\mathcal{N}(0, \sigma_n^2)$ is added
548 into activations to avoid infinite mutual information, and $I(X; T_i)$ is estimated as

$$I(X; T_i) \leq -\frac{1}{J}\sum_j \log \frac{1}{J}\sum_{j'} \exp(-\frac{\|\boldsymbol{t}_j^{(i)} - \boldsymbol{t}_{j'}^{(i)}\|_2^2}{2\sigma_n^2}) \tag{65}$$

549 where $J$ is the number of samples, $\boldsymbol{t}_j^{(i)}$ denote the activations vector of the $i$th hidden layer in response to the
550 input sample $\boldsymbol{x}^j$ (see Appendix B.1 in [33]). Therefore, the hyper-parameter of KDE is the noise variance $\sigma_n^2$.

551 Leveraging the same training method in Section 4.1, we achieves 100% training accuracy in MLP1 and MLP2
552 on the synthetic dataset. We specify 8 different values for each hyper-parameter, namely $bs$ and $\sigma_n^2$, in Table 3,
553 and use the empirical distribution and KDE to estimate $I(X; T_i)$ during training MLP1 and MLP2.

554 Figure 8 and 9 show that the empirical distribution is sensitive to the hyper-parameter, namely the bin size $bs$.
555 Figure 8 shows that $I(X; T_i)$ in different hidden layers of MLP1 converges to 1.5 as $bs$ increases from 0.001 to
556 8.0. Figure 9 shows that $I(X; T_1)$, $I(X; T_2)$, and $I(X; \hat{Y})$ in MLP2 converge to 1.2, 0.8, and 0.7, respectively,
557 as $bs$ increases from 0.001 to 8.0. Notably, since the synthetic dataset only has 2 bits information, $I(X; T_i)$
558 must be smaller than $H(X) = 2$ bits. However, we observe that if $bs < 1.0$, the empirical distribution derives
559 $I(X; T_i) > 2.0$ in both MLP1 and MLP2, thus the empirical distribution cannot correctly estimate $I(X; T_i)$ in
560 MLP1 and MLP2 on the synthetic dataset when $bs < 1.0$.

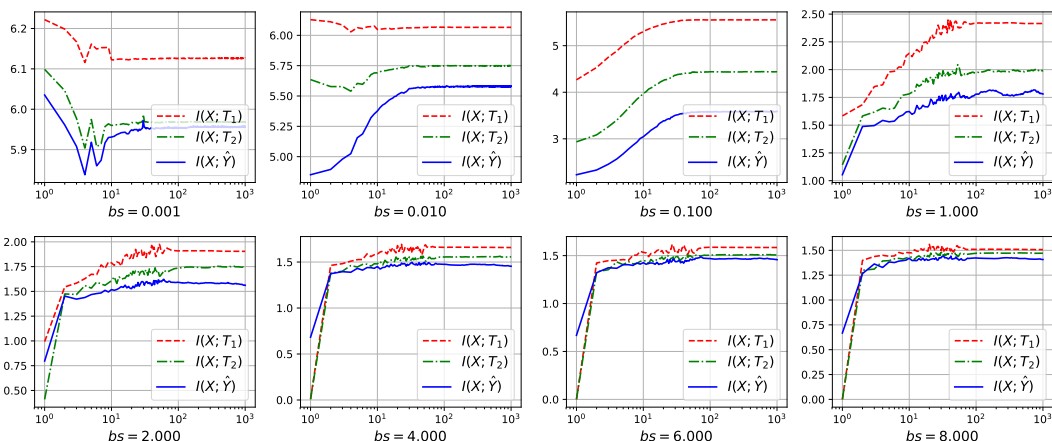

Figure 8: The estimation of $I(X; T_i)$ in MLP1 on the synthetic dataset via the empirical distribution with 8
different $bs$. All the x-axis index training epochs.

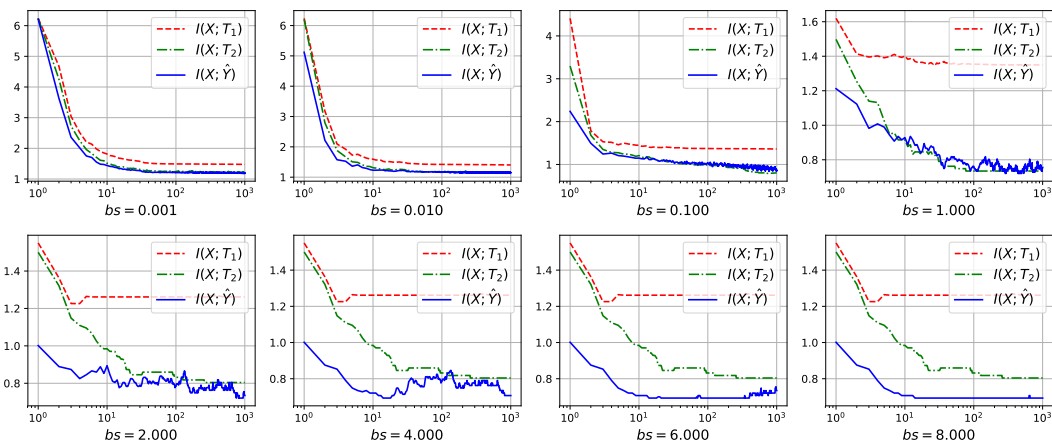

Figure 9: The estimation of $I(X; T_i)$ in MLP2 on the synthetic dataset via the empirical distribution with 8 different $bs$. All the x-axis index training epochs.

Similarly, Figure 10 and 11 show that KDE is also sensitive to the hyper-parameter, namely the noise variance $\sigma_n^2$. Figure 10 shows that $I(X; T_i)$ in different hidden layers of MLP1 converges to 2.0 as $\sigma_n^2$ increases from 0.01 to 16.0. Figure 11 shows that KDE derives different $I(X; T_1)$ and $I(X; T_2)$ in MLP2 given different $\sigma_n^2$, except $I(X; \hat{Y})$ converges to 1.0, as $bs$ increases from 0.01 to 16.0. Again, since the synthetic dataset only has 2 bits information, $I(X; T_i)$ must be smaller than $H(X) = 2$. However, we also observe that KDE derives $I(X; T_i) > 2.0$ when $\sigma_n^2 < 1.0$. Overall, different $\sigma_n^2$ make KDE to derive different mutual information estimations for $I(X; T_i)$ in MLP1 and MLP2 on the synthetic dataset, especially KDE does not correctly estimate $I(X; T_i)$ in MLP1 and MLP2 when $\sigma_n^2 < 1.0$.

In summary, the two non-parametric models are sensitive to hyper-parameters for mutual information estimation. Especially, since the entropy of the synthetic dataset is known, we can determine which hyper-parameter is appropriate to estimate the mutual information. However, if the entropy of dataset is unknown, it is very difficult to choose an appropriate hyper-parameter for non-parametric models to estimate the mutual information.

## G.2 Comparison to non-parametric models on the synthetic dataset

In this section, we compare the proposed mutual information estimator to the empirical distribution and KDE in MLP1 and MLP2 on the synthetic datset, and demonstrate that the proposed mutual information estimator derives more accurate mutual information estimation than non-parametric models. Based on Appendix G.1, we choose $bs = 2.0$ and $\sigma_n^2 = 2.0$ as the optimal hyper-parameters for the empirical distribution and KDE to estimate $I(X; T_i)$ and $I(Y; T_i)$. All the training methods are the same as Section 4.1.

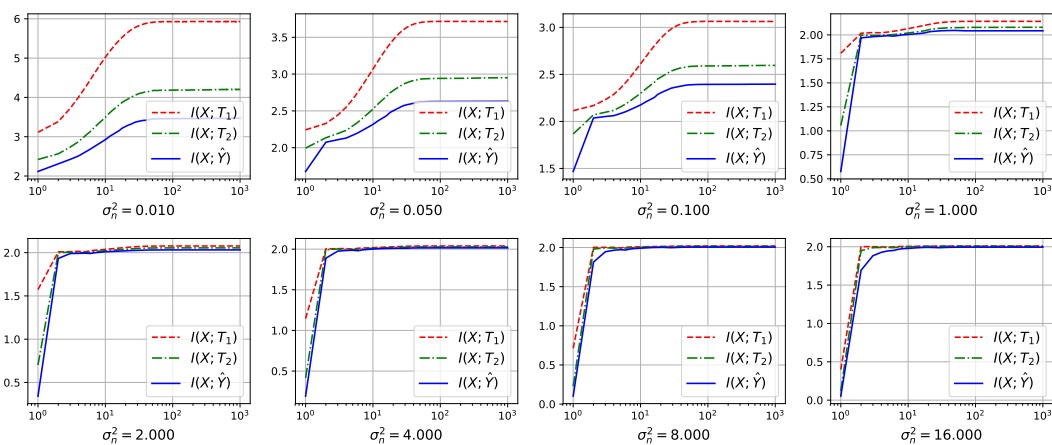

Figure 10: The estimation of $I(X; T_i)$ in MLP1 on the synthetic dataset by KDE with 8 different $\sigma_n^2$. All the x-axis index training epochs.

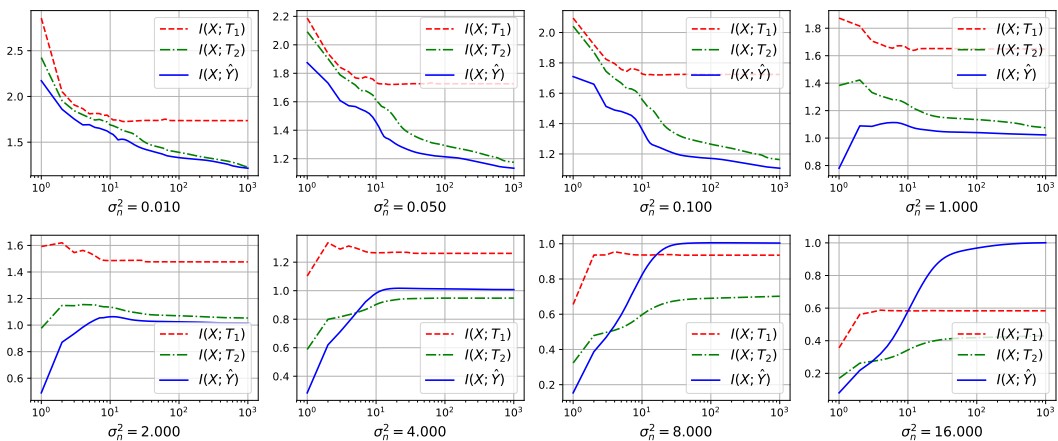

Figure 11: The estimation of $I(X; T_i)$ in MLP2 on the synthetic dataset by KDE with 8 different $\sigma_n^2$. All the x-axis index training epochs.

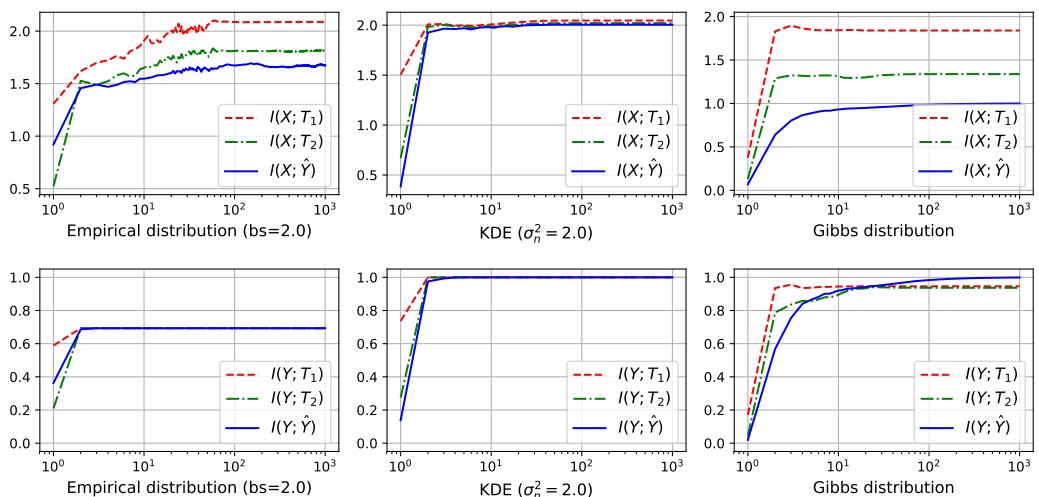

Figure 12: The estimation of $I(X; T_i)$ and $I(Y; T_i)$ in MLP1 based on the three mutual information estimators. All the x-axis index training epochs.

Figure 12 shows the estimation of $I(X; T_i)$ and $I(Y; T_i)$ in MLP1 derived by the three methods, namely the empirical distribution ($bs = 2.0$), KDE ($\sigma_n^2 = 2.0$), and the Gibbs distribution. Since $\hat{\boldsymbol{y}}$ only has two nodes, the maximal information of $X$ that $\hat{\boldsymbol{y}}$ can have is 1 bit, i.e., $I(X; \hat{Y}) \leq 1$, based on Definition 1. However, we observe that the empirical distribution derives $I(X; \hat{Y}) > 1.5$ and KDE derives $I(X; \hat{Y}) = 2.0$, thus the empirical distribution and KDE do not accurately estimate $I(X; \hat{Y})$. In addition, since MLP1 correctly predicts all the labels of synthetic images, it should have all the information of the labels. However, we observe that the empirical distribution estimates $I(Y; T_i) = 0.7$ bits, which contradicts the fact. As a comparison, the proposed method based on Gibbs distribution accurately estimate the information flow in MLP1.

Figure 13 shows the estimation of $I(X; T_i)$ and $I(Y; T_i)$ in MLP2 derived by the three methods. As shown in Figure 4, MLP2 quickly learns all the features of the synthetic dataset, thus $I(X; T_i)$ should have an increasing trend as training epochs increases. However, $I(X; T_i)$ estimated by the empirical distribution shows a decreasing trend, which contradicts the variation of the weights shown in Figure 4. Therefore, the empirical distribution does not accurately estimate $I(X; T_i)$ in MLP2. In addition, Section 4.2 shows that Tanh hinders $\boldsymbol{t}_1$ from correctly recognizing the features of input, thus $\boldsymbol{t}_1$ in MLP2 does not contain too much information of $X$, i.e., $I(X; T_1)$ is small. However, KDE estimates $I(X; T_1) > 1.5$, i.e., $\boldsymbol{t}_1$ in MLP2 has most information of $X$. Therefore, KDE does not correctly measures the effect of activation functions on the mutual information.

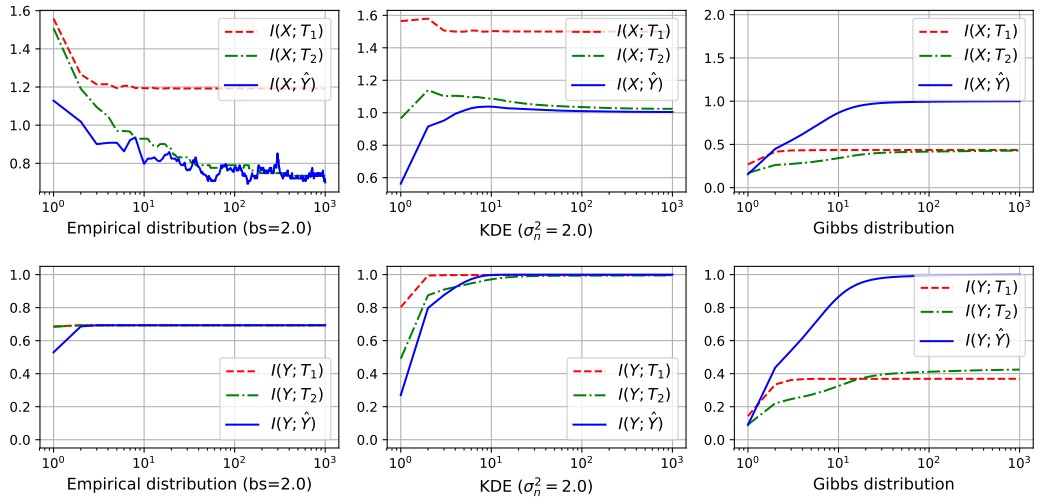

Figure 13: The estimation of $I(X; T_i)$ and $I(Y; T_i)$ in MLP2 based on the three mutual information estimators. All the x-axis index training epochs.

### G.3 Activations do not satisfy the i.i.d. prerequisite of non-parametric models

In this section, we demonstrate that one reason for non-parametric models deriving poor mutual information estimation is because activations do not satisfy the *i.i.d.* prerequisite of non-parametric models.

Given an input $\boldsymbol{x} \in \mathbb{R}^M$, we define the corresponding multivariate random variable as $X = [X_1, \cdots, X_M]$, where $X_m$ is the scalar-valued random variable of $x_m$. In the context of frequentist probability, all the parameters of MLPs are viewed as constants, thus the random variable of $\langle \boldsymbol{\omega}_n^{(1)}, \boldsymbol{x} \rangle = \sum_{m=1}^M \omega_{mn}^{(1)} \cdot x_m + b_{1n}$ is defined as $G_{1n} = \sum_{m=1}^M \omega_{mn}^{(1)} X_m + b_{1n}$, and the random variable of the activation $t_{1n} = \sigma_1(\langle \boldsymbol{\omega}_n^{(1)}, \boldsymbol{x} \rangle)$ is defined as $T_{1n} = \sigma_1(G_{1n})$. Therefore, the multivariate random variable of $\boldsymbol{t}_1 = [t_{11}, \cdots, t_{1N}]$ can be defined as $T_1 = [T_{11}, \cdots, T_{1N}]$. Similarly, we define the multivariate random variable of $\boldsymbol{t}_2$ as $T_2 = [T_{21}, \cdots, T_{2K}]$ and the multivariate random variable of $\hat{\boldsymbol{y}}$ as $\hat{Y} = [\hat{Y}_1, \cdots, \hat{Y}_L]$.

Samples being *i.i.d.* is the prerequisite of applying non-parametric models, e.g. the empirical distribution and KDE, to model the true distribution of a random variable [40]. In the context of MLPs, most previous works regard the activations of a layer as the samples of the random variable of the layer, and use non-parametric models to simulate the distribution of the layer. As a result, activations must be *i.i.d.* samples.

Since the necessary condition for samples being *i.i.d.* is the samples being uncorrelated, we can use the sample correlation to examine if activations being *i.i.d.*. More specifically, given two *i.i.d.* input samples $\boldsymbol{x}^j$ and $\boldsymbol{x}^{j'}$, the two activation vectors of the $i$th hidden layers are $\boldsymbol{t}_i^j$ and $\boldsymbol{t}_i^{j'}$. If $\boldsymbol{t}_i^j$ and $\boldsymbol{t}_i^{j'}$ are *i.i.d.* samples of $T_i$, the sample correlation $R(\boldsymbol{t}_i^j, \boldsymbol{t}_i^{j'})$ must be zero, namely

$$R(\boldsymbol{t}_i^j, \boldsymbol{t}_i^{j'}) = \frac{\sum_{n=1}^N (t_{in}^j - \bar{t}_i^j)(t_{in}^{j'} - \bar{t}_i^{j'})}{\sqrt{\sum_{n=1}^N (t_{in}^j - \bar{t}_i^j)^2 \sum_{n=1}^N (t_{in}^{j'} - \bar{t}_i^{j'})^2}} = 0, \tag{66}$$

where $\bar{t}_i^j = \frac{1}{N} \sum_{n=1}^N t_{in}^j$, and $N$ is the number of neurons in $\boldsymbol{t}_i$.

To study the sample correlation between activations given different samples, we use the Adam to train a MLP on the MNIST dataset [20] over 200 epochs with the learning rate $\alpha = 0.0005$. Since the dimension of each image is $28 \times 28$, the number of the input nodes is $M = 784$. In addition, $\boldsymbol{t}_1$, $\boldsymbol{t}_2$, and $\hat{\boldsymbol{y}}$ have $N = 96$, $K = 32$, and $L = 10$ neurons/nodes, respectively. All the activation functions are Tanh.

After training, we derive $R(\boldsymbol{t}_i^j, \boldsymbol{t}_i^{j'})$ on 5000 training samples $\{\boldsymbol{x}^j\}_{j=1}^{5000}$ and show the result in Figure 14. In particular, we rearrange the order of $\{\boldsymbol{x}^j\}_{j=1}^{5000}$ such that images with the same label have consecutive index, i.e., images with the label $l$ has the index $[l \times 500, (l + 1) \times 500)$, thus we can easily check the sample correlation between activations with the same label. Figure 14 shows that the sample correlation between activations with the same label becomes larger as the layer is closer to the output. In other words, activations are not *i.i.d.*. Therefore, it is invalid to apply non-parametric models to model the true distribution of all the layers of the MLP, because activations do not satisfy the *i.i.d.* prerequisite of non-parametric models.

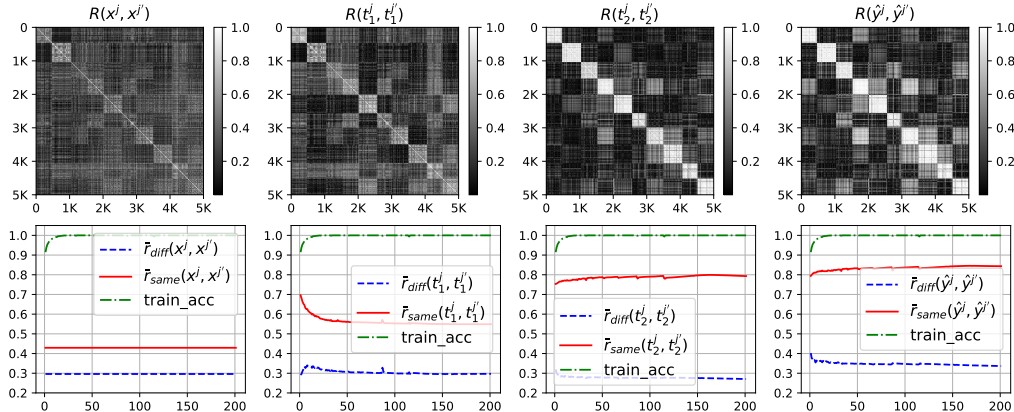

Figure 14: The first row shows that sample correlation between different samples/activations in each layer of the MLP after training. The second row shows the variation of the average sample correlation between different activations with different labels and with the same labels in each layer during training.

More specifically, Figure 14 shows that the sample correlation between each pair of training samples $\{x^j\}_{j=1}^{5000}$ is very small, thus *i.i.d.* can be viewed as a valid assumption for the input samples $\{x^j\}_{j=1}^{5000}$. However, we observe an ascending trend for the sample correlation between different activations with the same label as the layer is closer to the output. For instance, the pixels at the top-left corner of $R(t_i^j, t_i^{j'})$ becomes lighter as the layer is closer to the output, i.e., the sample correlation between the activations with the label 0 becomes larger.

In addition, the second row of Figure 14 also show the ascending trend, i.e., $\bar{r}_{\text{same}}(t_1^j, t_1^{j'})$, $\bar{r}_{\text{same}}(t_2^j, t_2^{j'})$, and $\bar{r}_{\text{same}}(\hat{y}^j, \hat{y}^{j'})$ converge to 0.55, 0.79, and 0.84, respectively, where $\bar{r}_{\text{same}}(t_i^j, t_i^{j'})$ denotes the average sample correlation of $\{t_i^j\}_{j=1}^{5000}$ with the same label in the $i$th hidden layer.

As a comparison, Figure 14 shows that the sample correlation of activations with different labels being relatively stable in different layers, because $\bar{r}_{\text{diff}}(t_1^j, t_1^{j'})$, $\bar{r}_{\text{diff}}(t_2^j, t_2^{j'})$, and $\bar{r}_{\text{diff}}(\hat{y}^j, \hat{y}^{j'})$ converge to 0.29, 0.27, and 0.33, respectively, where $\bar{r}_{\text{diff}}(t_i^j, t_i^{j'})$ denotes the average sample correlation of $\{t_i^j\}_{j=1}^{5000}$ with different labels.

In summary, the sample correlation of activations with the same label becomes larger as the layer is closer to the output, thus activations being *i.i.d.* is not valid for all the layers of the MLP. As a result, non-parametric models, e.g., the empirical distribution and KDE, cannot correctly simulate the true distribution of all the layers, thus they are invalid for estimating the mutual information between each layer and dataset.

# H  Experiments on benchmark dataset

To further demonstrate the information theoretic explanations for DNNs, we design more complicated neural networks and conduct experiments on the bechmark MNIST and Fashion-MNIST (abbr. FMNIST) dataset. The experiment codes are also available online[7].

## H.1  Experiments on the MNIST dataset

We design three MLPs, namely MLP4, MLP5, and MLP6, and summarize the architectures of the three MLPs in Table 4. We train the three MLPs on the MNIST dataset by Adam [15] over 500 epochs with the learning rate $\alpha = 0.0005$. Based on the mutual information estimator proposed in Section 4.1, we measure the information flow in the three MLPs during 500 training epochs.

In Figure 15, we observe that the information flow of $X$ in the three MLPs does not satisfy the Markov chain, namely Equation (2), proposed by previous works, *i.e.*, we further confirm that Equation (2) does not fully characterize the information flow of $X$, especially when taking into account of the back-propagation training.

Moreover, the second and the third row of Figure 15 show $I(\bar{X}; T_1) \geq I(\bar{X}; T_2) \geq I(\bar{X}; \hat{Y})$ and $I(Y; T_1) \leq I(Y; T_2) \geq I(Y; \hat{Y})$ in all the three MLPs, which further validate that Corollary 1, *i.e.*, Equation (14), correctly characterizes the information flow in MLPs.

The last row of Figure 15 shows that $I(X; T_{\text{MLP}}) > H(Y)$ and $I(Y; T_{\text{MLP}}) = H(Y)$ for most epochs in all the three MLPs. Though $H(X)$ is unknown for the MNIST dataset, we still can conclude that the three MLPs form three compressed representations of the data while preserve all the information of the labels. Hence, Figure 15 further confirms that a MLP satisfies the IB principle no matter what the architecture of the MLP is.

---

[7]`https://github.com/Dlib-NeurIPS/Deep-Learning-Information-Theory`

Table 4: The number of neurons(nodes) and the activation function in MLP4 - MLP6

|  | $x$ | $t_1$ | $t_2$ | $\hat{y}$ | $\sigma(\cdot)$ |
|---|---|---|---|---|---|
| MLP4 | 784 $(28 \times 28)$ | 96 | 32 | 10 | $\text{ReLU}(z) = \max(0, z)$ |
| MLP5 | 784 $(28 \times 28)$ | 96 | 32 | 10 | $\text{Tanh}(z) = (e^z - e^{-z})/(e^z + e^{-z})$ |
| MLP6 | 784 $(28 \times 28)$ | 32 | 96 | 10 | ReLU |

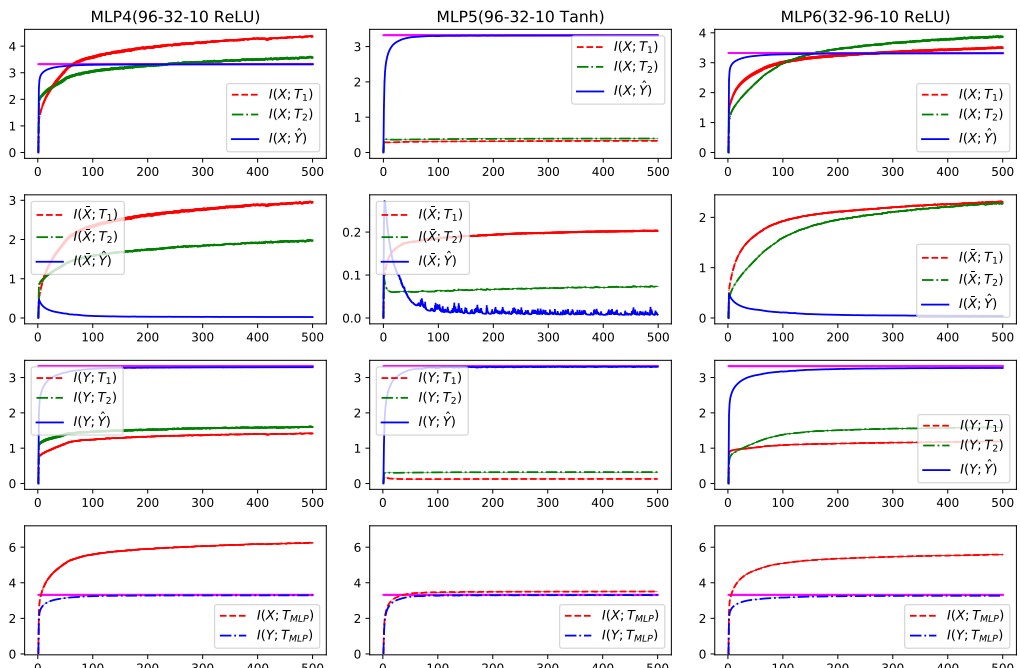

Figure 15: The information flow in MLP4, MLP5, and MLP6 on the MNIST dataset. All the x-axis index training epochs. In each column, the first three figures show $I(X; T_i)$, $I(\bar{X}; T_i)$, and $I(Y; T_i)$ respectively. The forth figure shows $I(X; T_{\text{MLP}})$ and $I(Y; T_{\text{MLP}})$ in a MLP. The pink line denotes $H(Y) = \log_2 10$.

## H.2 Experiments on the Fashion-MNIST dataset

We design three MLPs, namely MLP7, MLP8, and MLP9, and summarize the architectures of the three MLPs in Table 5. Compared to the MLPs on the MNIST dataset, the three MLPs has one more hidden layer and each hidden layer has more neurons, *i.e.*, the MLPs are more complicated. Similarly, we train the three MLPs by Adam [15] over 500 epochs with the learning rate $\alpha = 0.0005$. Based on the mutual information estimator proposed in Section 4.1, we measure the information flow in the three MLPs during 500 training epochs.

Figure 16 shows similar results as Section 4.3 and Section H.1, thus it further confirms the information theoretic explanations for DNNs.

Table 5: The number of neurons(nodes) and the activation function in MLP7 - MLP9

|  | $x$ | $t_1$ | $t_2$ | $t_3$ | $\hat{y}$ | $\sigma(\cdot)$ |
|---|---|---|---|---|---|---|
| MLP7 | 784 $(28 \times 28)$ | 256 | 128 | 96 | 10 | $\text{ReLU}(z) = \max(0, z)$ |
| MLP8 | 784 $(28 \times 28)$ | 256 | 128 | 96 | 10 | $\text{Tanh}(z) = (e^z - e^{-z})/(e^z + e^{-z})$ |
| MLP9 | 784 $(28 \times 28)$ | 96 | 128 | 256 | 10 | ReLU |

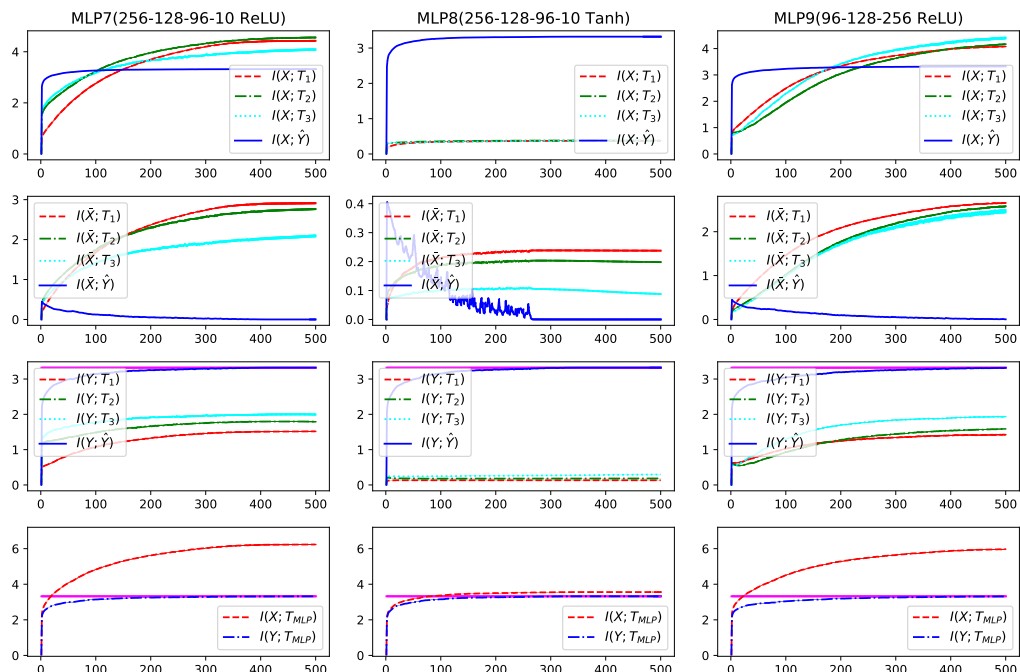

Figure 16: The information flow in MLP7, MLP8, and MLP9 on the MNIST dataset. All the x-axis index training epochs. In each column, the first three figures show $I(X; T_i)$, $I(\bar{X}; T_i)$, and $I(Y; T_i)$ respectively. The forth figure shows $I(X; T_{\text{MLP}})$ and $I(Y; T_{\text{MLP}})$ in a MLP. The pink line denotes $H(Y) = \log_2 10$.