# OpenReview forum: "A Probabilistic Representation for Deep Learning: Delving into The Information Bottleneck Principle"
_NeurIPS.cc/2021/Conference — NeurIPS 2021 Submitted_

### Official Review · Reviewer_hXwD · 2021-07-12

**Rating:** 3
**Confidence:** 4

**Summary:**

The paper presents a new probabilistic model for studying information flows in DNNs. To that end two Markov chains connecting the network's input, hidden layers, and output are introduced: (i) the forward chain models the conditional distribution of each layer given its input as a Gibbs distribution over the layer's weight vectors (the Hamiltonian is given by a non-linearity applied of dot products between weights and inputs); (ii) the backward chain, which is discussed much less, tries to capture the correlations introduces by BP. The authors show how to compute mutual information terms under their probabilistic model and provide some synthetic experiments under the proposed framework.

**Ethical Concerns:**

None.

**Limitations And Societal Impact:**

Not applicable.

**Main Review:**

The general idea of formulating a probabilistic model under which information flows in DNNs may be studied (as opposed to the statistical route based on estimation from samples) is solid. However, I am afraid that the execution presented in this paper is sub par. For such a newly proposed model to be adopted the formulation must be described in a crystal clear manner and be well motivated by theory and empirics. Unfortunately, the paper falls short on both aspects, as is elaborated below. As a result, I cannot recommend acceptance.

My main concerns (along with some smaller ones) are as follows:

1. The technical writing is sub par. The authors' choice to emphasize on the probability spaces corresponding to each layer does not make sense to me. They should just describe the random variables of interest, their distributions, supports, correlations, etc. Sample spaces and $\sigma$-algebras can be left implicit for several reasons. First, one can always find a common probability space over which all random variables are defined (e.g., any random variable with values in a polish space can be generated from a uniform (0,1) variable, so one only has to assume that this underlying space is 'rich enough' to support such a uniform variable). More importantly, the probability spaces $(\Omega_{T_i},\mathcal{F},P_{T_i})$ does not encode the entire probabilistic model since the latter is defined through conditional probabilities that are absent here.

2. Continuing from the above, the validity of the proposed model and its relation to the original IB formulation is unclear to me. The IB paradigm views $T_i$ as the vector of hidden representations (neuron values) corresponding to layer $i$. Here the authors think of $T_i$ as the weight vector whose correlation with the layer's input is highest. I can't see why this is a reasonable definition, and more importantly, its relation to the IB formulation is questionable.  This deviation from the IB principle is problematic given that the proposed model is motivated by it.

3. The Markov chain formulation is also unclear. First, if $X \to Y \to Z$ forms a Markov chain then so does $Z\to Y\to X$, so the discussion of the chain's directionality should take this into account. Namely, it is more a question of which conditional distributions are specified and not of whether there is a Markovity relation in a given direction. Since the authors discuss both directions of the chain, do we know that the conditional distributions correspond (i.e., give rise to the same joint measure)? Say we have $Y \to X \to T_1\to \hat{Y}$ with law $P_{X,Y,T,\hat{Y}}=P_{X,Y}P_{T|X}P_{\hat{Y}|T}$ and $Y\to \hat{Y} \to T$ with law $Q_{Y,T,\hat{Y}}$. Do we know that, e.g., $P_{Y,T,\hat{Y}}=Q_{Y,T,\hat{Y}}$? If the authors mean that there are two phases of updates (forward and backward), then they should define more variables to reflect that. The fact that this is still unclear to me after reading the paper testifies that writing (and possible the model itself) must be improved

4. The paper lacks in depth as far as the provided claims (Theorem 1, Proposition 1, etc.). These are trivial consequences of the proposed model. Some claims seem somewhat exaggerated, e.g., the proof in Appendix D that $H(Y)=I(X;Y)$. There is nothing to prove here as, given $X=\mathbf{x}^j$, the authors just take $Y=\ell \mathbf{1}_{\{y^j=\ell\}}$. This trend of overcomplicating simple observations repeats throughout the entire text.

6. The authors say their model alleviates the infinite mutual information issue. While this is true, it is achieved by trivializing the problem. The authors implicitly assume that the data distribution is supported only over the samples $\{\mathbf{x}^j\}$, in which case clearly $I(X;Y)\leq H(X)<\infty$. There is no probabilistic model for $P_X$ outside the sample set. Consequently there is no sense in which the proposed model can account for generalization, which is a key aspect to statistical learning theory.

7. The authors wait too long before they describe the proposed model. Vague descriptions are provided from the abstract through the introduction, but nowhere in the intro is the model actually described (even in words). There is a paragraph on page 2 that talks about the probability spaces attached to each layer, but as mentioned above this is quite redundant and doesn't clarify the probabilistic model. The model is described first only after Def 1 on page 3, which is too late. Mention of the Gibbs measure, its Hamiltonian, the induced Markov chains, etc. should be provided earlier in the text to improve readability.

8. The paper is tedious to read. The technical descriptions are swamped in verbal and heuristic statements, making it hard to extract the message being conveyed. Because of that I had to reread many parts several times. The authors should clean up and streamline the text. This would make it easier to understand the proposed model and appreciate its potential validity. I am not providing specific examples since there are just too many---the entire text needs work.

9. In Equation (10), do the authors mean conditional KL divergence (i.e., given $X=x$) or averaged one over the samples? The dependence of the KL term on the weights should be made explicit.

10. The proof outline for Theorem 2 is confusing. It uses undefined notation such as $\frac{\partial\ell_{\mathrm{CE}}}{\partial\omega}$ and does little to explain where the statement comes from. The authors should either shape it into a standalone argument (statements such as `it can be shown' should be shown or at least explain further) or omit it from the main text.

11. I can't see how the experiments validate the applicability of the proposed theory. The authors compute the proposed objects but whether these in fact describe real world DNNs remains unclear.

**Time Spent Reviewing:**

6 hours

---

> ### Author Response · Authors · 2021-08-06
> **Author response to specific questions of the reviewer**
>
> 1) 'the probability spaces does not encode the entire probabilistic model since the latter is defined through conditional probabilities that are absent here.'
>
> **We respond to most concerns of List 1 in [Author response to the concern of the main idea of the paper](https://openreview.net/forum?id=6ugK-RQhIP5&noteId=u3XDZl2fMYP). Here we only response the above concern. Directly explaining the entire network in a probabilistic way is very difficult. Alternatively, we define the probability space of a hidden layer and derive the distribution of the entire network based on Markov chains. Since many layers have the same architecture, e.g., all the layers of MLPs are fully connected layers, we think defining the probability space of a single hidden layer and using Markov chains to derive the probabilistic explanation for the entire network is still valid and novel.**
>
> 2) the validity of the proposed model and its relation to the original IB formulation is unclear to me. The IB paradigm views $T_i$ as the vector of hidden representations (neuron values) corresponding to layer. Here the authors think of $T_i$ as the weight vector whose correlation with the layer's input is highest.
>
> **Most previous works use non-parametric models to estimate $I(X;T_i)$ and $I(Y; T_i)$ and study the IB principle in DNNs. However, non-parametric models are known for poor mutual information estimation in high dimensional set and lead to severe limitations and controversies in terms of information theoretic explanations for DNNs. Therefore, the key to resolve the existing limitations of IB explanations for DNNs is accurately estimating $I(X;T_i)$ and $I(Y; T_i)$.**
>
> **As we state in [why should we study the probability space in the context of deep learning and information theory](https://openreview.net/forum?id=6ugK-RQhIP5&noteId=zIZqhpFOQtU), probability space is the foundation of information theoretic analysis. Specifically, specifying the probability space $(\Omega, \mathcal{F}, P)$ of a hidden layer $\boldsymbol{t}_i$, thus we can specify the random variable $T$ and the distribution of a hidden layer, namely $P(T)$, thereby accurately estimating $I(X;T)$ and $I(Y; T)$ and thoroughly studying the IB principle in DNNs.**
>
> 3) Since the authors discuss both directions of the chain, do we know that the conditional distributions correspond (i.e., give rise to the same joint measure)? Say we have $X \rightarrow T_1 \rightarrow \hat{Y}$ with law $P_{X, T_1, \hat{Y}} = P_{X}P_{T_1|X}P_{\hat{Y}|T_1}$ and $Y \rightarrow \hat{Y} \rightarrow T_1$ with law $Q_{Y, \hat{Y}, T_1}$. Do we know that, e.g., $P_{Y, \hat{Y}, T_1} = Q_{Y, \hat{Y}, T_1}$? If the authors mean that there are two phases of updates (forward and backward), then they should define more variables to reflect that.
>
> **That is a very good question. Thank you for asking. In a word, Figure 2 in the paper answers your question.**
>
> **In the forward phase, neural networks derive the probability of the marginal distribution $P(\hat{Y}|X)$ given input samples to simulate the statistical connection between input samples and target labels.**
>
> **In the backward phase, the back-propagation optimizes the weights to minimize the distance between $P(\hat{Y}|X)$  and $P(Y|X)$ via Equation (10). Though the weights are optimized, the distribution of each layer is still a Gibbs distribution. Therefore, we NOT need to define more variables for reflecting the backward phase. A more intuitive example could be using the Maximal Likelihood Estimation (MLE) to learn a Gaussian distribution $\mathcal{N}(\mu, \sigma^2)$ for fitting given samples. Even though $\mu$ and $\sigma^2$ would change after each learning iteration, only a single Gaussian distribution is defined.**
>
> **Overall, though there are two phases of updates (forward and backward), the distribution of each layer is still a Gibbs distribution in both phases, thus we can still use the same variable to show the information flow in DNNs. Please let me know if you have more questions.**
>
> 4) The paper lacks in depth as far as the provided claims (Theorem 1, Proposition 1, etc.). These are trivial consequences of the proposed model. This trend of overcomplicating simple observations repeats throughout the entire text.
>
> * **We hope the reviewer notice that all the claims (Theorem 1, Proposition 1, etc.) are derived from the probability space definition for a hidden layer, rather than neural networks basic. If the consequences of the proposed model are trivial, it exactly shows that the probabilistic explanations are consistent with the common sense of neural networks. In other words, it demonstrate that the probabilistic representation derived from the probability space correctly explains the functionality of neural network from the probabilistic viewpoint.**
>
> * **We NOT intend to overcomplicate simple observations. Instead, we aim to provide a clearly probabilistic explanation for the observations in neural networks.**
>
> 5) The authors say their model alleviates the infinite mutual information issue. While this is true, it is achieved by trivializing the problem. The authors implicitly assume that the data distribution is supported only over the samples $\boldsymbol{x}^j$, in which case clearly $I(X;Y) < H(X) < \infty$. There is no probabilistic model for outside the sample set. Consequently there is no sense in which the proposed model can account for generalization, which is a key aspect to statistical learning theory.
>
> * **The previous work  [1] and [2] show that $I(X;T_i)$ is unbounded because $T_i$ is regarded as a continuous random variable and DNNs are regarded as deterministic functions (Please refer Appendix C in [2] for detail). This paper aims to resolve the unbounded mutual information problem via specifying the random variable $T_i$ based on the probability space. Specifically, the paper shows that $I(X;T_i) < H(T_i)< \infty$  due to $T_i$ is a discrete random variable in Proposition 1. This is not related to the random variable $X$.**
>
> * **The probabilistic explanation is designed to model the architecture of a hidden layer, rather than the sample set. In the context of information bottleneck explanation for generalization [3], the compression of $I(X;T_i)$ is related to generalization, thus accurate mutual information based on the probabilistic representation still plays an important role in predicting the generalization.**
>
> [1] Rana Ali Amjad, et al. Learning representations for neural network-based classification using the information bottleneck principle. IEEE transactions on pattern analysis and machine intelligence, 2019.
>
> [2] Andrew Saxe, et al. On the information bottleneck theory of deep learning. In International Conference on Representation Learning, 2018.
>
> [3] Ravid Shwartz-Ziv and Naftali Tishby. Opening the black box of deep neural networks via information. arXiv preprint arXiv:1703.00810, 2017.
>
> 6) The authors wait too long before they describe the proposed model. Vague descriptions are provided from the abstract through the introduction, but nowhere in the intro is the model actually described (even in words). There is a paragraph on page 2 that talks about the probability spaces attached to each layer, but as mentioned above this is quite redundant and doesn't clarify the probabilistic model.
>
> **One important reason for feeling the paper takes too much space before describing the proposed model could be the different understanding of the importance of probability space. [Why should we study the probability space in the context of deep learning and information theory](https://openreview.net/forum?id=6ugK-RQhIP5&noteId=zIZqhpFOQtU) could help the reviewer to understand the main idea of the paper.**
>
> 7) The technical descriptions are swamped in verbal and heuristic statements, making it hard to extract the message being conveyed. Because of that I had to reread many parts several times. The authors should clean up and streamline the text. This would make it easier to understand the proposed model and appreciate its potential validity.
>
> **We suggest the reviewer specify the unclear parts or sections, so we can provide specific explanations.**
>
> 8) In Equation (10), do the authors mean conditional KL divergence (i.e., given $X=x$) or averaged one over the samples?
>
> **We use the classical cross entropy loss function, namely $\ell_{\text{CE}} = \frac{1}{J}\sum_{j=1}^J \text{KL}[P(Y|X=\boldsymbol{x}^j)||P(\hat{Y}|X=\boldsymbol{x}^j)]$ in Equation (10).**
>
>
> 9) The proof outline for Theorem 2 is confusing. It uses undefined notation such as $\frac{\partial \ell_{\text{CE}}}{\partial\omega}$ and does little to explain where the statement comes from.
>
> **We define $\frac{\partial \ell_{\text{CE}}}{\partial \omega}$ as the gradient of the cross entropy loss $\ell_{\text{CE}}$ with respect to the weight $\omega$ in L142 and Equation (10).**
>
> 10) I can't see how the experiments validate the applicability of the proposed theory. The authors compute the proposed objects but whether these in fact describe real world DNNs remains unclear.
>
> * **In Appendix G.2, the proposed mutual information estimator is shown outperforming the existing non-parametric models**
>
> * **In Appendix H.1, the proposed Markov chains are validated by three networks on the MNIST dataset**
>
> * **In Appendix H.2, the proposed Markov chains are validated by three networks on the Fashion-MNIST dataset**
>
> ***
>
> **Please let us know if you have more questions. Hope you can reconsider the paper. Thank you!**

---

> > ### Comment · Reviewer_hXwD · 2021-08-26
> > **Answer to authors' response**
> >
> > I thank the authors for their detailed answers to my questions. However, the response did little to mitigate my concerns. I find the formulation and presented derivations to lack rigor, the presentation is often verbose and makes it hard to extract the essence, but perhaps most importantly, I am not convinced by the proposed probabilistic model.
> >
> > The authors mention in their response that the simplicity of their derivations originates from pinning down a good probabilistic model. While the simple derivations indeed come from the way the problem is set up, this does not mean that the setup is good/meaningful, which further raises questions of how interesting the results are. Even if there is an interesting idea hiding here, the paper does not convey it in a clear and streamlined fashion. I think that the magnitude of changes needed to make this paper publishable is much beyond the scope of a revision.
> >
> > I also disagree with the authors' claim that their model enables to `estimate MI more accurately than non-parametric techniques'. Strictly speaking, they don't estimate MI (as in statistical estimation from samples). Rather, MI is computed based on the proposed model after imposing a lot of structure on the problem. As it is not clear to me if the hypothesized structure is valid, I am not sure what the resulting MI values teach us.
> >
> > The argued necessity of spelling out the entire probability spaces for each layer is also odd to me. The posed model amounts to discrete random variables and we know that Shannon entropy or MI are invariant to relabelings (bijections). So one can relabel their $\Omega_{T_i}$ sample sets arbitrarily, while keeping the same probability masses, and nothing would change in terms of the corresponding information measures.
> >
> > Overall, I think that the amount of back and forth needed to justify the model testifies to the fact that it needs more work or flushing out. Optimally, the convoluted and long descriptions in the paper and the author response would be substituted with a clear and rigorous mathematical formulation. Unfortunately the paper is not there yet, as evident by the reviews and the followup discussions. I am happy with my original score.

---

> ### Author Response · Authors · 2021-08-06
> **Author response to the concern of the main idea in the paper**
>
> Thank you for your time and efforts in the paper, we respond to your concerns of **the main idea of the paper** below.
>
> ---
>
> * "The general idea of formulating a probabilistic model under which information flows in DNNs may be studied (as opposed to the statistical route based on estimation from samples) is solid".
>
> **Thanks the reviewer agrees that the main idea of the paper (*i.e., an explicit probabilistic representation for DNNs is the foundation of accurate mutual information estimation*) is solid,**
>
> * "For such a newly proposed model to be adopted the formulation must be described in a crystal clear manner and be well motivated by theory and empirics".
>
> **We fully understand and agree with the reviewer that such a newly proposed model must be described in a crystal clear manner and be well motivated by theory and empirics. We actually spent lots of efforts to describing the probabilistic representation in a crystal clear manner in both theory and empirical aspects in the paper. However, there could be some misunderstanding or unclear points for the reviewer.**
>
> **Based on the above two points, we think that the authors and the reviewer keep consistent with the main idea of the paper. The only controversy is if the authors clearly and thoroughly present the main idea (*i.e., an explicit probabilistic representation for DNNs is the foundation of accurate mutual information estimation*). If there are something else related to the main idea, please feel free let us know. We response the reviewer's specific concerns of the main idea below.**
>
> ---
>
> 1) The authors' choice to emphasize on the probability spaces corresponding to each layer does not make sense to me. They should just describe the random variables of interest, their distributions, supports, correlations, etc. Sample spaces and $\sigma$-algebras can be left implicit for several reasons.
>
> **We respectfully disagree that we should leave the probability space be implicit because of two reasons.**
>
> **First, [why should we study the probability space in the context of deep learning and information theory](https://openreview.net/forum?id=6ugK-RQhIP5&noteId=zIZqhpFOQtU) shows that probability space is the foundation of information theoretic analysis.
> Specifically, if the probability space of a random process is clear, then the corresponding random variable and the distribution can be specified, thus accurate entropy estimation can be derived. i.e., accurate entropy estimation extremely depends the explicit definition of the probability space of a random process.**
>
> For a simple random process, e.g., flipping a coin or measuring the weight of a random coin, we could overlook the underlying the probability space. That is because the probability space definition is simple and straightforward. However, we actually still define the probability space in an implicit way.  A counter-example is that nobody uses the probability space of measuring the coin weight to derive the random variable of flipping coin. It implies that we have already implicitly defined the probability space of flipping a coin.
>
> **Second, the probability space provides a fundamental approach to resolve the limitations of  information theoretic explanations for DNNs. The classical non-parametric models derive severe limitations and controversies in terms of information theoretic explanations for DNNs. In addition, non-parametric models are known for poor mutual information estimation [1, 2] in high dimensional set. As a result, merely using non-parametric models cannot resolve the limitations and controversies. Following the logic in [why should we study the probability space in the context of deep learning and information theory](https://openreview.net/forum?id=6ugK-RQhIP5&noteId=zIZqhpFOQtU), if we specify the probability space of a hidden layer, then we can specify the random variable and the distribution of a hidden layer, and thoroughly study the information flow in DNNs.**
>
> **Compared to non-parametric models simply regard the activations of a hidden layer as the samples of the random variable and indirectly estimate the distribution, the probability space provides a comprehensive probabilistic framework to specify the rand variable and the distribution, thereby deriving more convincing explanations and more accurate mutual information estimation.**
>
> **Based on the above two reasons, we argue that probability space is the foundation of information theoretic analysis of DNNs. Please let us know if you have different opinions. Thank you!**
>
> [1] Liam Paninski. Estimation of entropy and mutual information. Neural computation, 15(6):1191–1253, 2003.
>
> [2] David McAllester and Karl Stratos. Formal limitations on the measurement of mutual information. In International Conference on Artificial Intelligence and Statistics, pages 875–884. PMLR, 2020.
>
> ---
>
> 2) one can always find a common probability space over which all random variables are defined  (e.g., any random variable with values in a polish space can be generated from a uniform (0,1) variable, so one only has to assume that this underlying space is 'rich enough' to support such a uniform variable)
>
> **The claim 'one can always find a common probability space over which all random variables are defined' intuitively looks correct. However, the claim contradicts the fundamental relation between probability space and random variable. Specifically, it is the probability space that determines the random variable, rather than the random variable determines the probability space.**
>
> **Let's first use your example to show the point. In the example, you state the random variable $X: \Omega \rightarrow E$ with values in a polish space, i.e., you regard the measurable space $E = \mathbb{R}$, thus you can rescale $\mathbb{R}$ to (0, 1) somehow. That is totally okay. However, your example implies that you implicitly regard the sample space $\Omega$ with real values as well based on the random variable definition. In other words, before you claim the random variable $X: \Omega \rightarrow E$ in a polish space, you already implicitly regard  the sample space $\Omega$ as real values, as the random variable definition $X: \Omega \rightarrow E$ indicates that the sample space  $\Omega$ determines the measurable space $E$.**
>
> **The second example is [why should we study the probability space in the context of deep learning and information theory](https://openreview.net/forum?id=6ugK-RQhIP5&noteId=zIZqhpFOQtU). Specifically, if it is to flip the coin, the sample space $\Omega = \\{\text{Head}, \text{Tail}\\}$, thus the random variable is $X = \\{0, 1\\}$. As a comparison, if it is to measuring the weight of the coin, $\Omega = \mathbb{R}$, thus $X = \mathbb{R}$. In other words, if only a coin is given without any information of the probability space (i.e., we do not know it is to flip the coin or to measure the weight of the coin), we cannot determine the random variable corresponding to the coin. The two examples show that it is the probability space that determines the random variable, rather than the random variable determines the probability space.**
>
> ---
>
> 3) **Finally, back to the mutual information estimation in DNNs.**
>
> **In contrast to most previous works simply using non-parametric models to estimate the mutual information, this paper follows the logic of in [why should we study the probability space in the context of deep learning and information theory](https://openreview.net/forum?id=6ugK-RQhIP5&noteId=zIZqhpFOQtU) for accurate mutual information estimation, which enable us to resolve the existing limitations of information theoretic explanations and validate the information bottleneck principle in DNNs.**
>
> Mutual Information: $I(X;T)$ and $I(Y;T)$
>
> $\quad$ $\quad$ $\qquad$ $\qquad$ $\qquad$ $\uparrow$
>
> Distribution: $\qquad$ $\quad$ $P(T)$
>
> $\quad$ $\quad$ $\qquad$ $\qquad$ $\qquad$ $\uparrow$
>
> Random Variable: $\quad \text{ }$ $T: \Omega \rightarrow E$
>
> $\quad$ $\quad$ $\qquad$ $\qquad$ $\qquad$ $\uparrow$
>
> Probability space $(\Omega,\mathcal{F},P)$
>
> ---
> **Please let us know if you have more questions. If the responses resolve the concerns, we hope the reviewer reconsider the paper.**

---

### Official Review · Reviewer_e1nM · 2021-07-15

**Rating:** 4
**Confidence:** 3

**Summary:**

This work introduces a novel probabilistic formulation for Deep Neural Network layers to circumvent the problem of estimating mutual information between continuous variables. The authors characterize two different directions for the information flow during training and inference time, empirically analyzing and validating their theory on a simple task.

**Limitations And Societal Impact:**

The paper does not clearly state the limitations of the proposed method.

the paper has a prevailing theoretical component, therefore the societal impact is not directly discussed.

**Main Review:**

The paper addresses the relevant problem of analyzing and characterizing the information flows in deep neural networks, by considering the Boltzman distribution on the activation of each layer and factorizing the dependencies in the backward training parameter update step.
Several parts of the paper are not entirely clear and not all the statements are strongly supported or sound.

Here I summarize my main concerns:

1) The discrete variable $T_i$ for each layer is introduced as a function of the activations $\bf t_i$, but clearly the activations of layer $i+1$ depend on $\bf t_i$ rather than the categorical variable denoting the "maximum feature correlation" $T_i$, which does not contain the entirety of the information regarding the activations. The corresponding graphical model follows the following chain of dependencies:
$$\begin{matrix} X & \to  & {\bf t}_1  & \to  & {\bf t}_2 & \to  & \hat{Y} \\\\  &  & \downarrow & & \downarrow \\\\ & & T_1 & & T_2\end{matrix}$$
Theorem 1 and equation 8 factorize $p(\hat{Y},T_2,T_1|X)$ as $p(\hat{Y}|T_2)p(T_2|T_1) p(T_1|X)$ without considering the dependencies $X\to T_2$ and $X\to\hat{Y}$. Why are these dependencies not considered?

2) Since $T_i$ is defined based on ${\bf t}_i$, as a consequence of the data processing inequality, any evaluation of $I(X; T_i)$ represents only a lower bound of the actual measure of information $I(X;{\bf t}_i)$ flowing through the layer of interest.

3) The paper claims that $I(X; T_i)$ is generally unbounded in the continuous case, but provides estimators based on the empirical distribution $P(X=x) = \frac{1}{|\mathcal{D}|}[x\in\mathcal{D}]$, which is discete by definition. In this case ,the empirical entropy is finite $H(X)=\log|\mathcal{D}|\le \infty$ and so is the mutual information in analysis $I(X; T_i)\le H(X) = \log|\mathcal{D}|$.

4) The main text does not contain a clear definition of $\bar{X}$ and $f_n$ (Line 103 and 192). The term "dataset" is used to refer to either $\mathcal{D}$ or $X$ in different sections.

5) The paper is generally quite notation-heavy, which makes it difficult to read in some sections.

6) The experimental section in the main text is not extensive enough to be significant or convincing

Minor comments:

- Figure 1 is difficult to interpret and does not add much to the main text of the paper.
- Line 106: $w_n$ used instead of $\omega_n$

**Time Spent Reviewing:**

4.5 h

---

> ### Author Response · Authors · 2021-08-05
> **Author Response**
>
> Thank you for the comments on the paper. We respond to your concerns below.
>
> 1.1) The discrete variable $T_i$ for each layer is introduced as a function of the activations $\boldsymbol{t}_i$, but clearly the activations of layer  $i+1$ depend on $\boldsymbol{t}_i$ rather than the categorical variable denoting the "maximum feature correlation", which does not contain the entirety of the information regarding the activations.
>
> **Based on the definition of random variable, the random variable $T_i: \Omega_i \rightarrow E_i$ is a function mapping the sample space $\Omega_i$ to the measurable space $E_i$. In other words, $T_i$ is actually not a function of $\boldsymbol{t}_i$. Please refer the examples in [why should we study the probability space in the context of deep learning and information theory](https://openreview.net/forum?id=6ugK-RQhIP5&noteId=zIZqhpFOQtU) for the definition of random variable.**
>
> **You are right, $t_{i+1}$ depend on ${t}_i$. This paper NOT aims to falsify or contradicts the fact, but tries to explicitly formulate the dependence in a probabilistic way, thus we can accurately measure the mutual information.**
>
> **Specifically, based on the definition of convolution, we explain $t_{i+1} = \sigma(<t_{i}, {\omega}_n>)$ as**
> **$t_{i+1}$ measuring the cross-correlation between $t_i$ and the feature $\omega_n$ defined by the weights of neurons in the $i+1$ layer, and define the Boltzmann distribution $P(\omega_n|t_i)$ to measure the probability of $\omega_n$ being recognized the feature with the largest cross-correlation to $t_i$**
>
> 1.2) Theorem 1 and equation 8 factorize $p(\hat{Y},T_2,T_1|X)$ as $p(\hat{Y}|T_2)p(T_2|T_1)p(T_1|X)$ without considering the dependencies $X \rightarrow T_2$ and $X \rightarrow \hat{Y}$. Why are these dependencies not considered?
>
> **That is a good question. Thank you for asking.  As we stated in L133, equation 8 discusses the information flow in DNNs without considering the back-propagation training. If not considering the back-propagation training, all the weights are fixed, thus $t_{i+1}$ only depends on $t_i$. In other words,  we do not need to consider the the dependencies of $t_{i+1}$ on $x$ if $t_i$ is given, namely $p(T_{i+1}|T_i, X) = p(T_{i+1}|T_i)$. Therefore, we can derive the Markov chain $X \rightarrow T_1 \rightarrow T_2 \rightarrow \hat{Y}$, and derive $p(\hat{Y},T_2,T_1|X) = p(\hat{Y}|T_2)p(T_2|T_1)p(T_1|X)$.**
>
> **The factorization $p(\hat{Y},T_2,T_1|X) = p(\hat{Y}|T_2)p(T_2|T_1)p(T_1|X)$ is a statistical property of Markov chain. Please refer Chapter 4.1 (Page 72) in [1] for detail. Hope that resolve your concern, feel free let me know if you have more questions.**
>
> 2) Since $T_i$ is defined based on $\boldsymbol{t}_i$, as a consequence of the data processing inequality, any evaluation of
> $I(X;T_i)$ represents only a lower bound of the actual measure of information $I(X; \boldsymbol{t}_i)$ flowing through the layer of interest.
>
> **In general, when we talk about the mutual information $I(A;B)$, $A$ and $B$ must be two random variables. Since $\boldsymbol{t}_i$ is an intermediate variable, not a random variable, it is inappropriate to discuss $I(X; \boldsymbol{t}_i)$.**
>
> **I guess your concern is that the proposed mutual information estimator is not accurate.  If my guess is correct, Section 4.2 can address your concern. It shows that the mutual information estimator correctly measures the information learned by networks from the synthetic dataset. In addition, Appendix G.2 shows that the proposed mutual information estimator outperforms the existing non-parametric models.**
>
> **If you have more questions, please let us know, we will try our best to resolve.**
>
> 3) The paper claims that $I(X;T_i)$ is generally unbounded in the continuous case, but provides estimators based on the empirical distribution $P(X=x)=\frac{1}{\mathcal{D}}[x\in\mathcal{D}]$, which is discrete by definition.
>
> **Actually, it is the previous work [2] and [3] that reveal $I(X;T_i)$ is unbounded because $T_i$ is regarded as a continuous random variable and DNNs are regarded as deterministic function (Please refer Appendix C in [3] for detail). This paper aims to resolve the unbounded mutual information problem via specifying the random variable $T_i$ based on the probability space. This is not related to the random variable $X$. Does that resolve your concern? Please let us know if you have more questions.**
>
> [1] Thomas Cover and Joy Thomas. Elements of Information Theory. Wiley-Interscience, Hoboken, New Jersy, 2006.
>
> [2] Rana Ali Amjad and Bernhard Claus Geiger. Learning representations for neural network-based classification using the information bottleneck principle. IEEE transactions on pattern analysis and machine intelligence, 2019.
>
> [3] Andrew Saxe, Yamini Bansal, Joel Dapello, Madhu Advani, Artemy Kolchinsky, Brendan Tracey, and David Cox. On the information bottleneck theory of deep learning. In International Conference on Representation Learning, 2018.
>
> 4) The main text does not contain a clear definition of $\bar{X}$ and $\boldsymbol{f}_n$ (Line 103 and 192). The term "dataset" is used to refer to either $\mathcal{D}$ or $X$ in different sections.
>
> **$\bar{X}$ is defined as a virtual random variable containing all the information of $X$ except $Y$, namely $H(\bar{X}) = H(X|Y)$, which is defined in the footnote of Page 2. In addition, $\boldsymbol{f}_n$ is a typo, it should be $\boldsymbol{t}_n$.**
>
> **In the preliminaries section, $\mathcal{D}$ denotes the dataset and $X$ denotes the random variable of  input samples.**
>
> 5) The paper is generally quite notation-heavy, which makes it difficult to read in some sections.
> **Could you please let us know which section is unclear to you? We appreciate your comments and would like to clarify your specific concern. Thank you!**
>
> 6) The experimental section in the main text is not extensive enough to be significant or convincing.
>
> **Could you please let us know which experimental section is not convincing? Based on your comments, we'd like to provide more explanations or implement extra experiments.** To demonstrate the theoretic explanations, we implement intensive experiments on both the synthetic dataset and benchmark dataset. Due to limited space, most experiments based on benchmark datasets are presented in Appendix.
>
> **A brief summary of the experiments in the paper is presented below.**
> - In Section 4.2, the probability space and the proposed mutual information estimator is validated on the synthetic dataset
>
> - In Section 4.3, the proposed Markov chains are validated by three networks on the synthetic dataset
>
> - In Appendix G.1, the limitation of the existing non-parametric models are presented
>
> - In Appendix G.2, the comparison between the proposed mutual information estimator and the existing non-parametric models are implemented
>
> - In Appendix H.1, the proposed Markov chains are validated by three networks on the MNIST dataset
>
> - In Appendix H.2, the proposed Markov chains are validated by three networks on the Fashion-MNIST dataset
>
> 7) The paper does not clearly state the limitations of the proposed method.
>
> **One limitation of the paper is that the probabilistic representation focuses on fully connected networks. However, since most previous works also study the information bottleneck principle and mutual information estimation in fully connected networks, we think it is okay for a theoretic paper. We will add the limitation discussion and refine the minor problems based on your comments in the final version.**
>
> 8) Figure 1 is difficult to interpret and does not add much to the main text of the paper.
>
> **Figure 1 explains the intuition of defining the random process of neural networks.**
>
> **It is known that a convolutional kernel defines a feature and the convolution output (feature map) measures the cross-correlation between the feature and input. Notice that if the receptive field of convolution is extended to the entire input then a convolutional layer becomes a fully connected layer, we generalize the explanation of convolution to fully connected layer and define the random process of a fully connected layer as recognizing the feature with the largest cross-correlation to the input. Given the random process, we define the probability space, specify the random variable and the distribution.**
>
> ---
>
> **Overall, thanks again for your comments. As another reviewer stated, this is a dense paper with lots of mathematical derivation, it could not be understood quickly. Please go through the response and let us know if you have more questions, we are ready to answer your questions. We hope you can reconsider the paper if the responses resolve your concerns. Thank you!**

---

> > ### Comment · Reviewer_e1nM · 2021-08-24
> > **Additional Comments**
> >
> > I thank the authors for their in-depth discussion and comments.
> >
> > First of all, I want to clarify that I agree with the authors in their statement that the definition of a probability space is the foundation for any information-theoretic analysis, but, unless strictly necessary, the definition of the probability space can be left implicit (as mentioned by reviewer hXwD) to shorten the notation. I am not entirely sure that the argument reported by the authors requires an explicit definition of the probability space.
> >
> > My main concern regards the statement that the variables $X \to T_1 \to T_2$ form a Markov chain.
> > As stated by the authors, $t_{2k} = \sigma_2(t_1, \omega_k)$ or, in other words, $t_2$ is expressed as a function of $t_1$. At the same time, $T_1$ and $T_2$ are discrete random variables representing which vectors ($\omega_n$ and $\omega_k$) correlate the most with $x$ and $t_1$ respectively.
> > As a result, I understand how $T_2$ is independent of $X$ when $t_1$ is given, but I do not see how $T_2$ is independent of $X$ when only $T_1$ is known.
> >
> > This property is used in the proof for Theorem 1, in which I find the following steps quite confusing:
> > 1) step 463: the authors expand $P_{T_2|T_1}(k|n)$, which shoud express a distribution over $k$ as a function of $n$, as a function of $t_1$ instead, while dropping the dependency on $n$. Note that $t_1$ does not appear in the conditioning of the aforementioned distribution (nor does $x$).
> > 2) step 466-467: the same can be found here. The proof claims that $P_{\hat{Y}|T_2}(l|k)$ is constant in $k$. This holds only when $t_2$ is observed. As a result, expression (32) removes  $P_{\hat{Y}|T_2}(l|k)$ from the summation on $k$, leaving $k$ undefined within the expression.
> >
> > I have the feeling that the proof is implicitly using $P(k|n, t_1)=P(k|t_1)$ and $P(l|k, t_2)=P(l|t_2)$ instead of $P(k|n)$ and $P(l|k)$ respectively. Can the authors please clarify this?
> >
> >
> > Regarding my comment on the experimental question, I want to clarify that, since the authors are suggesting a novel probabilistic formulation to characterize neural networks, I expected the experimental section in the main text to include the analysis of at least one of the commonly used datasets (MNIST, CIFAR10/CIFAR100) to connect to the existing work in literature.

---

> > > ### Author Response · Authors · 2021-08-25
> > > **Responses to the extra comments**
> > >
> > > We thank the reviewer for the extra comments. Our responses are posted below.
> > >
> > > **1. We are happy to see the reviewer agreeing that the definition of a probability space is the foundation for any information-theoretic analysis.**
> > >
> > > ***
> > >
> > > **2. Here want to explain a little bit why defining the probability space of a hidden layer is strictly necessary for information-theoretic analysis in deep learning because of two reasons.**
> > >
> > >
> > > 1) The estimation based on non-parametric models is commonly inaccurate in complex and high-dimensional dataset.
> > > Though non-parametric models, e.g., the empirical distribution [1] and the kernel density estimation [2], provides a simply practical approach to estimate the distribution of a random variable, and work well for simple and low dimensional dataset under the assumption that the samples of the random variable are i.i.d, we implement intensive experiments to show that non-parametric models are sensitive to hyper-parameters and cannot accurately estimate mutual information in DNNs in appendix G.1 and G.2, respectively. In addition, we show that activations do not satisfy the i.i.d. prerequisite of non-parametric models in appendix G.3, which explains why the estimation based on non-parametric models is not accurate in DNNs.
> > >
> > > 2) There are some theoretic problems of mutual information estimation in DNNs without specifying the random variable of a hidden layer, especially the infinite mutual information problem [2, 3] and inconsistent mutual information estimation in different networks [2]. However, a non-parametric model cannot solve the problems, because it merely estimate the distribution, but cannot specify the random variable.
> > >
> > > The flow chart below clearly shows that specifying the probability space provides a fundamental way to solve the above theoretic problems. In other words, it is strictly necessary to specify the probability space for improving the information theoretic interoperability of DNNs. In contrast, if the probability space is left implicit, it is hard to resolve the above theoretic problems.
> > > Following the logic showed in the flow chart, we finish this paper. To the best of our knowledge, this is the first work to improve the interoperability of DNNs based on probability space. We respectfully hope the reviewer can recognize the novelty of the paper.
> > >
> > > ***
> > >
> > > Resolve the theoretical limitations and validate the Information Bottleneck (IB) explanations for DNNs
> > >
> > > $\qquad$ $\uparrow$
> > >
> > > More accurate mutual information estimation: $I(X;T)$ and $I(Y;T)$
> > >
> > > $\qquad$ $\uparrow$
> > >
> > > **Distribution** $\quad$ $P(T)$: $\leftarrow$ **Non-parametric models**, e.g., the empirical distribution [1] and kernel density estimation [2]
> > >
> > > $\qquad$ **$\boldsymbol{\uparrow}$**
> > >
> > > **Random Variable** $\quad \text{ }$ $T: \Omega \rightarrow E$ $\quad$ $T(\omega_n) = n$
> > >
> > > $\qquad$ **$\boldsymbol{\uparrow}$**
> > >
> > > **Probability space** $\qquad$ $\Omega$ $\qquad$ $\qquad$ $\\{\omega_n\\}_{n=1}^N$
> > >
> > > $(\Omega,\mathcal{F},P):$ $\quad$ $\qquad$ $\quad$ $\mathcal{F}$ $\qquad$ $\qquad$ $\sigma$-algebra
> > >
> > > $\quad$ $\qquad$ $\qquad$ $\quad$ $\qquad \text{ }$ $P$ $\qquad$ $\qquad$ $P(\omega_n|x)=\frac{1}{Z}\exp[\sigma(<\omega_n, x>)]$
> > >
> > > [1] Ravid Shwartz-Ziv and Naftali Tishby. Opening the black box of deep neural networks via information. arXiv preprint arXiv:1703.00810, 2017.
> > >
> > > [2] Andrew Saxe, Yamini Bansal, Joel Dapello, Madhu Advani, Artemy Kolchinsky, Brendan Tracey, and David Cox. On the information bottleneck theory of deep learning. In International Conference on Representation Learning, 2018.
> > >
> > > [3] Rana Ali Amjad and Bernhard Claus Geiger. Learning representations for neural network-based classification using the information bottleneck principle. IEEE transactions on pattern analysis and machine intelligence, 2019.
> > >
> > > ***
> > >
> > > **3. The reviewer: 'I understand how T2 is independent of X when t1 is given, but I do not see how T2 is independent of X when only T1 is known'.**
> > >
> > > That is a very good and fundamental question! Thank you for asking.
> > >
> > > The key point is that the activations $\boldsymbol{t}_1$ is the **sufficient statistic** of the Gibbs distribution $P(T_1|X)$. Actually, the sufficient statistic is discussed in the 119th line of the paper, but we’d like to provide more explanations to the reviewer.
> > >
> > > The probability measure $P$ is defined as a Gibbs distribution (please refer Equation (5) in the paper for the detail of Gibbs distribution).
> > > Briefly speaking,
> > > \begin{equation}
> > > P_{T_1|X}(n|x) = \frac{1}{Z_{T_1}}\text{exp}(t_{1n})
> > > \end{equation}
> > > Gibbs distribution is one case of exponential family distributions (please refer Equation (8.1) in [4]), and an important property of Gibbs distribution is that the (negative) energy function $\boldsymbol{t_1}$  is the **sufficient statistic** of the Gibbs distribution, which means that if the activation vector $\boldsymbol{t_1}$ is known, then $P_{T_1|X}$ is known.
> > >
> > > In the context of information theory, if $X \rightarrow \boldsymbol{t_1} \leftrightarrow T_1 \rightarrow X_2$, then $\boldsymbol{t_1}$ is a sufficient statistic of $T_1$, namely $I(X;\boldsymbol{t_1}) = I(X; T_1)$ and $I(\boldsymbol{t_1};T_2) = I(T_1;T_2)$. It means that a sufficient statistics preserve mutual information, i.e., without information loss (Please refer Equation (2.124) Page 35-36 in [5] for the exact definition).
> > >
> > > It means that $\boldsymbol{t_1}$ and $T_1$ has the same information of the input $X$. Therefore, $T_2$ is independent on $X$ as long as we observe either $\boldsymbol{t_1}$ or $T_1$.
> > >
> > > Based on sufficient statistics, we can derive $P_{T_2|T_1}(k|n)$ in step 463 and $P_{\hat{Y}|T_2}(l|k)$ in step 466.
> > > Since $\boldsymbol{t_1}$ and $T_1$ has the same information, we can express $P(k|n, t_1) = P(k|n)$.
> > > BTW, we thank the reviewer carefully check the derivation!
> > >
> > > We hope that resolves the concern of the reviewer, and we will refine the paper to make it more clear in the final version. Again, thank you for asking this good question.
> > >
> > > In addition, using Gibbs distribution to model the activations will not lead to information loss. It further validates that a clear probability space would explain the random process (i.e., neural networks) more clearly.
> > >
> > > [4] The exponential family: Basics
> > >
> > > https://people.eecs.berkeley.edu/~jordan/courses/260-spring10/other-readings/chapter8.pdf
> > >
> > > [5] Thomas Cover and Joy Thomas. Elements of Information Theory. Wiley-Interscience, Hoboken, New Jersy, 2006.
> > > http://staff.ustc.edu.cn/~cgong821/Wiley.Interscience.Elements.of.Information.Theory.Jul.2006.eBook-DDU.pdf
> > >
> > >
> > > ***
> > >
> > > **4. For the experiment section, we’d like to follow the reviewer’s suggestion to move the experiments on benchmark dataset to the main paper.**
> > > Due to limited space, most experiments of complex networks on the MNIST and Fashion-MNIST are presented in Appendix H. We will follow the comment to refine the paper in the final version.
> > >
> > >
> > > ****
> > > Please let us know if you have more questions. Hope our responses address the concerns and help the reviewer to recognize the novelty of the paper as well. We hope the reviewer can reconsider the paper. Thank you!

---

> > > > ### Comment · Reviewer_e1nM · 2021-08-25
> > > > **Follow up discussion**
> > > >
> > > > I genuinely thank the authors for the clarifications and for the pointer to the experiments in the appendix. I strongly recommend moving this part of the appendix in the main text.
> > > >
> > > > Regarding their comment on **3**, I do agree that $\bf t_1$ is a sufficient statistic for $T_1$, what I don't understand is why the Markov chain $T_1\to{\bf t_1}\to T_2$ holds.
> > > >
> > > > I do not agree with the claims that $\bf t_1$ and $T_1$ have the same information about the input $X$ or $T_2$, so here I report a simple counter-example.
> > > >
> > > > Consider a discrete 2-dimensional $X$ that carries at least 2-bits of information  (e.g. the outcome of a 2-dice roll: $H(X)\ge 2$) and an invertible first layer (e.g. identity weight matrix and sigmoid activation), then:
> > > > 1) The mutual information between $X$ and $\bf t_1$ is maximal ($I(X;{\bf t_1})=H(X)\ge 2$) since the first layer is invertible ( $H(X|{\bf t_1})=0$).
> > > >
> > > > 2) The variable $T_1$ is effectively a Bernoulli random variable (because of the 2-dimensional input) and it can carry at most one bit of information: $H(T_1)\le 1$.
> > > >
> > > > 3) For discrete random variables, the entropy is an upper bound to the amount of mutual information: $I(T_1; X)\le H(T_1)$
> > > >
> > > > Using these 3 points we can show:
> > > > - $I(X;{\bf t_1})=H(X)\ge 2$
> > > > - $I(X;T_1)\le H(T_1) \le 1$
> > > >
> > > > Therefore we can conclude that $I(X;{\bf t_1})> I(X;T_1)$ (without equality).
> > > >
> > > > In general, I think $T_1$ carries much less information than $\bf t_1$ regarding $X$, and the fact that $\bf t_1$ is a sufficient statistic for $T_1$ only implies the Markow chain $X\to {\bf t_1}\to T_1$.
> > > >
> > > > Can the authors please clarify why their claims $I(X;T_1)=I(X;{\bf t_1})$ and $I(T_2;T_1) =  I(T_2;{\bf t_1})$ would hold?

---

> > > > > ### Author Response · Authors · 2021-08-26
> > > > > **Responses to the follow up discussion**
> > > > >
> > > > >
> > > > > **We are happy to see the reviewer recognizing our clarifications and the experiments summary in the appendix. We will absolutely follow the reviewer’s suggestion to reformat the paper, especially the experiment section. Thank the reviewer again for the helpful suggestions.**
> > > > >
> > > > > **In addition, we are happy to see the reviewer agreeing that the activation $\boldsymbol{t1}$ is a sufficient statistic of $T_1$, if the distribution of a hidden layer is the Gibbs defined by Equation (5) in the paper. The agreement will be helpful for the subsequent discussion**
> > > > >
> > > > > ---
> > > > >
> > > > > Thank the reviewer for providing a very interesting counter-example. We summarize four points below to understand the example.
> > > > >
> > > > > - The reviewer defines a simple network, the input layer has 2 discrete nodes, and the range of each node is 1 to 6 (corresponding to the 6 outcomes of a dice roll). A single hidden layer has 2 neurons, and the weights of each neuron are [1,0] and [0,1], respectively. The activation functions in the layer are sigmoid.
> > > > >
> > > > > - The reviewer sets the weights being identity matrix and the activation function being the invertible sigmoid, thus the distribution of $\boldsymbol{t}_1$ is identical to the distribution of $X$, thereby $I(X;\boldsymbol{t}_1) =H(X) > 2$.
> > > > >
> > > > > - Based on the probability space definition in the paper, the sample space is consistent of two features [1,0] and [0,1], and the random variable  $T_1$ of the hidden layer is binary, thus the Gibbs distribution becomes a Bernoulli distribution.
> > > > >
> > > > > - Based on the second point, $I(X;\boldsymbol{t}_1) = H(X)$ is derived
> > > > >
> > > > > $\quad$ Based on the third point, $I(X;T_1) < H(T_1)$ is derived.
> > > > >
> > > > > $\quad$ Finally, $I(X;\boldsymbol{t}_1) > I(X;T_1)$ is derived.
> > > > >
> > > > > If the four points are not consistent to the reviewer, please the reviewer correct us.
> > > > >
> > > > > If the four points are consistent, let us continue the discussion to clarify our opinions and answer the reviewer's questions.
> > > > >
> > > > > 1. **We totally agree the conclusion of the example with the reviewer.** Based on our understanding, an important assumption of the example is that the ground truth of the distribution of $\boldsymbol{t_1}$ is known and identical to $X$. Let $T_{GT}$ denotes the ground truth, the assumption means $T_{GT} = X$ and $I(X;T_{GT}) = H(X)$, thus we can derive $I(X;\boldsymbol{t_1}) = H(X) > I(X;T_1)$, where $\boldsymbol{t_1}$ can be broadly viewed as some samples of $T_{GT}$, namely $\boldsymbol{t_1} \sim T_{GT}$. In addition, here $T_1$ denotes the random variable defined based on someone's understanding of the functionality of the layer.
> > > > >
> > > > > 2. The example and the assumption could be a little bit beyond the scope of deep learning, in which the ground truth random variable $T_{GT}$ and the ground truth distribution  $P(T_{GT})$ of a hidden layer are commonly unknown, but only some samples/activations $\boldsymbol{t_1}$ are known given some inputs $x \sim X$. Probability space modeling aims to define a random variable $T_1$ to simulate $T_{GT}$ as accurate as possible for explaining the functionality of the hidden layer. **Therefore, we form a Markov chain $X \rightarrow T_{GT} \rightarrow  \boldsymbol{t_1} \rightarrow T_1$, and still can derive $I(X;\boldsymbol{t_1}) \geq I(X;T_1)$, which is the same conclusion in the example. The key point of the paper is to clearly explain the functionality of the hidden layer and the activations $\boldsymbol{t_1}$ for defining $T_1$ to approximate $T_{GT}$ as accurate as possible.**
> > > > >
> > > > > 3. Let's back to specific questions proposed by the reviewer. We try to use three points to explain that if $\boldsymbol{t}_1$ is a sufficient statistics for $T_1$, then $I(X;\boldsymbol{t}_1) = I(X;T_1)$.
> > > > >
> > > > > * First, the exact definition of sufficient statistics, i.e., a statistics $\boldsymbol{t}_1$ is called sufficient for $T_1$ if it contains all information of $T_1$ in $X$ (please refer the last two sentences of P35 in [5]).
> > > > >
> > > > > * Second, the theoretic derivation of sufficient statistics. For any statistics $t_1$ for $T_1$, the Markov chain $X \rightarrow T_1 \rightarrow t_1$ is valid, which derives $I(X;T_1) \geq I(X;t_1)$. We kindly remind the reviewer that you could overlook the Markov chain for arbitrary statistics (please refer Equation (2.123) in P35 of [5] for detail). As the reviewer stated above the Markov chain  $X \rightarrow t_1 \rightarrow T_1$ is valid for the sufficient statistics $t_1$ for $T_1$, which derives $I(X;t_1) \geq I(X;T_1)$. Considering the two inequalities, we can finally derive $I(X;\boldsymbol{t}_1) = I(X;T_1)$.
> > > > >
> > > > > * Third, a simple example to explain that the activations is the sufficient statistics for a Gibbs distribution of a hidden layer. For example, a hidden layer with 3 neurons $\\{\omega_1, \omega_2, \omega_3\\}$ would output 3 activations, e.g, $\\{\frac{2}{3}, 1, \frac{4}{3}\\}$. Based on the cross-correlation definition, each activation measures the cross-correlation between input $x$ and each feature $\omega_i$, e.g., $\frac{2}{3}$ means the cross-correlation between $x$ and $\omega_1$. Based on the Gibbs distribution, we have $P(\omega_1|x) = 0.23$, $P(\omega_2|x) = 0.32$, and $P(\omega_3|x) = 0.45$. Since the random process of a hidden layer is defined to recognize one of the features with the largest cross-correlation to input (Please refer Line 92-93 in the paper), $P(T = 1|x) = 0.23$ and $P(T=3|x) = 0.45$ mean that $\omega_1$ and $\omega_3$ have $23\\%$ and $45\\%$ to be regarded as the feature with the largest cross-correlation to $x$, respectively. We can observe that the probabilities are consistent with the cross-correlation between features and input. Specifically, if we know the activations, we can derive the probability; and if we know a feature has high probability, then we know the feature must have high activation(cross-correlation).
> > > > >
> > > > > [5] Thomas Cover and Joy Thomas. Elements of Information Theory. Wiley-Interscience, Hoboken, New Jersy, 2006. http://staff.ustc.edu.cn/~cgong821/Wiley.Interscience.Elements.of.Information.Theory.Jul.2006.eBook-DDU.pdf
> > > > >
> > > > > *****
> > > > >
> > > > > Thanks the reviewer again for the valuable comments! Please let us know if you have more questions, and hope you can reconsider the paper.

---

> > > > > > ### Comment · Reviewer_e1nM · 2021-08-26
> > > > > > **Thank you for the clarification**
> > > > > >
> > > > > > I confirm that the summary made by the authors of the counterexample is consistent with my previous comment and I thank them for their clarifications. Nevertheless, I still do not understand how $\bf t_1$ can be written as a statistic of $T_1$. Now I see how, for a sufficiently large set of samples from $T_1$, the relative frequency represents an estimation of the normalized activation, but how does one recover the normalization constant $Z_T$?
> > > > > >
> > > > > > I understand that the assumption that the layer of a neural network is invertible can feel a bit farfetched in the context of deep learning, but this example was designed to numerically underline the problem. In general, I believe that a high-dimensional activation vector $\bf t_1$  of size $n$ contains much more information when compared to the corresponding Gibbs distribution $T_1$ (which contains at most $\log n$ bits). This makes me question how the proposed analysis relates to the original Information Bottleneck formulation.

---

> > > > > > > ### Author Response · Authors · 2021-08-27
> > > > > > > **Follow up discussions**
> > > > > > >
> > > > > > >
> > > > > > > Thank the reviewer for confirming our summary and proposing follow up questions. We are happy to see that the reviewer somehow understand the statistical relation between activations $\boldsymbol{t_1}$ and the Gibbs distribution $P(T_1|X)$.
> > > > > > >
> > > > > > > In this comment, we will first discuss the simple example proposed by the reviewer, and then address specific questions from the reviewer.
> > > > > > >
> > > > > > > ***
> > > > > > >
> > > > > > > **1. Recapping the relation between $I(X;\boldsymbol{t}_1)$ and $I(X;T_1)$**
> > > > > > >
> > > > > > > The input signal is the outcomes of 2 dice-roll, so there are totally 6*6 = 36 cases. In the simple network, the input has two discrete node and the range of node value is from 1 to 6, and the hidden layer has two nodes with sigmoid function.
> > > > > > >
> > > > > > > Based on some simplifications, i.e.,  the weights of the hidden layer as identify matrix and the sigmoid function is invertible, the distribution of the hidden layer will be the same as the input, namely $T_{GT} = X$, where $T_{GT}$ and $X$ denote the ground truth random variable of the hidden layer and the input.
> > > > > > >
> > > > > > > As a result, one can derive $I(X;T_{GT}) = I(X;\boldsymbol{t_1}) = H(X) = \log_2(36)$.
> > > > > > > Since the hidden layer only has 2 nodes, the random variable of the hidden layer is binary based on the proposed probability space in the paper, thereby $I(X;T_1) \leq 1$. **We agree with reviewer that $I(X;\boldsymbol{t}) > I(X;T_1)$ in the example**.
> > > > > > >
> > > > > > > Moreover, we can further generalize the example to most practical deep learning cases, where the weights are set as variable, the input distribution is unknown, and activation functions being arbitrary, **we can still derive $H(X) \geq I(X;\boldsymbol{t}) \geq I(X;T_1)$ in general cases based on the Markov chain $X \rightarrow T_{GT} \rightarrow \boldsymbol{t_1} \rightarrow T_1$.**
> > > > > > >
> > > > > > > ***
> > > > > > >
> > > > > > > **2. The mutual information $I(X;T_1)$ based on the probability space can accurately quantify the information learned by $\boldsymbol{t_1}$ and is more explainable.**
> > > > > > >
> > > > > > > We not fully understand why the reviewer believes **'a high-dimensional activation vector $\boldsymbol{t_1}$ of size $n$ contains much more information when compared to the corresponding Gibbs distribution $T_1$'.**
> > > > > > >
> > > > > > > Given the architecture of networks, we think that accurately estimating the information of $X$ learned by $\boldsymbol{t_1}$ (namely $I(X;\boldsymbol{t_1})$) is determined by how to probabilistic model $\boldsymbol{t_1}$, specify $T_1$, and derive $P(T_1)$.
> > > > > > > For different ways of probabilistic modeling the 2-dimensional values $\boldsymbol{t_1}$, $I(X;T_1)$ being very high or very low are possible. Especially, an extreme case (problem) is $I(X;T_1) = \infty$ (please refer Appendix C in [1]), which is trivial in real-world cases.
> > > > > > >
> > > > > > > [1] Andrew Saxe, Yamini Bansal, Joel Dapello, Madhu Advani, Artemy Kolchinsky, Brendan Tracey, and David Cox. On the information bottleneck theory of deep learning. In International Conference on Representation Learning, 2018.
> > > > > > >
> > > > > > > We think the example proposed by the reviewer is very interesting and helpful to examine the proposed theoretic explanations.
> > > > > > > Following the exactly same logic as the reviewer, we designed the synthetic dataset (samples with 2 bits information and labels with 1 bit information) to examine the non-parametric models. Figure 8 in Appendix G.1 shows that empirical distributions estimate $I(X; T_1) > 5$ when the hyper-parameter $bs \leq 0.1$, and Figure 10 in Appendix G.1 shows that the Kernel Density Estimation (KDE) estimates  $I(X; T_1) > 5$  when the hyper-parameter $\sigma_n^2 \leq 0.1$.
> > > > > > > It means that even  $I(X; T_1)$ estimation is very high, it still could be inaccurate.
> > > > > > >
> > > > > > > Therefore, we argue that the criteria of a good mutual information estimator are explainable and accurate, i.e., we must have very clear definitions for the random variable $T_1$ and the distribution $P(T_1)$ to explain the functionality of the hidden layer and the activations $\boldsymbol{t_1}$, rather than using non-parametric models to indirectly estimate $P(T_1)$, for mutual information estimation.
> > > > > > > In that sense, we think Gibbs distribution is better than non-parametric models. For example, Figure 12 in Appendix G.2 shows that Gibbs distribution estimates $I(X;\hat{Y}) = 1$, whereas non-parametric models estimate $I(X;\hat{Y}) > 1.5$. Though Gibbs distribution estimation is lower than non-parametric estimations, Gibbs derives more accurate estimation than non-parametric models, because the output layer $\hat{Y}$ only has 2 nodes, namely the maximal mutual information $I(X;\hat{Y}) = 1$.
> > > > > > >
> > > > > > > We thanks the reviewer again for proposing the interesting example, but we could not fully understand the key point of  **'a high-dimensional activation vector $\boldsymbol{t_1}$ of size $n$ contains much more information when compared to the corresponding Gibbs distribution $T_1$'.** Please correct us, if you find something wrong.
> > > > > > >
> > > > > > >
> > > > > > > ***
> > > > > > > **3 Information Bottleneck (IB) principle** aims to optimize a random variable $T$, such that it can compress the information of input $X$ while preserve the information of label $Y$. Therefore, it formulates a Lagrangian (Please refer Equation (1) for detail)
> > > > > > >
> > > > > > > $T^* = \\argmin_{P(T|X) \quad I(X;T)-\\beta I(Y;T)}$
> > > > > > >
> > > > > > > In terms of the example proposed by the reviewer, $I(X;T_1)$ can learn at most one bit information from input, thus $H(X) > I(X;T_1)$ corresponds to the compression in IB principle. If the simple network is designed to predict which dice has the larger value, it will be a binary problem. The network would satisfy the IB principle if $T_1$ preserve the information labels, i.e., $I(Y;T_1)$ close to $H(Y) = 1$ bit,. That is a simple explanation for the IB principle and networks.
> > > > > > >
> > > > > > > Many recent works use the IB principle to explain DNNs, and the key of IB principle is to accurately estimate $I(X;T)$ and $I(Y;T)$. However, non-parametric models deriving inaccurate mutual information estimation and some information theoretic limitations, e.g., the infinite mutual information, weakens the validity of the IB explanations for DNNs. In this paper, we propose the probability space for deriving accurate mutual information estimation and resolving the information theoretic limitations, and finally confirm the IB principle in DNNs.
> > > > > > >
> > > > > > > ***
> > > > > > >
> > > > > > > **4. How does one recover the normalization constant $Z_T$?**
> > > > > > >
> > > > > > > That is a good question. Thank you for asking!
> > > > > > >
> > > > > > > Based on the Gibbs distribution defined in Equation (5) in the paper
> > > > > > >
> > > > > > > $P(T=n|X=x) = \frac{1}{Z_T}\exp(<\omega_n, x>)$
> > > > > > > where $Z_T = \sum_{n=1}^N\exp(<\omega_n, x>)$.
> > > > > > >
> > > > > > > It measures the probability of the feature $\omega_n$ being recognized the feature with the largest cross-correlation to the input $x$.
> > > > > > > We can understand that the N features $\\{\omega_n\\}_{n=1}^N$ form a feature space of $x$, a hidden layer aims to recognize which features can represent the input accurately, i.e., it aims to choose the features with the largest correlation the the input. Therefore, the key to recover $Z_T$ is learning the feature space $\\{\omega_1,\omega_2,\omega_3\\}$ accurately. Since the features are defined by the weights of neurons,  $Z_T$ can be recovered as long as a sufficiently large set of samples are available for training the layer.
> > > > > > >
> > > > > > > For example, a hidden layer has 3 neurons and formulates 3 features $\\{\omega_1,\omega_2,\omega_3\\}$. As long as the layer is well trained by sufficiently large samples, $\\{\omega_1,\omega_2,\omega_3\\}$ can form an accurate feature space of $x$, thus $Z_T = \sum_{n=1}^N\exp(<\omega_n, x>)$ would be accurate, and the Gibbs distribution would generate accurate probability.
> > > > > > >
> > > > > > > ****
> > > > > > > Thank the reviewer again for the follow up questions. Hope the comments address the reviewer's concern. Please let us know if you have more questions. Hope the reviewer can reconsider the paper. Thank you!

---

> > > > > ### Author Response · Authors · 2021-08-30
> > > > > **More clarifications to the reviewer**
> > > > >
> > > > >
> > > > > We genuinely appreciate the reviewer for the previous in-depth discussions. In this comment, we provide more clarifications and hope it can help the reviewer to understand the paper.
> > > > >
> > > > > ---
> > > > >
> > > > > **1. The maximal information (namely $\log_2n$ bits) of a fully connected layer with N neurons is determined by the cross-correlation definition of convolution.**
> > > > >
> > > > > We guess the reviewer's concern ('a high-dimensional activation vector
> > > > >  of size $n$ contains much more information when compared to the corresponding Gibbs distribution
> > > > >  (which contains at most $\log_2(n)$ bits)') is that the proposed probabilistic modeling could not fully describe the information of  a high-dimensional activation vector
> > > > >  of size $n$.
> > > > >
> > > > > **We totally agree with the reviewer if without any explanation (prior knowledge) of the $N$-dimensional activations and the functionality of a hidden layer.** In other words, if we only study a free $N$-dimensional vector, the information of which could be very large. However, as we discussed in the paper L81-97, given a fully connected layer with $N$ neurons, the $N$-dimensional activations measure the cross-correlations between the input $\boldsymbol{z}$ and the $N$ features defined by the weights of $N$ neurons.
> > > > > Therefore, the $N$-dimensional activations have specific meaning in the context of deep learning, rather than a free $N$-dimensional vector.
> > > > >
> > > > > Based on the data processing inequality and the activation definition $t_n = \sigma(<\omega_n, \boldsymbol{z}>)$, the information of the $N$-dimensional activations $\\{t_n\\}_{n=1}^N$ is no more than the information of the $N$ features $\\{\omega_1, \cdots, \omega_N\\}$. Since the $N$ features have at most $\log_2(n)$ bits information, the $N$-dimensional activations have at most $\log_2(n)$ bits information as well.
> > > > > For example, Figure 1 in the paper visualizes the  $4 \times 4$  dimensional input $\boldsymbol{z}$ and two $4 \times 4$  features $\\{\omega_1, \omega_2\\}$. The two activations $t_1$ and $t_2$ measures the cross-correlation between $\boldsymbol{z}$ and $\\{\omega_1, \omega_2\\}$. Therefore, the two activations $t_1$ and $t_2$ have at most $\log_2(2) = 1$ bit information of $\boldsymbol{z}$.
> > > > >
> > > > > Overall, since the $N$-dimensional activations of a hidden layer with $N$ neurons is explained to the cross-correlation between input and the feature defined by the weights, the $N$-dimensional activations have at most $\log_2(n)$ bits information, which could be lower than the information capacity of a free $N$-dimensional vector.
> > > > > **Notably, since the cross-correlation definition clearly explains the functionality of a fully connected layer, we do think the proposed probabilistic modeling can fully describe the information of  $N$-dimensional activations.**
> > > > >
> > > > > **In contrast, non-parametric models simply regard the $N$-dimensional activations as a free $N$-dimensional vector without considering any functionality of the hidden layer, which explains why non-parametric models, e.g., empirical distributions, cannot measure the mutual information as accurately as the proposed Gibbs distribution to some extend.**
> > > > >
> > > > > We hope the reviewer to reviewer the paper L81-97 and Figure 1, which could address the concern of the reviewer.
> > > > >
> > > > > ---
> > > > >
> > > > > **2. More explanations for the information bottleneck principle in DNNs based on the example.**
> > > > >
> > > > > Here we want to explain the information bottleneck principle in DNNs a little bit more based on the example proposed by the reviewer.
> > > > > In a word, many previous works argue that DNNs satisfy the information bottleneck principle, i.e., DNNs is trained to learn the information relevant to a task, e.g., classification, while compress irrelevant information of input.
> > > > >
> > > > > In the example proposed by the reviewer, the input is 2 dice-roll, thus it evenly has totally 6*6 = 36 cases, i.e., the input has $\log_2(36)$ bits information.
> > > > > Based on the cross-correlation definition of a fully connected layer $t$, it must has 36 neurons to represent all the information of the 36 cases. However, if we take into account the target of the network (e.g., the task is to classify which dice value is larger, the task requires only 1 bit information of input), the hidden layer with 2 neurons is enough to finish the task. In real applications, though networks are commonly complicated, i.e., the number of neurons is much more than the required to finish a task, many previous works argue that DNNs still satisfy the information bottleneck principle, and attempt to measure the mutual information between training samples/labels and each hidden layer to figure out how DNNs achieve the information bottleneck principle.
> > > > >
> > > > > We can observe that mutual information estimation is the key to study the information bottleneck principle in DNNs. However, simple mutual information estimators without specifying the random variable $T$ and the distribution $P(T)$ results in some limitations, e.g., the infinite mutual information and inaccurate mutual information estimation based on classical non-parametric models.
> > > > > Therefore, the paper aims to address the limitations thorough specifying the probability space of a hidden layer.
> > > > >
> > > > > ---
> > > > >
> > > > > **3. More clarifications about the main idea of the paper based the example**
> > > > >
> > > > > Following the example proposed by the reviewer, we consider a simple network with the input layer $x$ and one layer $t = f(x)$. The table below summarizes the ground truth random variables $X_{\text{GT}}$ and $T_{\text{GT}}$, and the hypothesis random variables $X$ and $T$.
> > > > >
> > > > > $\qquad \qquad \qquad  \quad$ Input  $\qquad$ A hidden layer
> > > > >
> > > > > ---
> > > > >
> > > > > Random variable $\quad$ $X_{\text{GT}}$    $\qquad \qquad$          $T_{\text{GT}}$
> > > > >
> > > > > (ground truth) $\quad$ **Unknown**    $\qquad \quad$          **Unknown**
> > > > >
> > > > > ---
> > > > >
> > > > > Samples     $\qquad$  $\qquad$ $x$   $\qquad \qquad$          $t = f(x)$
> > > > >
> > > > >  $\qquad$ $\qquad$ $\qquad$  $\quad$ Known    $\quad \qquad$          Known
> > > > >
> > > > > ---
> > > > >
> > > > > Modeling $\qquad$ $\quad$ **$x \sim i.i.d.$**     $\quad \quad$          $T$ and $P(T)$
> > > > >
> > > > >
> > > > > ---
> > > > >
> > > > >
> > > > > Estimation $\qquad$ $\quad$ $\qquad$ $\qquad$   $\quad \quad$          $I(X;T)$
> > > > >
> > > > >
> > > > > ---
> > > > >
> > > > > **Since $X_{\text{GT}}$ and $T_{\text{GT}}$ are unknown in most real cases, $I(X;T)$ could be equivalent to, higher, or lower than $I(X_{\text{GT}};T_{\text{GT}})$ depending on how to model $x$ and $t$, namely how to specify $X$, $T$ and $P(T)$.**
> > > > >
> > > > > A synthetic dataset means $X_{\text{GT}}$ being known, thus designing synthetic dataset is a good approach to simplify the problem (namely $I(X;T) = I(X_{\text{GT}};T)$ and examine any proposed modeling methods.
> > > > > The advantage of synthetic dataset is using $I(X_{\text{GT}}; T) \leq H(X_{\text{GT}})$ to **partially** examine any proposed mutual information estimator. Specifically, if $I(X_{\text{GT}}; T) > H(X_{\text{GT}})$, there must be some errors in the proposed mutual information estimator. However, if $I(X_{\text{GT}}; T) \leq H(X_{\text{GT}})$, one still cannot determine if the mutual information is accurate or not.
> > > > > Specifically, since $T_{\text{GT}}$ is still unknown, $I(X_{\text{GT}};T)$ still could be equivalent to, higher, or lower than $I(X_{\text{GT}};T_{\text{GT}})$ depending on how much knowledge of the layer one has, i.e., how to specify $T$ and $P(T)$ to approximate $T_{\text{GT}}$.
> > > > >
> > > > > Therefore, to accurately estimate the the ground truth mutual information $I(X_{\text{GT}};T_{\text{GT}})$ in deep learning, one must have a clear understanding of the functionality of a hidden layer to specify $T$ and $P(T)$ for making $T$ to approximate $T_{\text{GT}}$ as close as possible.  That is why the paper spends much effort in defining the probability space of a hidden layer.
> > > > >
> > > > > ---
> > > > >
> > > > > Finally, thanks the reviewer again for the in-depth discussions! Please let us know if you have more questions, and hope you can reconsider the paper if you are clear to the paper.

---

> > > > > > ### Author Response · Authors · 2021-09-02
> > > > > > **Any more questions?**
> > > > > >
> > > > > >
> > > > > > We thank the reviewer again for the in-depth discussion and paper reviewing.
> > > > > >
> > > > > > Based on the previous discussions, we feel that the reviewer has already understood the main contributions of the paper.
> > > > > > In a word, the probability space definition for a hidden layer enables us to specify the random variable and the distribution of a hidden layer, thereby deriving accurate mutual information for resolving the existing limitations of non-parametric models and further confirm the information bottleneck principle. To the best of our knowledge, this is the first work for improving the information theoretic interpretability of DNNs via establishing a solid probabilistic foundation.
> > > > > >
> > > > > > In addition, we think the previous discussions have already resolved most concerns of the reviewer. Especially, the latest comment [(more clarifications to the reviewer)](https://openreview.net/forum?id=6ugK-RQhIP5&noteId=A6tuWpY8OYb) answers the last question of the reviewer (i.e., why a fully connected layer with $n$ neurons only has at most $\log_2n$ bits information? that is because the n-dimensional activations has specific meaning in the cross-correlation definition of the layer, rather than being a free n-dimensional vector.)
> > > > > >
> > > > > > Since September 2 is the last day of paper discussion phase. We respectfully ask the reviewer. Is there any other questions that we can explain or clarify for you? Please feel free let us know. If you are clear for the paper and acknowledge the contributions of the paper, could you please give us some new feedback and consider boosting your rate of the paper? I understand you must be very busy for the reviewing task now, but I sincerely hope you can give us more feedback. Thank you! Really appreciate it!

---

> > > > > > > ### Comment · Reviewer_e1nM · 2021-09-02
> > > > > > > **Final comments**
> > > > > > >
> > > > > > > Once again, I thank the authors for their in-depth answers and clarifications.
> > > > > > >
> > > > > > > Using the 2-dice example,  I want to mention some critical aspects that are still unclear.
> > > > > > > The authors mention that, in order to cover all the 36 cases using cross-correlation, the hidden layer must have at least 36 hidden units.
> > > > > > > Nevertheless, consider a linear layer composed of the weights $\omega=[1, 6]$ and bias ${\bf b}=[0,6]$ (for dice outcomes $\in(0,1,..,5)$).
> > > > > > > Even if the linear layer output is 1-dimensional $t_1\in(0,.., 35)$, the information regarding the two dice can be fully recovered and used by the subsequent layers of the neural network.
> > > > > > >
> > > > > > > On the other hand, the variable $T_1$ gives little to no information regarding what the neural network is doing or how much information regarding input is kept. I think the relation between $T_1$ and $\bf t_1$, which is crucial to connect this paper to the literature on the Information Bottleneck principle, is not sufficiently explored and clarified in the main text.
> > > > > > >
> > > > > > > Overall I think the paper explores an interesting and original direction, and I genuinely thank the authors for the insightful discussion, but I believe the paper is not currently ready for publication, therefore I decided to keep my current score.

---

> > > > > > > > ### Author Response · Authors · 2021-09-02
> > > > > > > > **Thank you for the final comments**
> > > > > > > >
> > > > > > > >
> > > > > > > > We thank the reviewer for much time and effort on reviewing the paper.  Though we still hope the reviewer can reconsider the paper, the comment mainly gives some new ideas to the reviewer.
> > > > > > > >
> > > > > > > > ---
> > > > > > > >
> > > > > > > > 1. We agree with the reviewer for the 2-dice example. However, the prerequisite of  designing a linear function to model the 2-dice example is the random process being known, i.e., the distribution of input is known. However, **in most real cases, the distribution of input is unknown**. Therefore, one should pay more attention in probabilistic modeling how neural networks extract useful information from high-dimensional dataset with unknown distributions.
> > > > > > > > The underlying question is: if the only known information of input is 2 dimensional, how neural networks learn information from the dataset? At least, the proposed probability space clearly explain how neural network do it and further confirm the information bottleneck principle.
> > > > > > > >
> > > > > > > > 2. Though a hidden layer needs 36 nodes to model the all the information of 2-dice example based on the proposed probability space definition, the number of required nodes would be much smaller if taking into account the tasks. As we mentioned before, if the task of a network is to classify which dice is larger, the entire network only needs 2 nodes to learn 1 bit of information from the input to finish the task. That is strongly related to the information bottleneck principle.
> > > > > > > >
> > > > > > > > 3. I agree with the reviewer that the relation between $T_1$ and $\boldsymbol{t}_1$ is crucial. Compared to most previous works regarding $\boldsymbol{t}_1$ as free $N$-dimensional vector, the paper spends much effort to defining the random process and the probability space of a hidden layer, and demonstrate that the $N$-dimensional activations quantify the cross-correlations between the input and $N$ features defined by the $N$ neurons.
> > > > > > > >
> > > > > > > > Finally, thanks the reviewer again for your time and effort for reviewing the paper!

---

### Official Review · Reviewer_3nVv · 2021-07-18

**Rating:** 8
**Confidence:** 4

**Summary:**

The paper points out that the IB principle based on the Markov chain assumption ($Y \rightarrow X \rightarrow T_1 \rightarrow \cdots \rightarrow \hat{Y}$) cannot fully characterize the information flow in DNNs, and hence several claims previously made on the assumption may not be accurate. For better quantification of the mutual information, this paper proposes a probabilistic framework of IB in deep neural networks (DNNs) by taking into account the back-propagation training. Under the probabilistic framework, the paper derives two Markov chains that characterize the information flow in DNNs and provides experimental evidence for supporting the validity of the Markov chain assumptions. The paper claims that the probabilistic approaches allow us to more accurately estimate the mutual information than existing non-parametric approaches and experimentally demonstrates how different activation functions and hidden layers achieve IB trade-offs.


**Limitations And Societal Impact:**

The authors didn't address the limitations of their work in the conclusion section. The scope of claims should be specified clearly.
- Please specify the assumptions made for the probabilistic framework.
- Please specify claims that are only supported by evidence.

**Main Review:**

This paper provides a novel probabilistic framework for understanding IB principles and estimating the mutual information in DNN. The probabilistic view provided by the authors is very interesting and convincing. The claims are well supported by theoretical and experimental evidence. The only concern is on the clarity. This is a dense paper with a lot of mathematical proofs and experimental results, so improved clarity will help readers to understand the idea and findings of this paper as well as its limitations. See details below.

Major
- L63-66: Please specify clearly which claims are theoretically or experimentally supported in the introduction. Are those claims "different hidden layers achieve different IB trade-offs", and "MLP satisfies IB principle no matter what the architecture of the MLP is") supported by theoretical or experimental evidence?
- L81-87: It is not clear why this paragraph (generalizing CNN to MLP) is needed to explain that "the weights of a neuron can be viewed as a global feature, ... ."  If this is a rationale for the probabilistic representation, please consider re-write it.
- L129-130: Please provide a detailed rationale for Definition 2. I'm guessing this is because of the infinite mutual information problem. How can we use this definition without loss of generality?

Minor
- L45: Please also provide the subsection number. (i.e. Appendix G -> Appendix G.3)
- L63: What is \overline{X}?
- L103: Please provide the definition of $f_n$.
- L104: $n$th -> $n$-th
- L186, L228: Be consistent in referring equations (i.e. Use only one style between "Equation (2)" and "Equation 2" )

**Time Spent Reviewing:**

5

---

> ### Author Response · Authors · 2021-08-05
> **Author Response**
>
> Thank you for the positive and encouraging comments on the paper. We address your concerns below.
>
> (1) L63-66: Please specify clearly which claims are theoretically or experimentally supported in the introduction.
>
> **The two Markov chains in L63 is theoretically supported by Corollary 1 and Theorem 2.**
>
> (2) Are those claims "different hidden layers achieve different IB trade-offs", and "MLP satisfies IB principle no matter what the architecture of the MLP is") supported by theoretical or experimental evidence?
>
> **Yes, the two claims are experimentally demonstrated in Figure 5. The paragraph in L289-L295 derives the first claim. The paragraph in L296-301 derives the second claim.**
>
> (3) L81-87: It is not clear why this paragraph (generalizing CNN to MLP) is needed to explain that "the weights of a neuron can be viewed as a global feature, ... ." If this is a rationale for the probabilistic representation, please consider re-write it.
>
> **We think the cross-correlation definition of convolution is easily understandable for most people, thus we use the paragraph L81-87 to induce the readers to understand that "the weights of a neuron can be viewed as a feature". The paragraph works as an intuition, rather than the rationale for probabilistic representation. However, we will take into account the comment and refine the paper in the final version.**
>
> (4) L129-130: Please provide a detailed rationale for Definition 2. I'm guessing this is because of the infinite mutual information problem. How can we use this definition without loss of generality?
>
> **You are right, an important reason for Definition 2 is to avoid the infinite mutual information problem. Based on the explicit definition of probability space, Definition 2 specify the random variable of a fully connected hidden layer  being discrete, thereby circumventing the infinite mutual information problem. Moreover, we use Definition 2 to derive the probabilistic explanation for the entire architecture of MLP, namely Theorem 1.**
>
> **Definition 2 is based on the probability space of a fully connected layer. In other words, we can use Definition 2 as long as we study fully connected layers.**
>
> (5) L63: What is \overline{X}?
>
> **$\bar{X}$ is a virtual random variable has all the information of X except Y, namely $H(\bar{X}) = H(X|Y)$, which is defined in the footnote of page 2.**
>
>
> (6) L103: Please provide the definition of $\boldsymbol{f}_n$.
>
> **This is a typo. It should be $\boldsymbol{t}_n$. We will correct all the minor problems in the final version based on your comments. Thank you!**
>
> (7) The authors didn't address the limitations of their work in the conclusion section. Please specify the assumptions made for the probabilistic framework.
>
> **The limitation of the paper is that the probabilistic representation focuses on fully connected networks. However, since most previous works also study the information bottleneck principle and mutual information estimation in fully connected networks, we think it is okay for a theoretic paper.  We specify the assumption and scope of the probabilistic framework in the preliminaries section.**

---

### Author Response · Authors · 2021-08-05
**Why should we study the probability space in the context of deep learning and information theory?**

We appreciate all reviewers' time and efforts in the paper. Here we post a comment for helping reviewers to understand the key idea of the paper.

**Background.**
The Information Bottleneck (IB) principle recently attracts great attention in explaining the internal mechanism of Deep Neural Networks (DNNs). In a word, The IB principle is used to explain how much information of input data or target labels is learned by DNNs.
The key of studying IB principle in DNNs is to accurately estimate the mutual information between a hidden layer and dataset.
Most previous works use non-parametric models, e.g., the empirical distribution, to indirectly estimate the mutual information.
However, **the mutual information estimation based on non-parametric models are inaccurate and different non-parametric models derives different information theoretic results, which weaken the validity of the IB explanations for DNNs**.

----

**Goal**: **(i) improve the interpretability of DNNs, (ii) derive accurate mutual information, and (iii) validate the IB principle in DNNs**.

----

**Key Idea: probability space is the foundation of information theoretic analysis**. Specifically, **probability space** provides a formal mathematical framework for modeling a **random process.**
It is the foundation of specifying the random variable and the distribution of the random process. This paper focuses on specifying the probability space of a hidden layer for accurately estimating the mutual information in DNNs.

Below, we use two simple examples (namely, flipping a coin and measuring the weight of a coin) to show that **probability space** plays a fundamental role in the information theoretic analysis.

**Random process**: $\qquad$ $\qquad$ $\qquad$ $\quad$ **Flipping a coin** $\qquad$ $\qquad$ **Measuring the weight of a randomly chosen coin**

---

**Entropy**: $\qquad$ $\qquad$ $\quad$  $H(X)$ $\qquad$ $p\log_2(\frac{1}{p})+(1-p)log_2\frac{1}{1-p}$ $\quad$ $-\int \mathcal{N}(x;\mu,\sigma^2)log_2\mathcal{N}(x;\mu,\sigma^2)dx$

$\quad$ $\uparrow$

**Distribution**: $\qquad$ $\quad$ $P(X)$ $\qquad$ $\quad$ $P(X=0)=p$ $\qquad$ $\qquad$ $P(X=x)=\mathcal{N}(x;\mu,\sigma^2)$

$\qquad$ $\qquad$ $\qquad$ $\quad$ $\qquad$ $\qquad$ $\quad$ $\quad$ $P(X=1)=1-p$

$\quad$ $\uparrow$

**Random Variable**: $\quad \text{ }$ $X: \Omega \rightarrow E$ $\qquad$ $\text{Head} \rightarrow 0$ $\qquad$ $\qquad$ $\quad$ $\text{Weight} \rightarrow \mathbb{R}$

$\qquad$ $\qquad$ $\qquad$ $\qquad$ $\qquad$ $\qquad$ $\qquad$ $\text{ }$ $\text{ Tail} \rightarrow 1$

$\quad$ $\uparrow$

**Probability space** $\qquad$ $\Omega$ $\qquad$ $\qquad$ $\{\text{Head}, \text{Tail}\}$ $\qquad$ $\qquad$ $\qquad$ $\quad$ $\mathbb{R}$

$(\Omega,\mathcal{F},P):$ $\quad$ $\qquad$ $\quad$ $\mathcal{F}$ $\qquad$ $\qquad$ $\sigma$-algebra

$\quad$ $\qquad$ $\qquad$ $\quad$ $\qquad \text{ }$ $P$ $\qquad$ $\qquad$ $P(\text{Head})=p$ $\qquad$ $\quad$ $\quad$ $P(\text{Weight})$ is a Gaussian  distribution $\mathcal{N}(\mu,\sigma^2)$

$\quad$ $\qquad$ $\qquad$ $\qquad$ $\qquad$ $\qquad$ $\quad$ $\quad$ $P(\text{Tail})=1-p$

**The above two examples indicate that clearly defining the probability space of a random process is the foundation of information theoretic analysis of the random process.
Specifically, if the probability space of a random process is specified, then the random variable and the distribution can be easily specified, thereby deriving accurate entropy of the two random process.**

**Neural networks can be viewed as a complex random process. In order to accurately measure the mutual information in networks (namely the complex random process), the most straightforward and fundamental approach is to follow the logic in the two examples above: defining the probability space of neural networks, specifying the corresponding random variable, and calculate the mutual information or entropy.**
However, neural networks are such complicated that few previous works directly define the probability space.
**In other words, the sample space and probability measure of neural networks are actually unclear.**

In most previous works, **non-parametric models** (e.g., the empirical distribution [1], the kernel density estimation [2], and Gaussian convolution [3]) are commonly used for mutual information estimation, because non-parametric models are simple and easy for practical implementation.
However, **non-parametric models only indirectly and implicitly derive/model the random variable and the distribution of a hidden layers.**
In addition, non-parametric models are known for poor mutual information estimation [4] in high-dimensional set, and sensitive to hyper-parameters [5].
Even worse, non-parametric models lead to certain theoretic issues, e.g., the infinite mutual information, in deep learning [2].**

**Considering the existing limitations and controversies of the information theoretic explanations for DNNs, it is extremely necessary to specify the probability space of a hidden layer to resolve the existing limitations.**

[1] Ravid Shwartz-Ziv and Naftali Tishby. Opening the black box of deep neural networks via information. arXiv preprint arXiv:1703.00810, 2017.

[2] Andrew Saxe, Yamini Bansal, Joel Dapello, Madhu Advani, Artemy Kolchinsky, Brendan Tracey, and David Cox. On the information bottleneck theory of deep learning. In International Conference on Representation Learning, 2018.

[3] Ziv Goldfeld, Ewout Van Den Berg, Kristjan Greenewald, Igor Melnyk, Nam Nguyen, Brian Kingsbury, and Yury Polyanskiy. Estimating information flow in deep neural networks. In Proceedings of the 36th International Conference on Machine Learning, volume 97, pages 2299–2308, 2019.

[4] Liam Paninski. Estimation of entropy and mutual information. Neural computation, 15(6):1191–1253, 2003.

[5] David McAllester and Karl Stratos. Formal limitations on the measurement of mutual information. In International Conference on Artificial Intelligence and Statistics, pages 875–884. PMLR, 2020.

---

**Results**. Based on the cross-correlation definition of convolution, we explain the activation $t_n = \sigma(<x, {\omega}_n>)$ as
$t_n$ measuring the cross-correlation between the input $x$ and the feature $\omega_n$ defined by the weights of the $n$-th neuron in a layer, and define the random process of a hidden layer.

Following the logic above, we introduce **the probability space $(\Omega, \mathcal{F}, P)$ for a hidden layer $\boldsymbol{t}$**, **specify the random variable $T$ and the distribution of a hidden layer $P(T|X)$**, and derive probabilistic explanations for the entire architecture of DNNs.

Based on the probabilistic representation, we **resolve the limitations of existing information theoretic explanations for DNNs**, derive **more accurate mutual information estimation for $I(X;T)$ and $I(Y;T)$ than the existing non-parametric models**, and **confirm the IB principle in DNNs**.

---

In summary, the logic of the paper can be summarized as follows.

**Resolve the theoretical limitations and Validate the Information Bottleneck (IB) explanations for DNNs**

$\qquad$ $\uparrow$

More accurate mutual information estimation: $I(X;T)$ and $I(Y;T)$

$\qquad$ $\uparrow$

Distribution $\qquad$ $\quad$ $P(T)$: $\qquad$ $P_{T|X}(n|x)=\frac{1}{Z}\exp[\sigma(<\omega_n, x>)]$

$\qquad$ $\uparrow$

Random Variable $\quad \text{ }$ $T: \Omega \rightarrow E$ $\quad$ $T(\omega_n) = n$

$\qquad$ $\uparrow$

**Probability space** $\qquad$ $\Omega$ $\qquad$ $\qquad$ $\\{\omega_n\\}_{n=1}^N$

$(\Omega,\mathcal{F},P):$ $\quad$ $\qquad$ $\quad$ $\mathcal{F}$ $\qquad$ $\qquad$ $\sigma$-algebra

$\quad$ $\qquad$ $\qquad$ $\quad$ $\qquad \text{ }$ $P$ $\qquad$ $\qquad$ $P(\omega_n|x)=\frac{1}{Z}\exp[\sigma(<\omega_n, x>)]$

Please refer Definition 1 in the paper for detail.

---
Hope the comment can help reviewers to understand the main idea of this paper. We respect the reviewer's opinion, but hope they can reconsider the review.

---

### Decision · Program_Chairs · 2021-09-27

**Decision:**

Reject

**Comment:**

The direction suggested by this paper is potentially interesting, but there was substantial agreement (not consensus) among the reviewers that this paper is not ready for publication. Overall, the reviewers felt (even ater substantial author-reviewer discussion) that the paper is hard to understand, the overall experiments are unclear, and the experiments are not compelling.